# A Mean-Field Theory for Kernel Alignment with Random Features in Generative Adversarial Networks

## Abstract

We propose a novel supervised learning method to optimize the kernel in maximum mean discrepancy generative adversarial networks (MMD GANs). Specifically, we characterize a distributionally robust optimization problem to compute a good distribution for the random feature model of Rahimi and Recht to approximate a good kernel function. Due to the fact that the distributional optimization is infinite dimensional, we consider a Monte-Carlo sample average approximation (SAA) to obtain a more tractable finite dimensional optimization problem. We subsequently leverage a particle stochastic gradient descent (SGD) method to solve the derived finite dimensional optimization problem. Based on a mean-field analysis, we then prove that the empirical distribution of the interactive particles system at each iteration of the SGD follows the path of the gradient descent flow on the Wasserstein manifold. We also establish the non-asymptotic consistency of the finite sample estimator. Our empirical evaluation on synthetic data-set as well as MNIST and CIFAR-10 benchmark data-sets indicates that our proposed MMD GAN model with kernel learning indeed attains higher inception scores well as Frèchet inception distances and generates better images compared to the generative moment matching network (GMMN) and MMD GAN with untrained kernels.

## 1 Introduction

A fundamental and long-standing problem in unsupervised learning systems is to capture the underlying distribution of data. While deep generative models such as Boltzmann machines Salakhutdinov & Hinton (2009) and auto-encoding variational Bayes Kingma & Welling (2013) accomplish this task to some extent, they are inadequate for many intractable probabilistic computations that arise in maximum likelihood estimation. Moreover, in many machine learning tasks such as caption generation Xu et al. (2015), the main objective is to obtain new samples rather than to accurately estimate the underlying data distribution. Generative adverserial network (GAN) Goodfellow et al. (2014) provides a framework to directly draw new samples without estimating data distribution. It consists of a deep feedforward network to generate new samples from a base distribution (*e.g.* Gaussian distribution), and a discriminator network to accept or reject the generated samples. However, training GAN requires finding a Nash equilibrium of a non-convex minimax game with continuous, high-dimensional parameters. Consequently, it is highly unstable and prone to miss modes Salimans et al. (2016); Che et al. (2016). To obtain more stable models, the generative moment matching networks (GMMNs) Li et al. (2015) are proposed, wherein instead of training a discriminator network, a non-parametric statistical hypothesis test is performed to accept or reject the generated samples via the computation of the kernel maximum mean discrepancy Gretton et al. (2007). While leveraging a statistical test simplifies the loss function for training GMMN, in practice, the diversity of generated samples by GMMN is highly sensitive to the choice of the kernel. Thus, to improve the sampling performance, the kernel function also needs to be jointly optimized with the generator. Rather than

optimizing the kernel directly, the MMD GAN model Li et al. (2017) is proposed in which an embedding function is optimized in conjunction with a fixed user-defined kernel (*e.g.* RBF Gaussian kernel). However, there are no theoretical guarantees that the user-defined kernel is the 'right' kernel for embedded features.

**Contributions**. To address the kernel model selection problem in MMD GAN Li et al. (2017), in this paper we put forth a novel framework to learn a good kernel function from training data. Our kernel learning approach is based on a distributional optimization problem to learn a good distribution for the random feature model of Rahimi and Recht Rahimi & Recht (2008; 2009) to approximate the kernel. Since optimization with respect to the distribution of random features is infinite dimensional, we consider a Monte Carlo approximation to obtain a more tractable finite dimensional optimization problem with respect to the samples of the distribution. We then use a particle stochastic gradient descent (SGD) to solve the approximated finite dimensional optimization problem. We provide a theoretical guarantee for the consistency of the finite sample-average approximations. Based on a mean-field analysis, we also show the consistency of the proposed particle SGD. In particular, we show that when the number of particles tends to infinity, the empirical distribution of the particles in SGD follows the path of the gradient descent flow of the distributional optimization problem on the Wasserstein manifold.

## 2 Preliminaries of MMD GANs

Assume we are given data $\{\boldsymbol{v}_i\}_{i=1}^n$ that are sampled from an unknown distribution $P_{\boldsymbol{V}}$ with the support $\mathcal{V}$. In many unsupervised tasks, we wish to attain new samples from the distribution $P_{\boldsymbol{V}}$ without directly estimating it. Generative Adversarial Network (GAN) Goodfellow et al. (2014) provides such a framework. In vanilla GAN, a deep network $\mathcal{G}(\cdot; \boldsymbol{W})$ parameterized by $\boldsymbol{W} \in \mathcal{W}$ is trained as a generator to transform the samples $\boldsymbol{Z} \sim P_{\boldsymbol{Z}}, \boldsymbol{Z} \in \mathcal{Z}$ from a user-defined distribution $P_{\boldsymbol{Z}}$ (*e.g.* Gaussian) into a new sample $\mathcal{G}(\boldsymbol{Z}; \boldsymbol{W}) \sim P_{\boldsymbol{W}}$, such that the distributions $P_{\boldsymbol{W}}$ and $P_{\boldsymbol{V}}$ are close. In addition, a discriminator network $\mathcal{D}(\cdot; \boldsymbol{\delta})$ parameterized by $\boldsymbol{\delta} \in \Delta$ is also trained to reject or accept the generated samples as a realization of the data distribution. The training of the generator and discriminator networks is then accomplished via solving a minimax optimization problem as below

$$\min_{\boldsymbol{W} \in \mathcal{W}} \max_{\boldsymbol{\delta} \in \Delta} \mathbb{E}_{P_{\boldsymbol{V}}}[\mathcal{D}(\boldsymbol{X}; \boldsymbol{\delta})] + \mathbb{E}_{P_{\boldsymbol{Z}}}[\log(1 - \mathcal{D}(\mathcal{G}(\boldsymbol{Z}; \boldsymbol{W}); \boldsymbol{\delta}))]. \tag{1}$$

In the high dimensional settings, the generator trained via the min-max program of equation 1 can potentially collapse to a single mode of distribution where it always emits the same point Che et al. (2016). To overcome this shortcoming, other adversarial generative models are proposed in the literature, which propose to modify or replace the discriminator network by a statistical two-sample test based on the notion of the maximum mean discrepancy which is defined below:

**Definition 2.1.** (Maximum Mean Discrepancy Gretton et al. (2007)) Let $(\mathcal{X}, d)$ be a metric space, $\mathcal{F}$ be a class of functions $f : \mathcal{X} \to \mathbb{R}$, and $P, Q \in \mathcal{B}(\mathcal{X})$ be two probability measures from the set of all Borel probability measures $\mathcal{B}(\mathcal{X})$ on $\mathcal{X}$. The maximum mean discrepancy (MMD) between the distributions $P$ and $Q$ with respect to the function class $\mathcal{F}$ is defined below

$$D_{\mathcal{F}}[P, Q] \stackrel{\text{def}}{=} \sup_{f \in \mathcal{F}} \int_{\mathcal{X}} f(\boldsymbol{x})(P - Q)(\mathrm{d}\boldsymbol{x}). \tag{2}$$

Different choices of the function class $\mathcal{F}$ in equation 2 yield different adversarial models such as Wasserstein GANs (WGAN) Arjovsky et al. (2017), $f$-GANs Nowozin et al. (2016), and GMMN and MMD GAN Li et al. (2017; 2015). In the latter two cases, the function class $\mathcal{F} \stackrel{\text{def}}{=} \{f : \mathcal{X} \to \mathbb{R} : \|f\|_{\mathcal{H}_{\mathcal{X}}} \leq 1\}$, where $\mathcal{H}_{\mathcal{X}}$ is a reproducing kernel Hilbert space (RKHS) of functions with a kernel $K : \mathcal{X} \times \mathcal{X} \to \mathbb{R}$, denoted by $(\mathcal{H}_{\mathcal{X}}, K)$. Then, the *squared* MMD loss in equation 2 as a

measure of the distance between the distributions $P_V$ and $P_W$ has the following expression

$$D_K[P_V, P_W] \stackrel{\text{def}}{=} \sup_{f:\mathcal{X}\to\mathbb{R}:\|f\|_{\mathcal{H}_{\mathcal{X}}}\leq 1} \int_{\mathcal{X}} f(\boldsymbol{x})(P_V - P_W)(\mathrm{d}\boldsymbol{x}) \tag{3}$$

$$= \mathbb{E}_{V,V'\sim P_V}[K(V;V')] + \mathbb{E}_{W,W'\sim P_W}[K(W,W')] - 2\mathbb{E}_{V\sim P_V,W\sim P_W}[K(V;W)], \tag{4}$$

where $\mathcal{X} = \mathcal{V} \cup \mathcal{W}$. Instead of training the generator via solving the minimax optimization in equation 1, the MMD GAN model of Li et al. (2015) proposes to optimize the discrepancy between two distributions via optimization of an embedding function $\iota : \mathbb{R}^D \mapsto \mathbb{R}^p, p \leq D$, *i.e.*,

$$\min_{\boldsymbol{W}\in\mathcal{W}} \max_{\iota\in\mathcal{Q}} \mathrm{MMD}_{k\circ\iota}[P_V, P_W], \tag{5}$$

where $k : \mathbb{R}^p \times \mathbb{R}^p \to \mathbb{R}$ is a user-defined fixed kernel. In Li et al. (2015), the proposal for the kernel $k : \mathbb{R}^p \times \mathbb{R}^p \to \mathbb{R}$ is a mixture of the Gaussians,

$$k \circ \iota(\boldsymbol{x}, \boldsymbol{y}) = k(\iota(\boldsymbol{x}), \iota(\boldsymbol{y})) = \sum_{i=1}^{m} \left( \frac{\|\iota(\boldsymbol{x}) - \iota(\boldsymbol{y})\|_2^2}{\sigma_i^2} \right), \tag{6}$$

where the bandwidth parameters $\sigma_1, \cdots, \sigma_m > 0$ are manually selected. Nevertheless, in practice there is no guarantee that the user-defined kernel $k(\iota(\boldsymbol{x}), q(\boldsymbol{y}))$ can capture the structure of the embedded features $\iota(\boldsymbol{x})$.

## 3 PROPOSED APPROACH: KERNEL LEARNING WITH RANDOM FEATURES FOR MMD GANS

In this section, we first expound our kernel learning approach. Then, we describe a novel MMD GAN model based on the proposed kernel learning approach.

### 3.1 ROBUST DISTRIBUTIONAL OPTIMIZATION FOR KERNEL LEARNING

To address the kernel model selection issue in MMD GAN Li et al. (2017), we consider a kernel optimization scheme with random features Rahimi & Recht (2008; 2009). Let $\varphi : \mathbb{R}^d \times \mathbb{R}^D \to [-1, 1]$ denotes the explicit feature maps and $\mu \in \mathcal{M}(\mathbb{R}^D)$ denotes a probability measure from the space of probability measures $\mathcal{M}(\mathbb{R}^D)$ on $\mathbb{R}^D$. The kernel function is characterized via the explicit feature maps using the following integral equation

$$K_\mu(\boldsymbol{x}, \boldsymbol{y}) = \mathbb{E}_\mu[\varphi(\boldsymbol{x}; \boldsymbol{\xi})\varphi(\boldsymbol{y}; \boldsymbol{\xi})] = \int_\Xi \varphi(\boldsymbol{x}; \boldsymbol{\xi})\varphi(\boldsymbol{y}; \boldsymbol{\xi})\mu(\mathrm{d}\boldsymbol{\xi}). \tag{7}$$

Let $\mathrm{MMD}_\mu[P_V, P_W] \stackrel{\text{def}}{=} \mathrm{MMD}_{K_\mu}[P_V, P_W]$. Then, the kernel optimization problem in can be formulated as a distribution optimization for random features, *i.e*,

$$\min_{\boldsymbol{W}\in\mathcal{W}} \sup_{\mu\in\mathcal{P}} \mathrm{MMD}_\mu[P_V, P_W]. \tag{8}$$

Here, $\mathcal{P}$ is the set of probability distributions corresponding to a kernel class $\mathcal{K}$. In the sequel, we consider $\mathcal{P}$ to be the distribution ball of radius $R$ as below

$$\mathcal{P} \stackrel{\text{def}}{=} \mathbb{B}_R^p(\mu_0) \stackrel{\text{def}}{=} \{\mu \in \mathcal{M}(\mathbb{R}^D) : W_p(\mu, \mu_0) \leq R\}, \tag{9}$$

where $\mu_0$ is a user-defined base distribution, and To establish the proof, we consider the $W_p(\cdot, \cdot)$ is the $p$-Wasserstein distance defined as below

$$W_p(\mu_1, \mu_2) \stackrel{\text{def}}{=} \left( \inf_{\pi\in\Pi(\mu_1,\mu_2)} \int_{\mathbb{R}^D\times\mathbb{R}^D} \|\boldsymbol{\xi}_1 - \boldsymbol{\xi}_2\|_2^p \mathrm{d}\pi(\boldsymbol{\xi}_1, \boldsymbol{\xi}_2) \right)^{\frac{1}{p}}, \tag{10}$$

where the infimum is taken with respect to all couplings $\pi$ of the measures $\mu, \mu_0 \in \mathcal{M}(\mathbb{R}^D)$, and $\Pi(\mu, \mu_0)$ is the set of all such couplings with the marginals $\mu$ and $\mu_0$.

The kernel MMD loss function in equation 8 is defined with respect to the unknown distributions of the data-set $P_V$ and the model $P_W$. Therefore, we construct an unbiased estimator for the MMD loss function in equation 8 based on the training samples. To describe the estimator, sample the labels from a uniform distribution $y_1, \cdots, y_n \sim_{\text{i.i.d.}} \text{Uniform}\{-1, +1\}$, where we assume that the number of positive and negative labels are balanced. In particular, consider the set of positive labels $\mathcal{I} = \{i \in \{1, 2, \cdots, n\} : y_i = +1\}$, and negative labels $\mathcal{J} = \{1, 2, \cdots, n\}/\mathcal{I}$, where their cardinality is $|\mathcal{I}| = |\mathcal{J}| = \frac{n}{2}$. We consider the following assignment of labels:

- *Positive class labels*: If $y_i = +1$, sample the corresponding feature map from data-distribution $\boldsymbol{x}_i = \boldsymbol{v}_i \sim P_V$.
- *Negative class labels*: If $y_i = -1$, sample from the corresponding feature map from the generated distribution $\boldsymbol{x}_i = \mathcal{G}(\boldsymbol{Z}_i, \boldsymbol{W}) \sim P_W, \boldsymbol{Z}_i \sim P_Z$.

By this construction, the joint distribution of features and labels $P_{Y,\boldsymbol{X}}$ has the marginals $P_{\boldsymbol{X}|Y=+1} = P_V$, and $P_{\boldsymbol{X}|Y=-1} = P_W$. Moreover, the following statistic, known as the *kernel alignment* in the literature (see, *e.g.*, Sinha & Duchi (2016); Cortes et al. (2012)), is an unbiased estimator of the MMD loss in equation 8,

$$\min_{\boldsymbol{W} \in \mathcal{W}} \sup_{\mu \in \mathcal{P}} \widehat{\text{MMD}}_\mu \left[ P_V, P_W \right] \stackrel{\text{def}}{=} \frac{8}{n(n-1)} \sum_{1 \leq i < j \leq n} y_i y_j K_\mu(\boldsymbol{x}_i, \boldsymbol{x}_j). \tag{11}$$

See Appendix C.1 for the related proof. The kernel alignment in equation 11 can also be viewed through the lens of the risk minimization

$$\min_{\boldsymbol{W} \in \mathcal{W}} \inf_{\mu \in \mathcal{P}} \widehat{\text{MMD}}_\mu^\alpha \left[ P_V, P_W \right] \stackrel{\text{def}}{=} \frac{8}{n(n-1)\alpha} \sum_{1 \leq i < j \leq n} (\alpha y_i y_j - K_\mu(\boldsymbol{x}_i, \boldsymbol{x}_j))^2 \tag{12a}$$

$$= \frac{8}{n(n-1)\alpha} \sum_{1 \leq i < j \leq n} (\alpha y_i y_j - \mathbb{E}_\mu[\varphi(\boldsymbol{x}_i; \boldsymbol{\xi})\varphi(\boldsymbol{x}_j; \boldsymbol{\xi})])^2. \tag{12b}$$

Here, $\alpha > 0$ is a scaling factor that determines the separation between feature vectors, and $\boldsymbol{K}_* \stackrel{\text{def}}{=} \alpha \boldsymbol{y}\boldsymbol{y}^T$ is the ideal kernel that provides the maximal separation between the feature vectors over the training data-set, *i.e.*, $K_*(\boldsymbol{x}_i, \boldsymbol{x}_j) = \alpha$ when features have identical labels $y_i = y_j$, and $K_*(\boldsymbol{x}_i, \boldsymbol{x}_j) = -\alpha$ otherwise. Upon expansion of the risk function in equation 12, it can be easily shown that it reduces to the kernel alignment in equation 11 when $\alpha \to +\infty$. Intuitively, the risk minimization in equation 12 gives a feature space in which pairwise distances are similar to those in the output space $\mathcal{Y} = \{-1, +1\}$.

## 3.2 SAA FOR DISTRIBUTIONAL OPTIMIZATION

The distributional optimization problem in equation 8 is infinite dimensional, and thus cannot be solved directly. To obtain a tractable optimization problem, instead of optimizing with respect to the distribution $\mu$ of random features, we optimize the i.i.d. samples (particles) $\boldsymbol{\xi}^1, \cdots, \boldsymbol{\xi}^N \sim_{\text{i.i.d.}} \mu$ generated from the distribution. The empirical distribution of these particles is accordingly defined as follows

$$\widehat{\mu}^N(\boldsymbol{\xi}) \stackrel{\text{def}}{=} \frac{1}{N} \sum_{k=1}^N \delta(\boldsymbol{\xi} - \boldsymbol{\xi}^k), \tag{13}$$

where $\delta(\cdot)$ is the Dirac's delta function concentrated at zero. In practice, the optimization problem in equation 12 is solved via the Monte-Carlo sample average approximation of the objective function,

$$\min_{\boldsymbol{W} \in \mathcal{W}} \min_{\widehat{\mu}^N \in \mathcal{P}_N} \widehat{\text{MMD}}_{\widehat{\mu}^N}^\alpha \left[ P_V, P_W \right] = \frac{8}{n(n-1)\alpha} \sum_{1 \leq i < j \leq n} \left( \alpha y_i y_j - \frac{1}{N} \sum_{k=1}^N \varphi(\boldsymbol{x}_i; \boldsymbol{\xi}^k)\varphi(\boldsymbol{x}_j; \boldsymbol{\xi}^k) \right)^2, \tag{14}$$

where $\mathcal{P}_N \stackrel{\text{def}}{=} \mathbb{B}_R^N(\widehat{\mu}_0^N) = \left\{ \widehat{\mu}^N \in \mathcal{M}(\mathbb{R}^D) : W_p(\widehat{\mu}^N, \widehat{\mu}_0^N) \le R \right\}$, and $\widehat{\mu}_0^N$ is the empirical measure associated with the initial samples $\boldsymbol{\xi}_0^1, \cdots, \boldsymbol{\xi}_0^N \sim_{\text{i.i.d.}} \mu_0$. The empirical objective function in equation 14 can be optimized with respect to the samples $\boldsymbol{\xi}^1, \cdots, \boldsymbol{\xi}^N$ using the particle stochastic gradient descent. For the optimization problem in equation 14, the (projected) stochastic gradient descent (SGD) takes the following recursive form,[1]

$$\boldsymbol{\xi}_{m+1}^k = \boldsymbol{\xi}_m^k - \frac{\eta}{N} \left( y_m \widetilde{y}_m - \frac{1}{\alpha N} \sum_{k=1}^N \varphi(\boldsymbol{x}_m; \boldsymbol{\xi}_m^k) \varphi(\widetilde{\boldsymbol{x}}_m; \boldsymbol{\xi}_m^k) \right) \nabla_{\boldsymbol{\xi}} \left( \varphi(\boldsymbol{x}_m; \boldsymbol{\xi}_m^k) \varphi(\widetilde{\boldsymbol{x}}_m; \boldsymbol{\xi}_m^k) \right), \quad (15a)$$

for $k = 1, 2, \cdots, N$, where $(y_m, \boldsymbol{x}_m), (\widetilde{y}_m, \widetilde{\boldsymbol{x}}_m) \sim_{\text{i.i.d}} P_{\boldsymbol{x}, y}$ and $\eta \in \mathbb{R}_{>0}$ denotes the learning rate of the algorithm, and the initial particles are $\boldsymbol{\xi}_0^1, \cdots, \boldsymbol{\xi}_0^N \sim_{\text{i.i.d.}} \mu_0$. At each iteration of the SGD dynamic in equation 15, a feasible solution for the inner optimization of the empirical risk function in equation 14 is generated via the empirical measure

$$\widehat{\mu}_m^N(\boldsymbol{\xi}) = \frac{1}{N} \sum_{k=1}^N \delta(\boldsymbol{\xi} - \boldsymbol{\xi}_m^k). \quad (16)$$

Indeed, we prove in Section 4 that for an appropriate choice of the learning rate $\eta > 0$, the empirical measure in equation 16 remains inside the distribution ball $\widehat{\mu}_m^N \in \mathcal{P}_N$ for all $m \in [0, NT] \cap \mathbb{N}$, and is thus a feasible solution for the empirical risk minimization equation 14 (see Corollary 4.2.1 in Section 4).

### 3.3 PROPOSED MMD GAN WITH KERNEL LEARNING

In Algorithm 1, we describe the proposed method MMD GAN model with the kernel learning approach described earlier. Algorithm 1 has an inner loop for the kernel training and an outer loop for training the generator, where we employ RMSprop Tieleman & Hinton (2012). Our proposed MMD GAN model is distinguished from MMD GAN of Li et al. (2017) in that we learn a good kernel function in equation 17 of the inner loop instead of optimizing the embedding function that is implemented by an auto-encoder. However, we mention that our kernel learning approach is compatible with the auto-encoder implementation of Li et al. (2017) for the dimensionality reduction of features (and particles). In the case of including an auto-encoder, the inner loop in Algorithm 1 must be modified to add an additional step for training the auto-encoder. However, to convey the main ideas more clearly, the training step of the auto-encoder is omitted from Algorithm 1.

### 3.4 COMPUTATIONAL COMPLEXITY ANALYSIS

Sampling the labels $y, \widetilde{y} \sim_{\text{i.i.d.}}$ Uniform$\{-1, +1\}$ require $\mathcal{O}(1)$ complexity, while sampling $\widetilde{\boldsymbol{x}}|\widetilde{y} = +1, \boldsymbol{x}|y = +1 \sim P_{\boldsymbol{V}}$ and $\widetilde{\boldsymbol{x}}|\widetilde{y} = +1, \boldsymbol{x}|y = -1 \sim P_{\boldsymbol{W}}$ has a complexity of $\mathcal{O}(d)$. The computation of the stochastic gradient step in equation 17 requires computing the sum $\frac{1}{N} \sum_{i=1}^N \varphi(\boldsymbol{x}; \boldsymbol{\xi}^k) \varphi(\widetilde{\boldsymbol{x}}; \boldsymbol{\xi}^k)$. This can be done as a separate preprocessing step prior to executing the SGD in equation 17, and requires preparing the vectors $\boldsymbol{\varphi} \stackrel{\text{def}}{=} [\varphi(\boldsymbol{x}; \boldsymbol{\xi}^1), \cdots, \varphi(\boldsymbol{x}; \boldsymbol{\xi}^N)]$ and $\widetilde{\boldsymbol{\varphi}} \stackrel{\text{def}}{=} [\varphi(\widetilde{\boldsymbol{x}}; \boldsymbol{\xi}^1), \cdots, \varphi(\widetilde{\boldsymbol{x}}; \boldsymbol{\xi}^N)]$. Using the random feature model of Rahimi and Recht Rahimi & Recht (2008; 2009), where $\varphi(\boldsymbol{x}; \boldsymbol{\xi}) = \sqrt{2} \cos(\boldsymbol{x}^T \boldsymbol{\xi} + b)$. Here $b \sim$ Uniform$[-\pi, +\pi]$, the complexity of computing the vectors $\boldsymbol{\varphi}$ and $\widetilde{\boldsymbol{\varphi}}$ is of the order $\mathcal{O}(Nd)$ on a single processor. However, this construction is trivially parallelizable. Furthermore, computation can be sped up even further for certain distributions $\mu_0$. For example, the Fastfood technique can approximate $\boldsymbol{\varphi}$ and $\tilde{\boldsymbol{\varphi}}$ in $O(N \log(d))$

---

[1]To avoid clutter in our subsequent analysis, the normalization factor $\frac{16}{n(n-1)}$ of the gradient is omitted by modifying the step-size $\eta$.

---

**Algorithm 1** MMD GAN with a supervised kernel learning Method (Monte-Carlo Approach)

---

**Inputs:** The learning rates $\tilde{\eta}, \eta > 0$ , the number of iterations of discriminator per generator update $T \in \mathbb{N}$, the batch-size $n$, the number of random features $N \in \mathbb{N}$. Regularization parameter $\alpha > 0$.

**while** $\boldsymbol{W}$ has not converged **do**

  **for** $t = 1, 2, \cdots, T$ **do**

    Sample the labels $y, \widetilde{y} \sim_{\text{i.i.d}} \text{Uniform}\{-1, 1\}$.

    Sample the features $\boldsymbol{x}|y = +1 \sim P_{\boldsymbol{V}}$, and $\boldsymbol{x}|y = -1 \sim P_{\boldsymbol{W}}$. Similarly, $\widetilde{\boldsymbol{x}}|\widetilde{y} = +1 \sim P_{\boldsymbol{V}}$, and $\widetilde{\boldsymbol{x}}|\widetilde{y} = -1 \sim P_{\boldsymbol{W}}$.

    For all $k = 1, 2, \cdots, N$, update the particles,

$$\boldsymbol{\xi}^k \leftarrow \boldsymbol{\xi}^k - \frac{\eta}{N} \left( \alpha y \widetilde{y} - \frac{1}{N} \sum_{k=1}^{N} \varphi(\boldsymbol{x}; \boldsymbol{\xi}^k) \varphi(\widetilde{\boldsymbol{x}}; \boldsymbol{\xi}^k) \right) \nabla_{\boldsymbol{\xi}} \left( \varphi(\boldsymbol{x}; \boldsymbol{\xi}^k) \varphi(\widetilde{\boldsymbol{x}}; \boldsymbol{\xi}^k) \right), \tag{17}$$

  **end for**

  Sample a balanced minibatch of labels $\{y_i\}_{i=1}^n \sim_{\text{i.i.d.}} \text{Uniform}\{-1, +1\}$.

  Sample the minibatch $\{\boldsymbol{x}\}_{i=1}^n$ such that $\boldsymbol{x}_i|y_i = +1 \sim P_{\boldsymbol{V}}$, and $\boldsymbol{x}_i|y_i = -1 \sim P_{\boldsymbol{W}}$ for all $i = 1, 2, \cdots, n$.

  Update the generator

$$\boldsymbol{g}_{\boldsymbol{W}} \leftarrow \nabla_{\boldsymbol{W}} \widehat{D}_{\widehat{\mu}^N}^\alpha \left[ P_{\boldsymbol{V}}, P_{\boldsymbol{W}} \right], \quad \widehat{\mu}^N = \frac{1}{N} \sum_{k=1}^{N} \delta(\boldsymbol{\xi} - \boldsymbol{\xi}^k). \tag{18a}$$

$$\boldsymbol{W} \leftarrow \boldsymbol{W} - \tilde{\eta} \text{RMSprop}(\boldsymbol{g}_{\boldsymbol{W}}, \boldsymbol{W}). \tag{18b}$$

**end while**

---

time for the Gaussian kernel Le et al. (2013). Updating each particle in equation 17, involves the computation of the gradient $\nabla_{\boldsymbol{\xi}}(\varphi(\boldsymbol{\xi}; \boldsymbol{\xi}^k)\varphi(\widetilde{\boldsymbol{\xi}}; \boldsymbol{\xi}^k)))$ which is $\mathcal{O}(d)$. Thus, the complexity of one iteration of SGD for all the particles is $\mathcal{O}(Nd)$.

Overall, one step of the kernel learning has a complexity of $\mathcal{O}(Nd)$. On the other hand, to attain $\varepsilon$-suboptimal solution $\max_{k=1,2,\cdots,N} \|\boldsymbol{\xi}^k - \boldsymbol{\xi}_*^k\| \le \varepsilon$, the SGD requires has the sample complexity $\mathcal{O}\left(\log(\frac{1}{\varepsilon})\right)$. Consequently, the computational complexity of the kernel learning is of the order of $\mathcal{O}(Nd\log(\frac{1}{\varepsilon}))$. To compare this complexity with that of MMD GAN is of the order $\mathcal{O}\left(B^2\ell\log(\frac{1}{\varepsilon})\right)$, where $B$ is the batch size for approximation of the population MMD, and $\ell$ is the number of kernel mixtures.

## 4 CONSISTENCY AND A MEAN-FIELD ANALYSIS

In this section, we provide theoretical guarantees for the consistency of various approximations we made to optimize the population MMD loss function in equation 8. We defer the proofs of the following theoretical results to AppendixC. The main assumptions ((**A.1**),(**A.2**), and (**A.3**)) underlying our theoretical results are also stated in the same section.

**Consistency of finite-sample estimate**: In this part, we prove that the solution to finite sample optimization problem in equation 14 approaches its population optimum in equation 8 as the number of data points as well as the number of random feature samples tends to infinity.

**Theorem 4.1.** (NON-ASYMPTOTIC CONSISTENCY OF FINITE-SAMPLE ESTIMATOR) *Suppose conditions* (**A.1**)-(**A.3**) *of Appendix C are satisfied. Consider the distribution balls* $\mathcal{P}$ *and* $\mathcal{P}_N$ *that are defined with respect to the 2-Wasserstein distance* $(p = 2)$*. Furthermore, consider the optimal MMD values of the population optimization and its finite sample estimate*

$$(\boldsymbol{W}_*, \mu_*) \overset{def}{=} \arg\min_{\boldsymbol{W} \in \mathcal{W}} \arg\sup_{\mu \in \mathcal{P}} \text{MMD}_\mu[P_{\boldsymbol{V}}, P_{\boldsymbol{W}}]. \tag{19a}$$

$$(\widehat{\boldsymbol{W}}_*^N, \widehat{\mu}_*^N) \overset{def}{=} \arg\min_{\boldsymbol{W} \in \mathcal{W}} \arg\inf_{\widehat{\mu}^N \in \mathcal{P}_N} \widehat{\text{MMD}}_{\widehat{\mu}^N}^\alpha[P_{\boldsymbol{V}}, P_{\boldsymbol{W}}], \tag{19b}$$

*respectively. Then, with the probability of (at least) $1 - 3\varrho$ over the training data samples $\{(\boldsymbol{x}_i, y_i)\}_{i=1}^{n}$ and the random feature samples $\{\boldsymbol{\xi}_0^k\}_{k=1}^{N}$, the following non-asymptotic bound holds*

$$\left| \mathrm{MMD}_{\mu_*}[P_{\boldsymbol{V}}, P_{\boldsymbol{W}_*}] - \mathrm{MMD}_{\widehat{\mu}_*^N}[P_{\boldsymbol{V}}, P_{\widehat{\boldsymbol{W}}_*^N}] \right| \tag{20}$$

$$\leq \sqrt{\frac{L^2(d+2)}{N}} \ln^{\frac{1}{2}}\left(\frac{2^8 N \mathrm{diam}^2(\mathcal{X})}{\varrho}\right) + 2\max\left\{\frac{c_1 L^2}{n} \ln^{\frac{1}{2}}\left(\frac{4}{\varrho}\right), \frac{c_2 R L^4}{n^2} \ln\left(\frac{4 e^{\frac{L^4}{9}}}{\varrho}\right)\right\} + \frac{8L^2}{\alpha},$$

*where $c_1 = 3^{\frac{1}{4}} \times 2^4$, and $c_2 = 9 \times 2^{11}$.*

The proof of Theorem 4.1 is presented in Appendix C.1.

Notice that there are three key parameters involved in the upper bound of Theorem 4.1. Namely, the number of training samples $n$, the number of random feature samples $N$, and the regularization parameter $\alpha$. The upper bound in equation 20 thus shows that when $n, N, \alpha \to +\infty$, the solution obtained from solving the empirical risk minimization in equation 12 yields a MMD population value tending to the optimal value of the distributional optimization in equation 8.

**Consistency of particle SGD for solving distributional optimization**. The consistency result of Theorem 4.1 is concerned with the MMD value of the optimal empirical measure $\widehat{\mu}_*^N(\boldsymbol{\xi}) = \frac{1}{N}\sum_{k=1}^{N}\delta(\boldsymbol{\xi} - \boldsymbol{\xi}_*^k)$ of the empirical risk minimization equation 14. In practice, the particle SGD is executed for a few iterations and its values are used as an estimate for $(\boldsymbol{\xi}_*^1, \cdots, \boldsymbol{\xi}_*^N)$. Consequently, it is desirable to establish a similar consistency type result for the particle SGD estimates $(\boldsymbol{\xi}_m^1, \cdots, \boldsymbol{\xi}_m^N)$ at the $m$-th iteration. To reach this objective, we define the scaled empirical measure as follows

$$\mu_t^N = \widehat{\mu}_{\lfloor Nt \rfloor}^N = \frac{1}{N}\sum_{k=1}^{N}\delta(\boldsymbol{\xi} - \boldsymbol{\xi}_{\lfloor Nt \rfloor}), \quad 0 \leq t \leq T. \tag{21}$$

At any time $t$, the scaled empirical measure $\mu_t^N$ is a random element, and thus $(\mu_t^N)_{0 \leq t \leq T}$ is a measured-valued stochastic process. Therefore, we characterize the evolution of its Lebesgue density $p_t^N(\boldsymbol{\xi}) \stackrel{\mathrm{def}}{=} \mu_t^N(\mathrm{d}\boldsymbol{\xi})/\mathrm{d}\boldsymbol{\xi}$ in the following theorem:

**Theorem 4.2.** (MCKEAN-VLASOV MEAN-FIELD PDE) *Suppose conditions $(\mathbf{A}.1)$-$(\mathbf{A}.3)$ of Appendix C are satisfied. Further, suppose that the Radon-Nikodyme derivative $q_0(\boldsymbol{\xi}) = \mu_0(\mathrm{d}\boldsymbol{\xi})/\mathrm{d}\boldsymbol{\xi}$ exists. Then, there exists a unique solution $(p_t^*(\boldsymbol{\xi}))_{0 \leq t \leq T}$ to the following non-linear partial differential equation*

$$\begin{cases} \dfrac{\partial p_t(\boldsymbol{\xi})}{\partial t} &= -\dfrac{\eta}{\alpha}\iint_{\mathcal{X}\times\mathcal{Y}}\left(\int_{\mathbb{R}^p}\varphi(\boldsymbol{x}, \widetilde{\boldsymbol{\xi}})\varphi(\widetilde{\boldsymbol{x}}, \widetilde{\boldsymbol{\xi}})p_t(\widetilde{\boldsymbol{\xi}})\mathrm{d}\widetilde{\boldsymbol{\xi}} - \alpha y\widetilde{y}\right)\nabla_{\boldsymbol{\xi}}(p_t(\boldsymbol{\xi})\nabla_{\boldsymbol{\xi}}(\varphi(\boldsymbol{x};\boldsymbol{\xi})\varphi(\widetilde{\boldsymbol{x}};\boldsymbol{\xi}))\mathrm{d}P_{\boldsymbol{X},Y}^{\otimes 2}, \\ p_0(\boldsymbol{\xi}) &= q_0(\boldsymbol{\xi}). \end{cases}$$

$$\tag{22}$$

*Moreover, the measure-valued process $\{(\mu_t^N)_{0 \leq t \leq T}\}_{N \in \mathbb{N}}$ defined in equation 21 converges (weakly) to the unique solution $\mu_t^*(\boldsymbol{\xi}) = p_t^*(\boldsymbol{\xi})\mathrm{d}\boldsymbol{\xi}$ as the number of particles tend to infinity $N \to \infty$. [2].*

Due to the mean-field analysis of Theorem 4.2, we can prove that the empirical measure $\widehat{\mu}_m^N$ of the particles in SGD dynamic equation 15 remains inside the feasible distribution ball $\mathcal{P}_N$:

**Corollary 4.2.1.** *Consider the learning rate $\eta = \mathcal{O}\left(\frac{R^p}{T\sqrt{NT}\log(2/\delta)}\right)$ for the SGD in equation 15. Then, the empirical measure $\widehat{\mu}_m^N$ of the particles remains inside the distributional ball $\widehat{\mu}_m^N \in \mathcal{P}_N = \{\widehat{\mu}^N \in \mathcal{M}(\mathbb{R}^D) : W_p(\widehat{\mu}^N, \widehat{\mu}_0^N) \leq R\}$ for all $m \in [0, NT] \cap \mathbb{N}$, with the probability of (at least) $1 - \delta$.*

---

[2]The notion of the weak convergence of a sequence of empirical measures is formally defined in Supplementary.

Let us make two remarks about the PDE in equation 22. First, the seminal works of Otto Otto (2001), and Jordan, *et al.* Jordan et al. (1998) establishes a deep connection between the McKean-Vlasov type PDEs specified in equation 22 and the gradient flow on the Wasserstein manifolds. More specifically, the PDE equation in equation 22 can be thought of as the minimization of the energy functional

$$\inf_{\mu \in \mathcal{M}(\mathbb{R}^D)} E_\alpha(p_t(\boldsymbol{\xi})) \overset{\text{def}}{=} \frac{1}{\alpha} \int_{\mathbb{R}^D} R_\alpha(\boldsymbol{\xi}, p_t(\boldsymbol{\xi})) p_t(\boldsymbol{\xi}) \mathrm{d}\boldsymbol{\xi} \tag{23a}$$

$$R_\alpha(\boldsymbol{\xi}, p_t(\boldsymbol{\xi})) \overset{\text{def}}{=} -\alpha (\mathbb{E}_{P_{\boldsymbol{X},Y}}[y\varphi(\boldsymbol{x}; \boldsymbol{\xi})])^2 + \mathbb{E}_{\widetilde{\boldsymbol{\xi}} \sim p_t} \left[ \left( \mathbb{E}_{P_{\boldsymbol{X}}}[\varphi(\boldsymbol{x}; \boldsymbol{\xi})\varphi(\boldsymbol{x}; \widetilde{\boldsymbol{\xi}})] \right)^2 \right], \tag{23b}$$

using the following gradient flow dynamics

$$\frac{\mathrm{d}p_t(\boldsymbol{\xi})}{\mathrm{d}t} = -\eta \cdot \mathrm{grad}_{p_t} E_\alpha(p_t(\boldsymbol{\xi})), \quad p_0(\boldsymbol{\xi}) = q_0(\boldsymbol{\xi}), \tag{24}$$

where $\mathrm{grad}_{p_t} E(p_t(\boldsymbol{\xi})) = \nabla_{\boldsymbol{\xi}} \cdot (p_t(\boldsymbol{\xi})\nabla_{\boldsymbol{\xi}} R_\alpha(p_t(\boldsymbol{\xi})))$ is the Riemannian gradient of $R_\alpha(\mu_t(\boldsymbol{\xi}))$ with respect to the metric of the Wasserstein manifold . This shows that when the number of particles in particle SGD equation 15 tends to infinity ($N \rightarrow +\infty$), their empirical distribution follows a gradient descent path for minimization of the population version (with respect to data samples) of the distributional risk optimization in equation 12. In this sense, the particle SGD is the 'consistent' approximation algorithm for solving the distributional optimization.

## 4.1 RELATED WORKS

The mean-field description of SGD dynamics has been studied in several prior works for different information processing tasks. Wang *et al.* Wang et al. (2017) consider the problem of online learning for the principal component analysis (PCA), and analyze the scaling limits of different online learning algorithms based on the notion of *finite exchangeability*. In their seminal papers, Montanari and co-authors Mei et al. (2018); Javanmard et al. (2019); Mei et al. (2019) consider the scaling limits of SGD for training a two-layer neural network, and characterize the related Mckean-Vlasov PDE for the limiting distribution of the empirical measure associated with the weights of the input layer. Similar mean-field type results for two-layer neural networks are also studied recently in Rotskoff & Vanden-Eijnden (2018); Sirignano & Spiliopoulos (2018). Our work is also related to the unpublished work of Wang, *et al.* Wang et al., which proposes a solvable model of GAN and analyzes the scaling limits. However, our GAN model is different from Wang et al. and is based on the notion of the kernel MMD. Our work is also closely related to the recent work of Li, *et al* Li et al. (2019) which proposes an implicit kernel learning method based on the following kernel definition

$$K_h(\iota(\boldsymbol{x}), \iota(\boldsymbol{y})) = \mathbb{E}_{\boldsymbol{\xi} \sim \mu_0} \left[ e^{(ih(\boldsymbol{\xi})(\iota(\boldsymbol{x}) - \iota(\boldsymbol{y})))} \right], \tag{25}$$

where $\mu_0$ is a user defined base distribution, and $h \in \mathcal{H}$ is a function that transforms the base distribution $\mu_0$ into a distribution $\mu$ that provides a better kernel. Therefore, the work of Li, *et al* Li et al. (2019) *implicitly* optimizes the distribution of random features via transforming a random variable with a function. In contrast, the proposed distributional optimization framework in this paper optimizes the distribution of random feature *explicitly*, via optimizing their empirical measures. Perhaps more importantly from a practical perspective is the fact that our kernel learning approach does not require the user-defined function class $\mathcal{H}$. Moreover, our particle SGD method in equation 15 obviates tuning hyper-parameters related to the implicit kernel learning method such as the gradient penalty factor and the variance constraint factor (denoted by $\lambda_{GP}$ and $\lambda_h$, respectively, in Algorithm 1 of Li et al. (2019)).

## 5 EMPIRICAL EVALUATION

### 5.1 SYNTHETIC DATA-SET

Due to the space limitation, the experiments on the synthetic data are deferred to Appendix A.

## 5.2 Performance on benchmark datasets

We evaluate our kernel learning approach on large-scale benchmark data-sets. We train our MMD GAN model on two distinct types of datasets, namely on MNIST LeCun et al. (1998) and CIFAR-10 LeCun et al. (1998), where the size of training instances are $60 \times 10^3$ and $50 \times 10^3$, respectively. All the generated samples are from a fixed noise random vectors and are not singled out.

**Implementation and hyper-parameters.** We implement Algorithm 1 as well as MMD GAN Li et al. (2017) in `Pytorch` using NVIDIA Titan V100 32GB graphics processing units (GPUs). The source code of Algorithm 1 is built upon the code of Li et al. (2017), and retains the auto-encoder implementation. In particular, we use a sequential training of the auto-encoder and kernel as explained in the Synthetic data in Section A of Supplementary. For a fair comparison, our hyper-parameters are adjusted as in Li et al. (2017), *i.e.*, the learning rate of 0.00005 is considered for RMSProp Tieleman & Hinton (2012). Moreover, the batch-size for training the generator and auto-encoder is $n = 64$. The learning rate of particle SGD is tuned to $\eta = 10$.

**Random Feature Maps.** To approximate the kernel, we use the the random feature model of Rahimi and Recht Rahimi & Recht (2008; 2009), where $\varphi(\boldsymbol{x}; \boldsymbol{\xi}) = \sqrt{2}\cos(\boldsymbol{x}^T\boldsymbol{\xi} + b)$. Here $b \sim$ Uniform$\{-\pi, +\pi\}$ is a random bias term.

**Practical considerations.** When data-samples $\{\boldsymbol{V}_i\} \in \mathbb{R}^d$ are high dimensional (as in CIFAR-10), the particles $\boldsymbol{\xi}^1, \cdots, \boldsymbol{\xi}^N \in \mathbb{R}^D, D = d$ in SGD equation 15 are also high-dimensional. To reduce the dimensionality of the particles, we apply an auto-encoder architecture similar to Li et al. (2017), and train our kernel on top of learned embedded features. More specifically, in our simulations, we train an auto-encoder where the dimensionality of the latent space is $h = 10$ for MNIST, and $h = 128$ (thus $D = d = 128$) for CIFAR-10. Therefore, the particles $\boldsymbol{\xi}^1, \cdots, \boldsymbol{\xi}^N$ in subsequent kernel training phase have the dimension of $D = 10$, and $D = 128$, respectively.

**Choice of the scaling parameter $\alpha$.** There is a trade-off in the choice of $\alpha$. While for large values of $\alpha$, the kernel is better able to separate data-samples from generated samples, in practice, it slows down the convergence of particle SGD. This is due to the fact that the coupling between the particle dynamics in equation 15 decrease as $\alpha$ increase. The scaling factor is set to be $\alpha = 1$ in all the following experiments.

**Qualitative comparison.** We now show that *without* the bandwidth tuning for Gaussian kernels and using the particle SGD to learn the kernel, we can attain better visual results on benchmark data-sets. In Figure 1, we show the generated samples on CIFAR-10 and MNIST data-sets, using our Algorithm 1, MMD GAN Li et al. (2017) with a mixed and homogeneous Gaussian RBF kernels, and GMMN Li et al. (2015).

Figure 1(a) shows the samples from Algorithm 1, Figure 1(b) shows the samples from MMD GAN Li et al. (2017) with a mixture RBF Gaussian kernel $\kappa(\boldsymbol{x}, \boldsymbol{y}) = \sum_{k=1}^{5} \kappa_{\sigma_k}(\boldsymbol{x}, \boldsymbol{y})$, where $\sigma_k \in \{1, 2, 4, 8, 16\}$ are the bandwidths of the Gaussian kernels that are fine tuned and optimized. We observe that our MMD GAN with *automatic* kernel learning visually attains similar results to MMD GAN Li et al. (2017) which requires *manual* tuning of the hyper-parameters. In Figure 1(c), we show the MMD GAN result with a single kernel RBF Gaussian kernel whose bandwidth is manually tuned at $\sigma = 16$. Lastly, in Figure 1(d), we show the samples from GMMN Li et al. (2015) which does not exploit an auto-encoder or kernel training. Clearly, GMMN yield a poor results compared to other methods due to high dimensionality of features, as well as the lack of an efficient method to train the kernel.

On MNIST data-set in Figure 1,(e)-(h), the difference between our method and MMD GAN Li et al. (2017) is visually more pronounced. We observe that without a manual tuning of the kernel bandwidth and by using the particle SGD equation 15 to optimize the kernel, we attain better generated images in Figure 1(e), compared to MMD GAN with mixed RBF Gaussian kernel and *manual* bandwidth tuning in Figure 1(f). Moreover, using a single RBF Gaussian kernel yields a poor result regardless of the choice of its bandwidth. The generated images from GMMN is also shown in Figure 1(h).

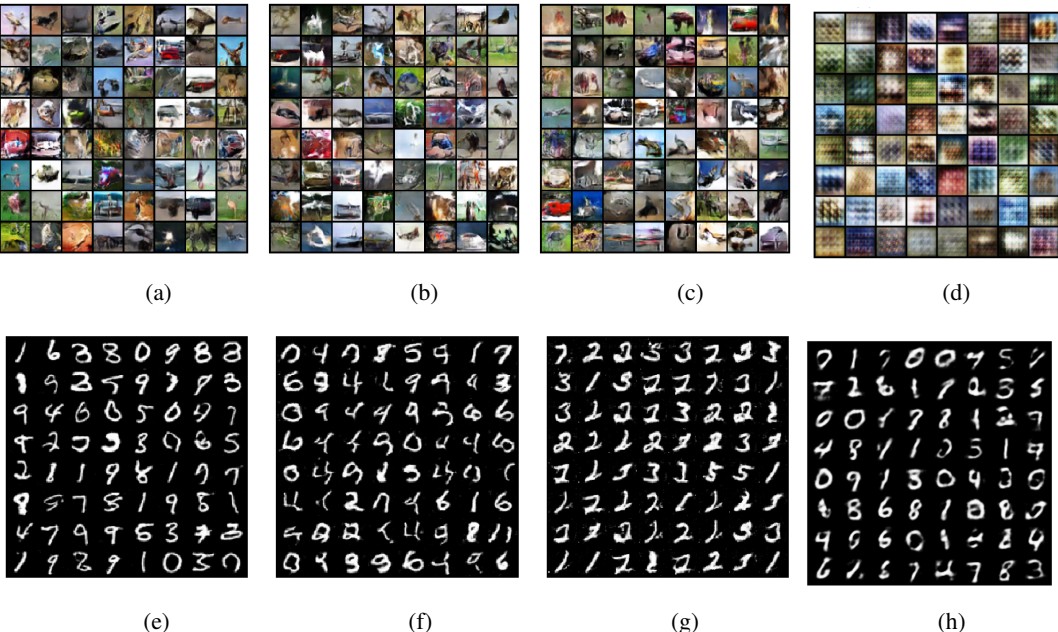

**Figure 1:** Sample generated images using CIFAR-10 (top row), and MNIST (bottom row) data-sets. Panels (a)-(e): Proposed MMD GAN with an *automatic* kernel selection via the particle SGD (Algorithm 1), Panels (b)-(f): MMD GAN Li et al. (2017) with an auto-encoder for dimensionality reduction in conjunction with a mixed RBF Gaussian kernel whose bandwidths are *manually* tuned, Panels (c)-(g): MMD GAN in Li et al. (2017) with a single RBF Gaussian kernel with an auto-encoder for dimensionality reduction in conjunction with a single RBF Gaussian kernel whose bandwidth is *manually* tuned, Panel (d)-(g): GMMN without an auto-encoder Li et al. (2015).

**Quantitivative comparison.** To quantitatively measure the quality and diversity of generated samples, we compute the inception score (IS) Salimans et al. (2016) as well as Frèchet Inception Distance (FID) Heusel et al. (2017) on CIFAR-10 images. Intuitively, the inception score is used for GANs to measure samples quality and diversity. Accordingly, for generative models that are collapsed into a single mode of distribution, the inception score is relatively low. The FID improves on IS by actually comparing the statistics of generated samples to real samples, instead of evaluating generated samples independently.

In Table 1, we report the quantitative measures for different MMD GAN model using different scoring metric. Note that in Table 1 lower FID scores and higher IS scores indicate a better performance. We observe from Table 1 that our approach attain lower FID score, and higher IS score compared to MMD GAN with single Gaussian kernel (bandwidth $\sigma = 16$), and a mixture Gaussian kernel (bandwidths $\{1, 2, 4, 8, 16\}$).

| Method | FID ($\downarrow$) | IS ($\uparrow$) |
|---|---|---|
| MMD GAN (Gaussian) Li et al. (2017) | $67.244 \pm 0.134$ | $5.608 \pm 0.051$ |
| MMD GAN (Mixture Gaussian) Li et al. (2017) | $67.129 \pm 0.148$ | $5.850 \pm 0.055$ |
| SGD Alg. 1 | $\mathbf{65.059 \pm 0.153}$ | $\mathbf{5.97 \pm 0.046}$ |
| Real Data | $0$ | $11.237 \pm 0.116$ |

**Table 1:** Comparison of the quantitative performance measures of MMD GANs with different kernel learning approaches.

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

## APPENDIX

We provide additional material to support the content presented in the paper. This Appendix is organized as follows:

**Appendix A**. We provide the results of our experiments on the synthetic data-set described in Section 5.1 of the main text.

**Appendix B**. We provide the results of our experiments on the LSUN and CelebA data-sets.

**Appendix C**. We present the proofs of the main theoretical results of Section 4 in the main text.

**Appendix D**. We present the proofs of auxiliary lemmas used to support the proof of main results.

**Appendix E**. We prove additional theoretical results regarding the so-called *chaoticity* of the particle SGD in equation 15.

## A  EXPERIMENTAL RESULTS ON THE SYNTHETIC DATA-SET

The synthetic data-set we consider is as follows:

- The distribution of training data is $P_{\boldsymbol{V}} = \mathsf{N}(\boldsymbol{0}, (1 + \lambda)\boldsymbol{I}_{d \times d})$,
- The distribution of generated data is $P_{\boldsymbol{W}} = \mathsf{N}(\boldsymbol{0}, (1 - \lambda)\boldsymbol{I}_{d \times d})$.

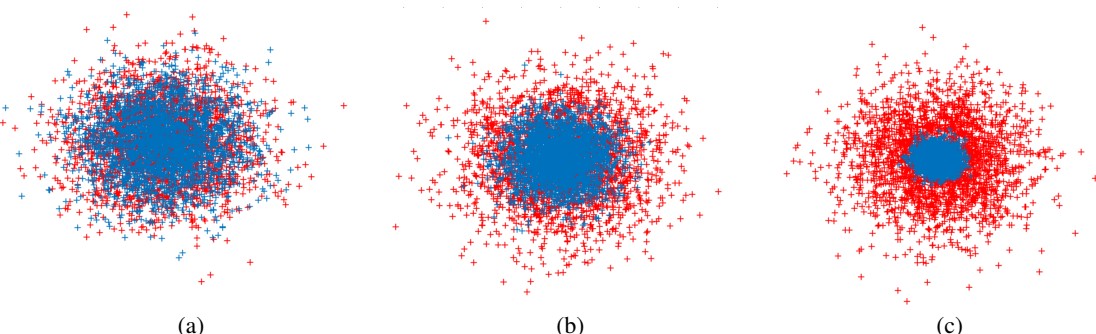

**Figure 2:** Visualization of data-points from the synthetic data-set $P_V = \mathsf{N}(\mathbf{0}, (1+\lambda)\boldsymbol{I}_{d\times d})$ and $P_W = \mathsf{N}(\mathbf{0}, (1-\lambda)\boldsymbol{I}_{d\times d})$ for $d=2$. Panel (a): $\lambda = 0.1$, Panel (b): $\lambda = 0.5$, and Panel (c): $\lambda = 0.9$.

To reduce the dimensionality of data, we consider the embedding $\iota : \mathbb{R}^d \mapsto \mathbb{R}^p, \boldsymbol{x} \mapsto \iota(\boldsymbol{x}) = \boldsymbol{\Sigma}\boldsymbol{x}$, where $\boldsymbol{\Sigma} \in \mathbb{R}^{p\times d}$ and $p < d$. In this case, the distribution of the embedded features are $P_{\boldsymbol{X}|Y=+1} = \mathsf{N}(\mathbf{0}, (1+\lambda)\boldsymbol{\Sigma}\boldsymbol{\Sigma}^T)$, and $P_{\boldsymbol{X}|Y=-1} = \mathsf{N}(\mathbf{0}, (1-\lambda)\boldsymbol{\Sigma}\boldsymbol{\Sigma}^T)$.

Note that $\lambda \in [0,1]$ is a parameter that determines the separation of distributions. In particular, the Kullback-Leibler divergece of the two multi-variate Gaussian distributions is controlled by $\lambda \in [0,1]$,

$$D_{\mathrm{KL}}(P_{\boldsymbol{X}|Y=-1}, P_{\boldsymbol{X}|Y=+1}) = \frac{1}{2}\left[\log\left(\frac{1-\lambda}{1+\lambda}\right) - p + p(1-\lambda^2)\right]. \tag{26}$$

In Figure 2, we show the distributions of *i.i.d.* samples from the distributions $P_V$ and $P_W$ for different choices of variance parameter of $\lambda = 0.1$, $\lambda = 0.5$, and $\lambda = 0.9$. Notice that for larger $\lambda$ the divergence is reduced and thus performing the two-sample test is more difficult. From Figure 2, we clearly observe that for large values of $\lambda$, the data-points from the two distributions $P_V$ and $P_W$ have a large overlap and conducting a statistical test to distinguish between these two distributions is more challenging.

### A.0.1 KERNEL LEARNING APPROACH

Figure 4 depicts our two-phase kernel learning procedure which we also employed in our implementations of Algorithm 1 on benchmark data-sets of Section 5.2 in the main text. The kernel learning approach consists of training the auto-encoder and the kernel optimization sequentially, *i.e.*,

$$\sup_{\widehat{\mu}^N \in \mathcal{P}_N} \sup_{\iota \in \mathcal{Q}} \widehat{\mathrm{MMD}}^{\alpha}_{K_{\widehat{\mu}^N} \circ \iota}[P_V, P_W]. \tag{27}$$

where the function class is defined $\mathcal{Q} \stackrel{\text{def}}{=} \{\iota(\boldsymbol{z}) = \boldsymbol{\Sigma}\boldsymbol{z}, \boldsymbol{\Sigma} \in \mathbb{R}^{D\times d}\}$, and $(K_{\widehat{\mu}^N} \circ \iota)(\boldsymbol{x}_1, \boldsymbol{x}_2) = K_{\widehat{\mu}^N}(\iota(\boldsymbol{x}_1), \iota(\boldsymbol{x}_2))$. Now, we consider a two-phase optimization procedure:

- **Phase (I):** we fix the kernel function, and optimize the auto-encoder to compute a co-variance matrix $\boldsymbol{\Sigma}$ for dimensionality reduction
- **Phase (II):** we optimize the kernel based on the learned embedded features.

This two-phase procedure significantly improves the computational complexity of SGD as it reduces the dimensionality of random feature samples $\boldsymbol{\xi} \in \mathbb{R}^D$, $D \ll d$. When the kernel function $K$ is fixed, optimizing the auto-encoder is equivalent to the kernel learning step of Li et al. (2017).

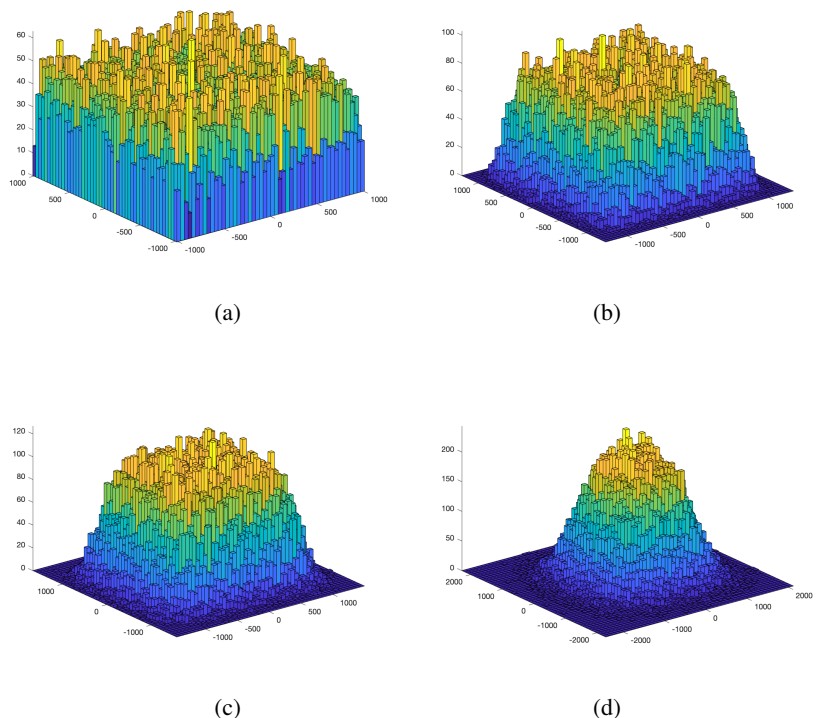

**Figure 3:** The evolution of the empirical measure $\mu_m^N(\boldsymbol{\xi}) = \frac{1}{N}\sum_{k=1}^{N}\delta(\boldsymbol{\xi} - \boldsymbol{\xi}_m^k)$ of the SGD particles $\boldsymbol{\xi}_m^1, \cdots, \boldsymbol{\xi}_m^N \in \mathbb{R}^2$ at different iterations $m$. The empirical measure of random feature maps seemingly converges to a Gaussian stationary measure corresponding to a Gaussian RBF kernel. Panel (a): $m = 0$, Panel (b): $m = 300$, Panel (c): $m = 1000$, and Panel (d): $m = 2500$.

### A.0.2 STATISTICAL HYPOTHESIS TESTING WITH THE KERNEL MMD

Let $\boldsymbol{V}_1, \cdots, \boldsymbol{V}_m \sim_{\text{i.i.d.}} P_{\boldsymbol{V}} = \mathsf{N}(\boldsymbol{0}, (1 + \lambda)\boldsymbol{I}_{d \times d})$, and $\boldsymbol{W}_1, \cdots, \boldsymbol{W}_n \sim_{\text{i.i.d.}} P_{\boldsymbol{W}} = \mathsf{N}(\boldsymbol{0}, (1 - \lambda)\boldsymbol{I}_{d \times d})$. Given these i.i.d. samples, the statistical test $\mathcal{T}(\{\boldsymbol{V}_i\}_{i=1}^m, \{\boldsymbol{W}_i\}_{j=1}^n) : \mathcal{V}^m \times \mathcal{W}^n \to \{0, 1\}$ is used to distinguish between these hypotheses:

- **Null hypothesis** $\mathsf{H}_0 : P_{\boldsymbol{V}} = P_{\boldsymbol{W}}$ (thus $\lambda = 0$),
- **Alternative hypothesis** $\mathsf{H}_1 : P_{\boldsymbol{V}} \neq P_{\boldsymbol{W}}$ (thus $\lambda > 0$).

To perform hypothesis testing via the kernel MMD, we require that $\mathcal{H}_\mathcal{X}$ is a universal RKHS, defined on a compact metric space $\mathcal{X}$. Universality requires that the kernel $K(\cdot, \cdot)$ be continuous and, $\mathcal{H}_\mathcal{X}$ be dense in $C(\mathcal{X})$. Under these conditions, the following theorem establishes that the kernel MMD is indeed a metric:

**Theorem A.1.** (METRIZABLITY OF THE RKHS) *Let $\mathcal{F}$ denotes a unit ball in a universal RKHS $\mathcal{H}_\mathcal{X}$ defined on a compact metric space $\mathcal{X}$ with the associated continuous kernel $K(\cdot, \cdot)$. Then, the kernel MMD is a metric in the sense that $\mathrm{MMD}_K[P_{\boldsymbol{V}}, P_{\boldsymbol{W}}] = 0$ if and only if $P_{\boldsymbol{V}} = P_{\boldsymbol{W}}$.*

To design a test, let $\widehat{\mu}_m^N(\boldsymbol{\xi}) = \frac{1}{N}\sum_{k=1}^{N}\delta(\boldsymbol{\xi} - \boldsymbol{\xi}_m^k)$ denotes the solution of SGD in equation 15 for solving the optimization problem. Consider the following MMD estimator consisting of two

$U$-statistics and an empirical function

$$
\widehat{\text{MMD}}_{K_{\hat{\mu}_m^N \circ \iota}}\big[\{V_i\}_{i=1}^m, \{W_i\}_{i=1}^n\big] = \frac{1}{m(m-1)} \sum_{k=1}^N \sum_{i \neq j} \varphi(\iota(V_i), \xi_m^k) \varphi(\iota(V_j), \xi_m^k)
$$
$$
+ \frac{1}{n(n-1)} \sum_{k=1}^N \sum_{i \neq j} \varphi(\iota(W_i), \xi_m^k) \varphi(\iota(W_j), \xi_m^k)
$$
$$
- \frac{2}{nm} \sum_{k=1}^N \sum_{i=1}^m \sum_{j=1}^n \varphi(\iota(W_i), \xi_m^k) \varphi(\iota(V_j), \xi_m^k). \qquad (28)
$$

Given the samples $\{V_i\}_{i=1}^m$ and $\{W_i\}_{i=1}^n$, we design a test statistic as below

$$
\mathcal{T}(\{V_i\}_{i=1}^m, \{W_i\}_{i=1}^n) \stackrel{\text{def}}{=} \begin{cases} \mathsf{H}_0 & \text{if } \widehat{\text{MMD}}_{K_{\hat{\mu}_m^N \circ \iota}}\big[\{V_i\}_{i=1}^m, \{W_i\}_{i=1}^n\big] \leq \tau \\ \mathsf{H}_1 & \text{if } \widehat{\text{MMD}}_{K_{\hat{\mu}_m^N \circ \iota}}\big[\{V_i\}_{i=1}^m, \{W_i\}_{i=1}^n\big] > \tau, \end{cases}. \qquad (29)
$$

where $\tau \in \mathbb{R}$ is a threshold. Notice that the unbiased MMD estimator of equation 28 can be negative despite the fact that the population MMD is non-negative. Consequently, negative values for the statistical threshold $\tau$ equation 29 are admissible. In the following simulations, we only consider non-negative values for the threshold $\tau$.

A Type I error is made when $\mathsf{H}_0$ is rejected based on the observed samples, despite the null hypothesis having generated the data. Conversely, a Type II error occurs when $\mathsf{H}_0$ is accepted despite the alternative hypothesis $\mathsf{H}_1$ being true. The *significance level* of a test is an upper bound on the probability of a Type I error: this is a design parameter of the test which must be set in advance, and is used to determine the threshold to which we compare the test statistic. The *power of a test* is the probability of rejecting the null hypothesis $\mathsf{H}_0$ when it is indeed incorrect. In particular,

$$
\text{Power} \stackrel{\text{def}}{=} \mathbb{P}(\text{reject } \mathsf{H}_0 | \mathsf{H}_1 \text{ is true}). \qquad (30)
$$

In this sense, the statistical power controls the probability of making Type II errors.

### A.0.3 EMPIRICAL RESULTS

In Figure 3, we show the evolution of the empirical measure $\mu_m^N(\xi)$ of SGD particles by plotting the 2D histogram of the particles $\xi_m^1, \cdots, \xi_m^N \in \mathbb{R}^D$ at different iterations of SGD ($D = d$). Clearly, starting with a uniform distribution in 3(a), the empirical measure seemingly evolves into a Gaussian measure in Figure 3(d). The evolution to a Gaussian distribution demonstrates that the RBF Gaussian kernel corresponding to a Gaussian distribution for the random features indeed provides a good kernel function for the underlying hypothesis test with Gaussian distributions.

In Figure 4, we evaluate the power of the test for 100 trials of hypothesis test using the test statistics of equation 29. To obtain the result, we used an autoencoder to reduce the dimension from $d = 100$ to $p = 50$. Clearly, for the trained kernel in Panel (a) of Figure 4, the threshold $\tau$ for which $\text{Power} = 1$ increases after learning the kernel via the two phase procedure described earlier. In comparison, in Panel (b), we observe that training an auto-encoder only with a fixed standard Gaussian kernel $K(x, y) = e^{-\|x-y\|_2^2}$ attains lower thresholds compared to our two-phase procedure. In Panel (c), we demonstrate the case of a fixed Gaussian kernel without an auto-encoder. In this case, the threshold is significantly lower due to the large dimensionality of the data.

From Figure 4, we also observe that interestingly, the phase transition in the statistical threshold $\tau$ is not sensitive to the parameter $\lambda$. This phenomenon can be justified by the fact that the kernel learning indeed improves the MMD value more significantly for smaller values of $\lambda$ (*i.e.*, more difficult hypothesis testing problems) than larger values of $\lambda$. See Figure 5.

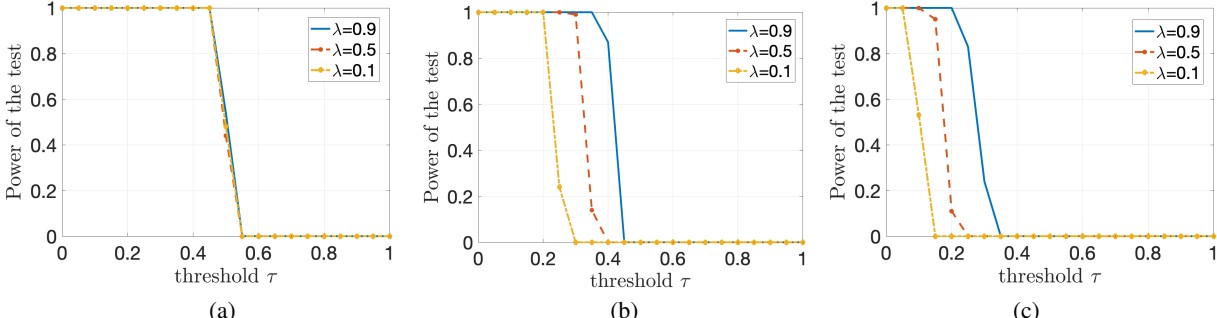

(a)  (b)  (c)

**Figure 4:** The statistical power versus the threshold $\tau$ for the binary hypothesis testing via the unbiased estimator of the kernel MMD. The parameters for this simulations are $\lambda \in \{0.1, 0.5, 0.9\}$, $d = 100$, $n + m = 100$, $p = 50$. Panel (a): Trained kernel using the two-phase procedure with the particle SGD in equation 15 and an auto-encoder, Panel (b): Trained kernel with an auto-encoder and a fixed Gaussian kernel with the bandwidth $\sigma = 1$, Panel (c): Untrained kernel without an auto-encoder.

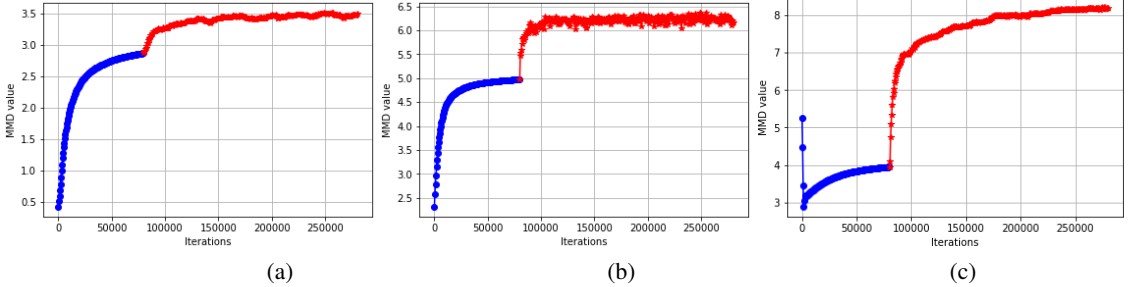

(a)  (b)  (c)

**Figure 5:** The MMD value during the two phase procedure for the kernel training. In the first phase, an auto-encoder is trained (blue curve). In the second phase, the kernel is trained using the embedded features (red curve). Panel (a): $\lambda = 0.9$, Panel (b): $\lambda = 0.5$, Panel (c): $\lambda = 0.1$ .

## B    EXPERIMENTAL RESULTS ON LSUN BEDROOM AND CELEBA DATA-SETS

In this section, we present additional simulations using CelebA Liu et al. (2015), and LSUN bedroom Yu et al. (2015) data-sets. The LSUN dataset of bedroom pictures resized to $64 \times 64$, and the CelebA dataset of celebrity face images resized and cropped to $160 \times 160$. The sample generated images are shown in Figure 6.

The inception score for LSUN bedroom data set is $3.860 \pm 0.0423$ using a single Gaussian kernel with the bandwidth of $\sigma = 16$, and $4.064 \pm 0.061$ using our proposed method in Algorithm 1.

## C    PROOFS OF MAIN THEORETICAL RESULTS

Before we delve into proofs, we state the main assumptions underlying our theoretical results:

**Assumptions**:

(**A**.1) The feature space $\mathcal{X} = \mathcal{V} \cup \mathcal{W} \subset \mathbb{R}^d$ is compact with a finite diameter $\mathrm{diam}(\mathcal{X}) < \infty$, where $\mathcal{V} = \mathrm{support}(P_{\boldsymbol{V}})$ and $\mathcal{W} = \mathrm{support}(P_{\boldsymbol{W}})$ are the supports of the distributions $P_{\boldsymbol{V}}$ and $P_{\boldsymbol{W}}$ respectively.

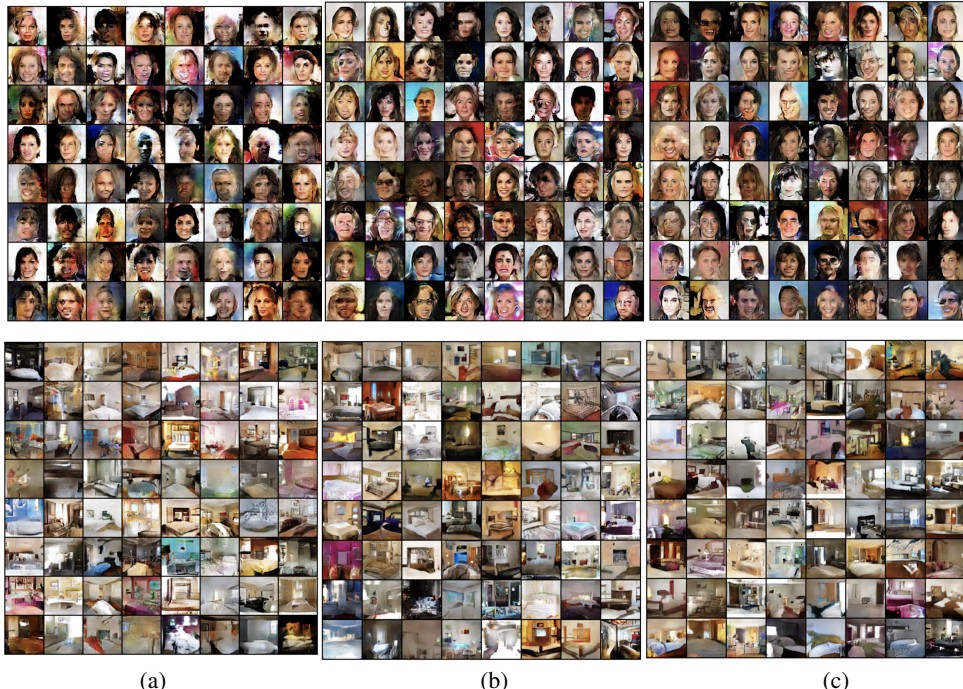

|  (a)  |  (b)  |  (c)  |

**Figure 6:** The sample generated images from CelebA (top row), and LSUN bedroom data-sets (bottom row). Panels (a): Proposed MMD GAN with an *automatic* kernel selection via the particle SGD (Algorithm 1), Panels (b): MMD GAN Li et al. (2017) with an auto-encoder for dimensionality reduction in conjunction with a mixed RBF Gaussian kernel whose bandwidths are *manually* tuned, Panels (c): MMD GAN in Li et al. (2017) with a single RBF Gaussian kernel with an auto-encoder for dimensionality reduction in conjunction with a single RBF Gaussian kernel whose bandwidth is *manually* tuned.

(**A.2**) The feature maps are bounded and Lipchitz almost everywhere (a.e.) $\boldsymbol{\xi} \in \mathbb{R}^D$. In particular, $\sup_{\boldsymbol{x} \in \mathcal{X}} |\varphi(\boldsymbol{x}; \boldsymbol{\xi})| < L_0$, $\sup_{\boldsymbol{x} \in \mathcal{X}} \|\nabla_{\boldsymbol{\xi}} \varphi(\boldsymbol{x}; \boldsymbol{\xi})\|_2 \leq L_1$, and $\sup_{\boldsymbol{\xi} \in \mathbb{R}^D} \|\nabla_{\boldsymbol{x}} \varphi(\boldsymbol{x}; \boldsymbol{\xi})\| < L_2$. Let $L \stackrel{\text{def}}{=} \max\{L_0, L_1, L_2\} < +\infty$.

(**A.3**) Let $\widehat{\mu}_0^N(\boldsymbol{\xi}) \stackrel{\text{def}}{=} \frac{1}{N} \sum_{k=1}^N \delta(\boldsymbol{\xi} - \boldsymbol{\xi}_0^k)$ denotes the empirical measure for the initial particles $\boldsymbol{\xi}_0^1, \cdots, \boldsymbol{\xi}_0^N$. We assume that $\widehat{\mu}_0^N(\boldsymbol{\xi})$ converges (weakly) to a deterministic measure $\mu_0 \in \mathcal{M}(\mathbb{R}^D)$. Furthermore, we assume the limiting measure $\mu_0$ is absolutely continuous *w.r.t.* Lebesgue measure and has a compact support $\text{support}(\mu_0) = \Xi \subset \mathbb{R}^D$.

**Notation**: We denote vectors by lower case bold letters, *e.g.* $\boldsymbol{x} = (x_1, \cdots, x_n) \in \mathbb{R}^n$, and matrices by the upper case bold letters, *e.g.*, $\boldsymbol{M} = [M_{ij}] \in \mathbb{R}^{n \times m}$. The Frobenius norm of a matrix is denoted by $\|\boldsymbol{M}\|_F = \sum_{i=1}^n \sum_{j=1}^m |M_{ij}|^2$. Let $\mathbb{B}_r(\boldsymbol{x}) \stackrel{\text{def}}{=} \{\boldsymbol{y} \in \mathbb{R}^d : \|\boldsymbol{y} - \boldsymbol{x}\|_2 \leq r\}$ denote the Euclidean ball of radius $r$ centered at $\boldsymbol{x}$. For a given metric space $\mathcal{X}$, Let $C_b(\mathbb{R}^d)$ denote the space of bounded and continuous functions on $\mathcal{X}$ equipped with the usual supremum norm

$$\|f\|_\infty \stackrel{\text{def}}{=} \sup_{\boldsymbol{x} \in \mathcal{X}} |f(\boldsymbol{x})|. \tag{31}$$

Further, $C_b^k(\mathcal{X})$ the space of all functions in $C_b(\mathcal{X})$ whose partial derivatives up to order $k$ are bounded and continuous, and $C_c^k(\mathcal{X})$ the space of functions whose partial derivatives up to order $k$ are continuous with compact support.

We denote the class of the integrable functions $f$ with $f(t) \geq 0$ *a.e.*, on $0 \leq t \leq T$ by $L^1_+[0,T]$. Similarly, $L^\infty_+[0,T]$ will denote the essentially bounded functions with $f(t) \geq 0$ almost everywhere. For a given metric space $\mathcal{X}$, we denote the Borel $\sigma$-algebra by $\mathcal{B}(\mathcal{X})$. For a Borel set $B \in \mathcal{B}(\mathcal{X})$, the measure value of the set $B$ with respect to the measure is given by $\mu(B)$. The space of finite non-negative measures defined on $\mathcal{X}$ is denoted by $\mathcal{M}(\mathcal{X})$. The Dirac measure with the unit mass at $x \in \mathcal{X}$ is denoted by $\delta(x)$. For any measure $\mu \in \mathcal{M}(\mathcal{X})$ and any bounded function $f \in C_b(\mathcal{X})$, we define

$$\langle \mu, f \rangle \overset{\text{def}}{=} \int_\mathcal{X} f(x)\mu(\mathrm{d}x). \tag{32}$$

The space $\mathcal{M}(\mathcal{X})$ is equipped with the weak topology, *i.e.*, a (random) sequence $\{\mu^N\}_{N \in \mathbb{N}}$ converges weakly to a deterministic measure $\mu \in \mathcal{M}(\mathcal{X})$ if and only if $\langle \mu^N, f \rangle \to \langle \mu, f \rangle$ for all $f \in C^b(\mathcal{X})$. We denote the weak convergence by $\mu^N_t \overset{\text{weakly}}{\to} \mu$. Notice that when $\mathcal{X}$ is Polish, then $\mathcal{M}(\mathcal{X})$ equipped with the weak topology is also Polish.[3] For a Polish space $\mathcal{X}$, let $\mathcal{D}_\mathcal{X}([0,T])$ denotes the Skorokhod space of the cádlág functions that take values in $\mathcal{X}$ defined on $[0,T]$. We assume that $\mathcal{D}_\mathcal{X}([0,T])$ is equipped with the Skorokhod's $J_1$-topology Billingsley (2013), which in that case $\mathcal{D}_\mathcal{X}([0,T])$ is also a Polish space.

We use asymptotic notations throughout the paper. We use the standard asymptotic notation for sequences. If $a_n$ and $b_n$ are positive sequences, then $a_n = \mathcal{O}(b_n)$ means that $\limsup_{n\to\infty} a_n/b_n < \infty$, whereas $a_n = \Omega(b_n)$ means that $\liminf_{n\to\infty} a_n/b_n > 0$. Furthermore, $a_n = \widetilde{\mathcal{O}}(b_n)$ implies $a_n = \mathcal{O}(b_n \text{poly} \log(b_n))$. Moreover $a_n = o(b_n)$ means that $\lim_{n\to\infty} a_n/b_n = 0$ and $a_n = \omega(b_n)$ means that $\lim_{n\to\infty} a_n/b_n = \infty$. Lastly, we have $a_n = \Theta(b_n)$ if $a_n = \mathcal{O}(b_n)$ and $a_n = \Omega(b_n)$. Finally, for positive $a, b > 0$, denote $a \lesssim b$ if $a/b$ is at most some universal constant.

**Definition C.1.** (ORLICZ NORM) The Young-Orlicz modulus is a convex non-decreasing function $\psi : \mathbb{R}_+ \to \mathbb{R}_+$ such that $\psi(0) = 0$ and $\psi(x) \to \infty$ when $x \to \infty$. Accordingly, the Orlicz norm of an integrable random variable $X$ with respect to the modulus $\psi$ is defined as

$$\|X\|_\psi \overset{\text{def}}{=} \inf\{\beta > 0 : \mathbb{E}[\psi(\|X\| - \mathbb{E}[\|X\|]\|/\beta)] \leq 1\}. \tag{33}$$

In the sequel, we consider the Orlicz modulus $\psi_\nu(x) \overset{\text{def}}{=} (x^\nu) - 1$ . Accordingly, the cases of $\|\cdot\|_{\psi_2}$ and $\|\cdot\|_{\psi_1}$ norms are called the sub-Gaussian and the sub-exponential norms and have the following alternative definitions:

**Definition C.2.** (SUB-GAUSSIAN NORM) The sub-Gaussian norm of a random variable $Z$, denoted by $\|Z\|_{\psi_2}$, is defined as

$$\|Z\|_{\psi_2} = \sup_{q \geq 1} q^{-1/2}(\mathbb{E}|Z|^q)^{1/q}. \tag{34}$$

For a random vector $\boldsymbol{Z} \in \mathbb{R}^n$, its sub-Gaussian norm is defined as follows

$$\|\boldsymbol{Z}\|_{\psi_2} = \sup_{\boldsymbol{x} \in \mathrm{S}^{n-1}} \|\langle \boldsymbol{x}, \boldsymbol{Z} \rangle\|_{\psi_2}. \tag{35}$$

**Definition C.3.** (SUB-EXPONENTIAL NORM) The sub-exponential norm of a random variable $Z$, denoted by $\|Z\|_{\psi_1}$, is defined as follows

$$\|Z\|_{\psi_1} = \sup_{q \geq 1} q^{-1}(\mathbb{E}[|Z|^q])^{1/q}. \tag{36}$$

For a random vector $\boldsymbol{Z} \in \mathbb{R}^n$, its sub-exponential norm is defined below

$$\|\boldsymbol{Z}\|_{\psi_1} = \sup_{\boldsymbol{x} \in \mathrm{S}^{n-1}} \|\langle \boldsymbol{Z}, \boldsymbol{x} \rangle\|_{\psi_1}. \tag{37}$$

---

[3]A topological space is Polish if it is homeomorphic to a complete, separable metric space.

## C.1 PROOF OF THEOREM 4.1

By the triangle inequality, we have that

$$\left| \mathrm{MMD}_{\mu_*}[P_{\boldsymbol{V}}, P_{\boldsymbol{W}_*}] - \mathrm{MMD}_{\widehat{\mu}_*^N}\left[P_{\boldsymbol{V}}, P_{\widehat{\boldsymbol{W}}_*^N}\right]\right| \leq \mathsf{A}_1 + \mathsf{A}_2 + \mathsf{A}_3 + \mathsf{A}_4, \tag{38}$$

where the terms $\mathsf{A}_i, i = 1, 2, 3, 4$ are defined as follows

$$\mathsf{A}_1 \stackrel{\text{def}}{=} \left| \mathrm{MMD}_{\mu_*}[P_{\boldsymbol{V}}, P_{\boldsymbol{W}_*}] - \min_{\boldsymbol{W} \in \mathcal{W}} \sup_{\mu \in \mathcal{P}} \widehat{\mathrm{MMD}}_\mu[P_{\boldsymbol{V}}, P_{\boldsymbol{W}}]\right|$$

$$\mathsf{A}_2 \stackrel{\text{def}}{=} \left| \min_{\boldsymbol{W} \in \mathcal{W}} \sup_{\mu \in \mathcal{P}} \widehat{\mathrm{MMD}}_\mu[P_{\boldsymbol{V}}, P_{\boldsymbol{W}}] - \min_{\boldsymbol{W} \in \mathcal{W}} \sup_{\widehat{\mu}^N \in \mathcal{P}_N} \widehat{\mathrm{MMD}}_{\widehat{\mu}^N}\left[P_{\boldsymbol{V}}, P_{\boldsymbol{W}}\right]\right|$$

$$\mathsf{A}_3 \stackrel{\text{def}}{=} \left| \min_{\boldsymbol{W} \in \mathcal{W}} \sup_{\widehat{\mu}^N \in \mathcal{P}_N} \widehat{\mathrm{MMD}}_{\widehat{\mu}^N}\left[P_{\boldsymbol{V}}, P_{\boldsymbol{W}}\right] - \widehat{\mathrm{MMD}}_{\widehat{\mu}_*^N}\left[P_{\boldsymbol{V}}, P_{\boldsymbol{W}}\right]\right|$$

$$\mathsf{A}_4 \stackrel{\text{def}}{=} \left| \widehat{\mathrm{MMD}}_{\widehat{\mu}_*^N}\left[P_{\boldsymbol{V}}, P_{\widehat{\boldsymbol{W}}_*^N}\right] - \mathrm{MMD}_{\widehat{\mu}_*^N}\left[P_{\boldsymbol{V}}, P_{\widehat{\boldsymbol{W}}_*^N}\right]\right|.$$

In the sequel, we compute an upper bound for each term on the right hand side of equation 38:

**Upper bound on $\mathsf{A}_1$:**

First, notice that the squared kernel MMD loss in equation 4 can be characterized in terms of class labels and features defined in Section 3.1 as follows

$$\mathrm{MMD}_\mu[P_{\boldsymbol{V}}, P_{\boldsymbol{W}}] = 4\mathbb{E}_{P_{\boldsymbol{x}, y}^{\otimes 2}}\left[y\widehat{y}K_\mu(\boldsymbol{x}, \widehat{\boldsymbol{x}})\right]. \tag{39}$$

To see this equivalence, we first rewrite the right hand side of equation 39 as follows

$$\mathbb{E}_{P_{y, \boldsymbol{x}}^{\otimes 2}}\left[y\widehat{y}K_\mu(\boldsymbol{x}, \widehat{\boldsymbol{x}})\right] = \mathbb{P}\{y = +1\}\mathbb{P}\{\widehat{y} = +1\}\mathbb{E}_{\boldsymbol{x}, \widehat{\boldsymbol{x}} \sim P_{\boldsymbol{x}|y=+1}^{\otimes 2}}\left[K_\mu(\boldsymbol{x}, \widehat{\boldsymbol{x}})\right]$$

$$+ \mathbb{P}\{y = -1\}\mathbb{P}\{\widehat{y} = -1\}\mathbb{E}_{\boldsymbol{x}, \widehat{\boldsymbol{x}} \sim P_{\boldsymbol{x}|y=-1}^{\otimes 2}}\left[K_\mu(\boldsymbol{x}, \widehat{\boldsymbol{x}})\right]$$

$$- \mathbb{P}\{y = -1\}\mathbb{P}\{\widehat{y} = +1\}\mathbb{E}_{\boldsymbol{x} \sim P_{\boldsymbol{x}|y=-1}, \widehat{\boldsymbol{x}} \sim P_{\boldsymbol{x}|y=+1}}\left[K_\mu(\boldsymbol{x}, \widehat{\boldsymbol{x}})\right]$$

$$- \mathbb{P}\{y = +1\}\mathbb{P}\{\widehat{y} = -1\}\mathbb{E}_{\boldsymbol{x} \sim P_{\boldsymbol{x}|y=+1}, \widehat{\boldsymbol{x}} \sim P_{\boldsymbol{x}|y=-1}}\left[K_\mu(\boldsymbol{x}, \widehat{\boldsymbol{x}})\right]. \tag{40}$$

Now, recall from Section 3.1 that $P_{\boldsymbol{x}|y=+1} = P_{\boldsymbol{V}}$, and $P_{\boldsymbol{x}|y=-1} = P_{\boldsymbol{W}}$ by construction of the labels and random features. Moreover, $y, \widehat{y} \sim_{\text{i.i.d.}}$ Uniform$\{-1, +1\}$, and thus $\mathbb{P}\{y = -1\} = \mathbb{P}\{y = +1\} = \frac{1}{2}$. Therefore, from equation 40, we derive

$$\mathbb{E}_{P_{y, \boldsymbol{x}}^{\otimes 2}}\left[y\widehat{y}K_\mu(\boldsymbol{x}, \widehat{\boldsymbol{x}})\right] = \frac{1}{4}\mathbb{E}_{P_{\boldsymbol{V}}^{\otimes 2}}[K_\mu(\boldsymbol{x}; \widehat{\boldsymbol{x}})] + \frac{1}{4}\mathbb{E}_{P_{\boldsymbol{W}}^{\otimes 2}}[K_\mu(\boldsymbol{x}; \widehat{\boldsymbol{x}})] - \frac{1}{2}\mathbb{E}_{P_{\boldsymbol{V}}, P_{\boldsymbol{W}}}[K_\mu(\boldsymbol{x}; \widehat{\boldsymbol{x}})]$$

$$= \frac{1}{4}\mathrm{MMD}_\mu[P_{\boldsymbol{V}}, P_{\boldsymbol{W}}].$$

For any given $\boldsymbol{W} \in \mathcal{W}$, we have that

$$\left| \sup_{\mu \in \mathcal{P}} \widehat{\mathrm{MMD}}_\mu[P_{\boldsymbol{V}}, P_{\boldsymbol{W}}] - \sup_{\mu \in \mathcal{P}} \mathrm{MMD}_\mu[P_{\boldsymbol{V}}, P_{\boldsymbol{W}}]\right|$$

$$\leq \sup_{\mu \in \mathcal{P}} \left| \widehat{\mathrm{MMD}}_\mu[P_{\boldsymbol{V}}, P_{\boldsymbol{W}}] - \mathrm{MMD}_\mu[P_{\boldsymbol{V}}, P_{\boldsymbol{W}}]\right|$$

$$= 4\sup_{\mu \in \mathcal{P}} \left| \frac{1}{n(n-1)}\sum_{i \neq j} y_i y_j K_\mu(\boldsymbol{x}_i, \boldsymbol{x}_j) - \mathbb{E}_{P_{y, \boldsymbol{x}}^{\otimes 2}}\left[y\widehat{y}K_\mu(\boldsymbol{x}, \widehat{\boldsymbol{x}})\right]\right|$$

$$= 4\sup_{\mu \in \mathcal{P}} \left| \mathbb{E}_\mu[E_n(\boldsymbol{\xi})]\right|$$

$$\leq 4\left| \sup_{\mu \in \mathcal{P}} \mathbb{E}_\mu[E_n(\boldsymbol{\xi})]\right|,$$

where the error term is defined using the random features

$$E_n(\boldsymbol{\xi}) \stackrel{\text{def}}{=} \frac{1}{n(n-1)} \sum_{i \neq j} y_i y_j \varphi(\boldsymbol{x}_i; \boldsymbol{\xi}) \varphi(\boldsymbol{x}_j; \boldsymbol{\xi}) - \mathbb{E}_{P_{\boldsymbol{x},y}^{\otimes 2}} [y\widehat{y}\varphi(\boldsymbol{x}; \boldsymbol{\xi})\varphi(\widehat{\boldsymbol{x}}, \boldsymbol{\xi})]. \tag{41}$$

Now, we invoke the following strong duality theorem Gao & Kleywegt (2016):

**Theorem C.4.** (STRONG DUALITY FOR ROBUST OPTIMIZATION, (GAO & KLEYWEGT, 2016, THEOREM 1)) *Consider the general metric space* $(\Xi, d)$, *and any normal distribution* $\nu \in \mathcal{M}(\Xi)$, *where* $\mathcal{M}(\Xi)$ *is the set of Borel probability measures on* $\Xi$. *Then,*

$$\sup_{\mu \in \mathcal{M}(\Xi)} \left\{ \mathbb{E}_\mu[\Psi(\boldsymbol{\xi})] : W_p(\mu, \nu) \leq R \right\} = \min_{\lambda \geq 0} \left\{ \lambda R^p - \int_\Xi \inf_{\boldsymbol{\xi} \in \Xi} [\lambda d^p(\boldsymbol{\xi}, \boldsymbol{\zeta}) - \Psi(\boldsymbol{\xi})] \nu(\mathrm{d}\boldsymbol{\zeta}) \right\}, \tag{42}$$

*provided that* $\Psi$ *is upper semi-continuous in* $\boldsymbol{\xi}$.

Under the strong duality of Theorem C.4, we obtain that

$$\left| \sup_{\mu \in \mathcal{P}} \widehat{\text{MMD}}_\mu[P_{\boldsymbol{V}}, P_{\boldsymbol{W}}] - \sup_{\mu \in \mathcal{P}} \text{MMD}_\mu[P_{\boldsymbol{V}}, P_{\boldsymbol{W}}] \right|$$

$$\leq 4 \left| \min_{\lambda \geq 0} \left\{ \lambda R^p - \int_{\mathbb{R}^D} \inf_{\boldsymbol{\zeta} \in \mathbb{R}^D} \left[ \lambda \|\boldsymbol{\xi} - \boldsymbol{\zeta}\|_2^p - E_n(\boldsymbol{\zeta}) \right] \mu_0(\mathrm{d}\boldsymbol{\xi}) \right\} \right|. \tag{43}$$

In the sequel, let $p = 2$. The *Moreau's envelope* Parikh & Boyd (2014) of a function $f : \mathcal{X} \to \mathbb{R}$ is defined as follows

$$M_f^\beta(\boldsymbol{y}) \stackrel{\text{def}}{=} \inf_{\boldsymbol{x} \in \mathcal{X}} \left\{ \frac{1}{2\beta} \|\boldsymbol{x} - \boldsymbol{y}\|_2^2 + f(\boldsymbol{x}) \right\}, \quad \forall \boldsymbol{y} \in \mathcal{X}, \tag{44}$$

where $\beta > 0$ is the regularization parameter. When the function $f$ is differentiable, the following lemma can be established:

**Lemma C.5.** (MOREAU'S ENVELOPE OF DIFFERENTIABLE FUNCTIONS) *Suppose the function* $f : \mathcal{X} \to \mathbb{R}$ *is differentiable. Then, the Moreau's envelope defined in equation 44 has the following upper bound and lower bounds*

$$f(\boldsymbol{y}) - \frac{\beta}{2} \int_0^1 \sup_{\boldsymbol{x} \in \mathcal{X}} \|\nabla f(\boldsymbol{y} + s(\boldsymbol{x} - \boldsymbol{y}))\|_2^2 \mathrm{d}s \leq M_f^\beta(\boldsymbol{y}) \leq f(\boldsymbol{y}). \tag{45}$$

*In particular, when* $f$ *is* $L_f$-*Lipschitz, we have*

$$f(\boldsymbol{y}) - \frac{\beta L_f^2}{2} \leq M_f^\beta(\boldsymbol{y}) \leq f(\boldsymbol{y}). \tag{46}$$

The proof is presented in Appendix D.1.

Now, we return to Equation equation 43. We leverage the lower bound on Moreau's envelope in equation 45 of Lemma C.5 as follows

$$\left| \sup_{\mu \in \mathcal{P}} \widehat{\text{MMD}}_\mu[P_{\boldsymbol{V}}, P_{\boldsymbol{W}}] - \sup_{\mu \in \mathcal{P}} \text{MMD}_\mu[P_{\boldsymbol{V}}, P_{\boldsymbol{W}}] \right|$$

$$\leq 4 \left| \min_{\lambda \geq 0} \left\{ \lambda R^2 - \int_{\mathbb{R}^D} M_{-E_n}^{\frac{1}{2\lambda}}(\boldsymbol{\xi}) \mu_0(\mathrm{d}\boldsymbol{\xi}) \right\} \right|$$

$$\leq 4 \left| \min_{\lambda \geq 0} \left\{ \lambda R^2 + \mathbb{E}_{\mu_0}[E_n(\boldsymbol{\xi})] + \frac{1}{4\lambda} \mathbb{E}_{\mu_0} \left[ \int_0^1 \sup_{\boldsymbol{\zeta} \in \mathbb{R}^D} \|\nabla E_n((1-s)\boldsymbol{\xi} + s\boldsymbol{\zeta})\|_2^2 \mathrm{d}s \right] \right\} \right|$$

$$\leq 4|\mathbb{E}_{\mu_0}[E_n(\boldsymbol{\xi})]| + 4R\mathbb{E}_{\mu_0} \left[ \int_0^1 \sup_{\boldsymbol{\zeta} \in \mathbb{R}^D} \|\nabla E_n((1-s)\boldsymbol{\xi} + s\boldsymbol{\zeta})\|_2^2 \mathrm{d}s \right]. \tag{47}$$

Let $\boldsymbol{\zeta}_* = \boldsymbol{\zeta}_*(\boldsymbol{\xi}, s) = \arg \sup_{\boldsymbol{\zeta} \in \mathbb{R}^D} \|\nabla E_n (1-s)\boldsymbol{\xi} + s\boldsymbol{\zeta}\|_2$. Then, applying the union bound in conjunction with Inequality equation 47 yields

$$
\mathbb{P} \left( \left| \sup_{\mu \in \mathcal{P}} \widehat{\text{MMD}}_\mu[P_{\boldsymbol{V}}, P_{\boldsymbol{W}}] - \sup_{\mu \in \mathcal{P}} \text{MMD}_\mu[P_{\boldsymbol{V}}, P_{\boldsymbol{W}}] \right| \geq \delta \right)
$$

$$
\leq \mathbb{P} \left( \left| \int_{\mathbb{R}^D} E_n(\boldsymbol{\xi}) \mu_0(\text{d}\boldsymbol{\xi}) \right| \geq \frac{\delta}{8} \right) + \mathbb{P} \left( \int_{\mathbb{R}^D} \int_0^1 \|\nabla E_n((1-s)\boldsymbol{\xi} + s\boldsymbol{\zeta}_*)\|_2^2 \text{d}s \mu_0(\text{d}\boldsymbol{\xi}) \geq \frac{\delta}{8R} \right).
$$

$$(48)$$

Now, we state the following lemma:

**Lemma C.6.** (TAIL BOUNDS FOR THE FINITE SAMPLE ESTIMATION ERROR) *Consider the estimation error $E_n$ defined in equation 41. Then, the following statements hold:*

- *$Z = \|\nabla E_n(\boldsymbol{\xi})\|_2^2$ is a sub-exponential random variable with the Orlicz norm of $\|Z\|_{\psi_1} \leq \frac{9 \times 2^9 \times L^4}{n^2}$ for every $\boldsymbol{\xi} \in \mathbb{R}^D$. Moreover,*

$$
\mathbb{P} \left( \int_{\mathbb{R}^D} \int_0^1 \|\nabla E_n((1-s)\boldsymbol{\xi} + s\boldsymbol{\zeta}_*)\|_2^2 \text{d}s \mu_0(\text{d}\boldsymbol{\xi}) \geq \delta \right) \leq 2e^{-\frac{n^2\delta}{9 \times 2^9 \times L^4} + \frac{L^4}{9}},
$$

$$(49)$$

- *$E_n(\boldsymbol{\xi})$ is zero-mean sub-Gaussian random variable with the Orlicz norm of $\|E_n(\boldsymbol{\xi})\|_{\psi_2} \leq \frac{16\sqrt{3}L^4}{n}$ for every $\boldsymbol{\xi} \in \mathbb{R}^D$. Moreover,*

$$
\mathbb{P} \left( \left| \int_{\mathbb{R}^D} E_n(\boldsymbol{\xi}) \mu_0(\text{d}\boldsymbol{\xi}) \right| \geq \delta \right) \geq 2e^{-\frac{n^2\delta^2}{16\sqrt{3}L^4}}.
$$

$$(50)$$

The proof of Lemma C.6 is presented in Appendix C.2.

Now, we leverage the probability bounds equation 49 and equation 50 of Lemma C.6 to upper bound the terms on the right hand side of equation 48 as below

$$
\mathbb{P} \left( \left| \sup_{\mu \in \mathcal{P}} \widehat{\text{MMD}}_\mu[P_{\boldsymbol{V}}, P_{\boldsymbol{W}}] - \sup_{\mu \in \mathcal{P}} \text{MMD}_\mu[P_{\boldsymbol{V}}, P_{\boldsymbol{W}}] \right| \geq \delta \right) \leq 2e^{-\frac{n^2\delta^2}{\sqrt{3} \times 2^{11} \times L^4}} + 2e^{-\frac{n^2\delta}{9 \times 2^{12} \times RL^4} + \frac{L^4}{9}}
$$

$$
\leq 4 \max \left\{ e^{-\frac{n^2\delta^2}{\sqrt{3} \times 2^{11} \times L^4}}, e^{-\frac{n^2\delta}{9 \times 2^{12} \times RL^4} + \frac{L^4}{9}} \right\},
$$

$$(51)$$

where the last inequality comes from the basic inequality $a + b \leq 2 \max\{a, b\}$. Therefore, with the probability of at least $1 - \varrho$, we have that

$$
\left| \sup_{\mu \in \mathcal{P}} \widehat{\text{MMD}}_\mu[P_{\boldsymbol{V}}, P_{\boldsymbol{W}}] - \sup_{\mu \in \mathcal{P}} \text{MMD}_\mu[P_{\boldsymbol{V}}, P_{\boldsymbol{W}}] \right|
$$

$$
\leq \max \left\{ \frac{3^{\frac{1}{4}} \times 2^{\frac{11}{2}} \times L^2}{n} \ln^{\frac{1}{2}} \left( \frac{4}{\varrho} \right), \frac{9 \times 2^{12} \times RL^4}{n^2} \ln \left( \frac{4e^{\frac{L^4}{9}}}{\varrho} \right) \right\},
$$

$$(52)$$

for all $\boldsymbol{W} \in \mathcal{W}$.

**Lemma C.7.** (DISTANCE BETWEEN MINIMA OF ADJACENT FUNCTIONS) *Let $\Psi(\boldsymbol{W}) : \mathcal{W} \to \mathbb{R}$ and $\Phi(\boldsymbol{W}) : \mathcal{W} \to \mathbb{R}$. Further, suppose $\|\Psi(\boldsymbol{W}) - \Phi(\boldsymbol{W})\|_\infty \leq \delta$ for some $\delta > 0$. Then,*

$$
\left| \min_{\boldsymbol{W} \in \mathcal{W}} \Psi(\boldsymbol{W}) - \min_{\boldsymbol{W} \in \mathcal{W}} \Phi(\boldsymbol{W}) \right| \leq \delta.
$$

$$(53)$$

See Appendix D.4 for the proof.

Let $\Psi(\boldsymbol{W}) \stackrel{\text{def}}{=} \sup_{\mu \in \mathcal{P}} \widehat{\text{MMD}}_\mu[P_{\boldsymbol{V}}, P_{\boldsymbol{W}}]$, and $\Phi(\boldsymbol{W}) \stackrel{\text{def}}{=} \sup_{\mu \in \mathcal{P}} \text{MMD}_\mu[P_{\boldsymbol{V}}, P_{\boldsymbol{W}}]$. Then, from Inequality equation 53, we have the following upper bound on $\mathsf{A}_1$

$$
\begin{aligned}
\mathsf{A}_1 &= \left| \text{MMD}_{\mu_*}[P_{\boldsymbol{V}}, P_{\boldsymbol{W}_*}] - \min_{\boldsymbol{W} \in \mathcal{W}} \sup_{\mu \in \mathcal{P}} \widehat{\text{MMD}}_\mu[P_{\boldsymbol{V}}, P_{\boldsymbol{W}}] \right| \\
&= \left| \min_{\boldsymbol{W} \in \mathcal{W}} \sup_{\mu \in \mathcal{P}} \text{MMD}_\mu[P_{\boldsymbol{V}}, P_{\boldsymbol{W}}] - \min_{\boldsymbol{W} \in \mathcal{W}} \sup_{\mu \in \mathcal{P}} \widehat{\text{MMD}}_\mu[P_{\boldsymbol{V}}, P_{\boldsymbol{W}}] \right| \\
&\leq \max \left\{ \frac{3^{\frac{1}{4}} \times 2^4 \times L^2}{n} \ln^{\frac{1}{2}}\left(\frac{4}{\varrho}\right), \frac{9 \times 2^{11} \times RL^4}{n^2} \ln\left(\frac{4e^{\frac{L^4}{9}}}{\varrho}\right) \right\}.
\end{aligned} \tag{54}
$$

with the probability of (at least) $1 - \varrho$.

**Upper bound on $\mathsf{A}_2$:**

To establish the upper bound on $\mathsf{A}_2$, we recall that

$$
\widehat{\text{MMD}}_{\widehat{\mu}^N}[P_{\boldsymbol{V}}, P_{\boldsymbol{W}}] = \frac{1}{n(n-1)} \frac{1}{N} \sum_{i \neq j} \sum_{k=1}^N y_i y_j \varphi(\boldsymbol{x}_i; \boldsymbol{\xi}^k) \varphi(\boldsymbol{x}_j; \boldsymbol{\xi}^k) \tag{55a}
$$

$$
\widehat{\text{MMD}}_\mu[P_{\boldsymbol{V}}, P_{\boldsymbol{W}}] = \frac{1}{n(n-1)} \sum_{i \neq j} y_i y_j \mathbb{E}_\mu[\varphi(\boldsymbol{x}_i; \boldsymbol{\xi}) \varphi(\boldsymbol{x}_j; \boldsymbol{\xi})]. \tag{55b}
$$

Therefore,

$$
\begin{aligned}
&\left| \sup_{\mu \in \mathcal{P}} \widehat{\text{MMD}}_\mu[P_{\boldsymbol{V}}, P_{\boldsymbol{W}}] - \sup_{\widehat{\mu}^N \in \mathcal{P}_N} \widehat{\text{MMD}}_{\widehat{\mu}^N}[P_{\boldsymbol{V}}, P_{\boldsymbol{W}}] \right| \\
&\qquad\qquad \leq \left| \sup_{\mu \in \mathcal{P}} \mathbb{E}_\mu[\varphi(\boldsymbol{x}_i; \boldsymbol{\xi}) \varphi(\boldsymbol{x}_j; \boldsymbol{\xi})] - \sup_{\widehat{\mu}^N \in \mathcal{P}_N} \mathbb{E}_{\widehat{\mu}^N}[\varphi(\boldsymbol{x}_i; \boldsymbol{\xi}) \varphi(\boldsymbol{x}_j; \boldsymbol{\xi})] \right|. \tag{56}
\end{aligned}
$$

Here, the last inequality is due to Theorem C.4 and the following duality results hold

$$
\sup_{\mu \in \mathcal{P}} \mathbb{E}_\mu[\varphi(\boldsymbol{x}; \boldsymbol{\xi}) \varphi(\widehat{\boldsymbol{x}}; \boldsymbol{\xi})] = \inf_{\lambda \geq 0} \left\{ \lambda R^2 - \int_{\mathbb{R}^D} \inf_{\boldsymbol{\zeta} \in \mathbb{R}^D} \{\lambda \|\boldsymbol{\xi} - \boldsymbol{\zeta}\|_2^2 - \varphi(\boldsymbol{x}_i; \boldsymbol{\zeta}) \varphi(\boldsymbol{x}_j; \boldsymbol{\zeta})\} \mu_0(\mathrm{d}\boldsymbol{\xi}) \right\}
$$

$$
\sup_{\widehat{\mu}^N \in \mathcal{P}_N} \mathbb{E}_{\widehat{\mu}^N}[\varphi(\boldsymbol{x}_i; \boldsymbol{\xi}) \varphi(\boldsymbol{x}_j; \boldsymbol{\xi})] = \inf_{\lambda \geq 0} \left\{ \lambda R^2 - \frac{1}{N} \sum_{k=1}^N \inf_{\boldsymbol{\zeta} \in \mathbb{R}^D} \{\lambda \|\boldsymbol{\xi}_0^k - \boldsymbol{\zeta}\|_2^2 - \varphi(\boldsymbol{x}_i; \boldsymbol{\zeta}) \varphi(\boldsymbol{x}_j; \boldsymbol{\zeta})\} \right\}.
$$

Now, in the sequel, we establish a uniform concentration result for the following function

$$
T_{(\boldsymbol{x}, \widehat{\boldsymbol{x}})}^\lambda : \mathbb{R}^{N \times D} \mapsto \mathbb{R}
$$

$$
(\boldsymbol{\xi}_0^1, \cdots, \boldsymbol{\xi}_0^N) \mapsto T_{(\boldsymbol{x}, \widehat{\boldsymbol{x}})}^\lambda(\boldsymbol{\xi}_0^1, \cdots, \boldsymbol{\xi}_0^N) = \frac{1}{N} \sum_{k=1}^N M_{-\varphi(\boldsymbol{x}, \cdot)\varphi(\widehat{\boldsymbol{x}}, \cdot)}^{\frac{1}{2\lambda}}(\boldsymbol{\xi}_0^k) - \int_{\mathbb{R}^N} M_{-\varphi(\boldsymbol{x}, \cdot)\varphi(\widehat{\boldsymbol{x}}, \cdot)}^{\frac{1}{2\lambda}}(\boldsymbol{\xi}) \mu_0(\mathrm{d}\boldsymbol{\xi}).
$$

Then, from equation 56 we have

$$
\left| \sup_{\mu \in \mathcal{P}} \widehat{\text{MMD}}_\mu[P_{\boldsymbol{V}}, P_{\boldsymbol{W}}] - \sup_{\widehat{\mu}^N \in \mathcal{P}_N} \widehat{\text{MMD}}_{\widehat{\mu}^N}[P_{\boldsymbol{V}}, P_{\boldsymbol{W}}] \right| \leq \sup_{\lambda \geq 0} \sup_{\boldsymbol{x}, \widehat{\boldsymbol{x}} \in \mathcal{X}} |T_{(\boldsymbol{x}, \widehat{\boldsymbol{x}})}^\lambda(\boldsymbol{\xi}_0^1, \cdots, \boldsymbol{\xi}_0^N)|. \tag{57}
$$

We now closely follow the argument of Rahimi & Recht (2008) to establish a uniform concentration result with respect to the data points $\boldsymbol{x}, \widehat{\boldsymbol{x}} \in \mathcal{X}$. In particular, consider an $\epsilon$-net cover of $\mathcal{X} \subset \mathbb{R}^d$.

Then, we require $N_\epsilon = \left(\frac{4\mathrm{diam}(\mathcal{X})}{\epsilon}\right)^d$ balls of the radius $\epsilon > 0$, *e.g.*, see (Pollard, 1990, Lemma 4.1, Section 4). Let $\mathcal{Z} = \{\boldsymbol{z}_1, \cdots, \boldsymbol{z}_{N_\epsilon}\} \subset \mathcal{X}$ denotes the center of the covering net. Now, let $(\boldsymbol{\xi}_0^1, \cdots, \boldsymbol{\xi}_0^k, \cdots, \boldsymbol{\xi}_0^N) \in \mathrm{I\!R}^{N \times D}$ and $(\boldsymbol{\xi}_0^1, \cdots, \widetilde{\boldsymbol{\xi}}_0^k, \cdots, \boldsymbol{\xi}_0^N) \in \mathrm{I\!R}^{N \times D}$ be two sequences that differs in the $k$-th coordinate for $1 \le k \le N$. Then,

$$\left| T_{(\boldsymbol{z}_i, \boldsymbol{z}_j)}(\boldsymbol{\xi}_0^1, \cdots, \boldsymbol{\xi}_0^k, \cdots, \boldsymbol{\xi}_0^N) - T_{(\boldsymbol{z}_i, \boldsymbol{z}_j)}(\boldsymbol{\xi}_0^1, \cdots, \widetilde{\boldsymbol{\xi}}_0^k, \cdots, \boldsymbol{\xi}_0^N) \right|$$
$$= \frac{1}{N} \left| M_{-\varphi(\boldsymbol{z}_i; \cdot)\varphi(\boldsymbol{z}_j; \cdot)}^{\frac{1}{2\lambda}}(\boldsymbol{\xi}_0^k) - M_{-\varphi(\boldsymbol{z}_i, \cdot)\varphi(\boldsymbol{z}_j, \cdot)}^{\frac{1}{2\lambda}}(\widetilde{\boldsymbol{\xi}}_0^k) \right|. \qquad (58)$$

Without loss of generality suppose $M_{-\varphi(\boldsymbol{z}_i; \cdot)\varphi(\boldsymbol{z}_j; \cdot)}^{\frac{1}{2\lambda}}(\boldsymbol{\xi}_0^k) \ge M_{-\varphi(\boldsymbol{z}_i; \cdot)\varphi(\boldsymbol{z}_j; \cdot)}^{\frac{1}{2\lambda}}(\widetilde{\boldsymbol{\xi}}_0^k)$. Then,

$$M_{-\varphi(\boldsymbol{z}_i; \cdot)\varphi(\boldsymbol{z}_j; \cdot)}^{\frac{1}{2\lambda}}(\boldsymbol{\xi}_0^k) - M_{-\varphi(\boldsymbol{z}_i, \cdot)\varphi(\boldsymbol{z}_j, \cdot)}^{\frac{1}{2\lambda}}(\widetilde{\boldsymbol{\xi}}_0^k)$$
$$= \inf_{\boldsymbol{\zeta} \in \mathrm{I\!R}^D} \left\{ \lambda \|\boldsymbol{\zeta} - \boldsymbol{\xi}_0^k\|_2^2 - \varphi(\boldsymbol{z}_i; \boldsymbol{\zeta})\varphi(\boldsymbol{z}_j; \boldsymbol{\zeta}) \right\} - \inf_{\boldsymbol{\zeta} \in \mathrm{I\!R}^D} \left\{ \lambda \|\boldsymbol{\zeta} - \widetilde{\boldsymbol{\xi}}_0^k\|_2^2 - \varphi(\boldsymbol{z}_i; \boldsymbol{\zeta})\varphi(\boldsymbol{z}_j; \boldsymbol{\zeta}) \right\}$$
$$\overset{(a)}{\le} -\varphi(\boldsymbol{z}_i; \boldsymbol{\xi}_0^k)\varphi(\boldsymbol{z}_j; \boldsymbol{\xi}_0^k) - \inf_{\boldsymbol{\zeta} \in \mathrm{I\!R}^D} \left\{ \lambda \|\boldsymbol{\zeta} - \widetilde{\boldsymbol{\xi}}_0^k\|_2^2 - \varphi(\boldsymbol{z}_i; \boldsymbol{\zeta})\varphi(\boldsymbol{z}_j; \boldsymbol{\zeta}) \right\}$$
$$\overset{(b)}{\le} -\varphi(\boldsymbol{z}_i; \boldsymbol{\xi}_0^k)\varphi(\boldsymbol{z}_j; \boldsymbol{\xi}_0^k) + \sup_{\boldsymbol{\zeta} \in \mathrm{I\!R}^D} \left\{ \varphi(\boldsymbol{z}_i; \boldsymbol{\zeta})\varphi(\boldsymbol{z}_j; \boldsymbol{\zeta}) \right\}$$
$$\overset{(c)}{\le} 2L^2, \qquad (59)$$

where (a) follows by letting $\boldsymbol{\zeta} = \boldsymbol{\xi}_0^k$ in the first optimization problem, (b) follows by using the fact that $-\lambda \|\boldsymbol{\zeta} - \widetilde{\boldsymbol{\xi}}_0^k\|_2$ is non-positive for any $\boldsymbol{\zeta} \in \mathrm{I\!R}^D$ and can be dropped, and (c) follows from Assumption (**A.2**).

Now, plugging the upper bound in equation 59 into equation 58 yields

$$\left| T_{(\boldsymbol{z}_i, \boldsymbol{z}_j)}^\lambda(\boldsymbol{\xi}_0^1, \cdots, \boldsymbol{\xi}_0^k, \cdots, \boldsymbol{\xi}_0^N) - T_{(\boldsymbol{z}_i, \boldsymbol{z}_j)}^\lambda(\boldsymbol{\xi}_0^1, \cdots, \widetilde{\boldsymbol{\xi}}_0^k, \cdots, \boldsymbol{\xi}_0^N) \right| \le \frac{2L^2}{N}.$$

From McDiarmid's Martingale inequality McDiarmid (1989) and the union bound, we obtain that

$$\mathrm{I\!P} \left( \cup_{\boldsymbol{z}_i, \boldsymbol{z}_j \in \mathcal{Z}} |T_{(\boldsymbol{z}_i, \boldsymbol{z}_j)}^\lambda(\boldsymbol{\xi}_0^1, \cdots, \boldsymbol{\xi}_0^N)| \ge \delta \right) \le \left( \frac{4\mathrm{diam}(\mathcal{X})}{\epsilon} \right)^d \cdot \left( -\frac{N\delta^2}{L^2} \right), \qquad (60)$$

for all $\lambda \ge 0$. Now, consider arbitrary points $(\boldsymbol{x}, \widehat{\boldsymbol{x}}) \in \mathcal{X} \times \mathcal{X}$. Let the center of the balls containing those points be $\boldsymbol{z}_i, \boldsymbol{z}_j \in \mathcal{Z}$, *i.e.*, $\boldsymbol{x} \in \mathrm{I\!B}_\varepsilon(\boldsymbol{z}_i)$ and $\widehat{\boldsymbol{x}} \in \mathrm{I\!B}_\varepsilon(\boldsymbol{z}_j)$ for some $\boldsymbol{z}_i, \boldsymbol{z}_j \in \mathcal{Z}$. Then, by the triangle inequality, we have that

$$|T_{(\boldsymbol{x}, \widehat{\boldsymbol{x}})}^\lambda(\boldsymbol{\xi}_0^1, \cdots, \boldsymbol{\xi}_0^N) - T_{(\boldsymbol{z}_i, \boldsymbol{z}_j)}(\boldsymbol{\xi}_0^1, \cdots, \boldsymbol{\xi}_0^N)|$$
$$\le |T_{(\boldsymbol{x}, \widehat{\boldsymbol{x}})}^\lambda(\boldsymbol{\xi}_0^1, \cdots, \boldsymbol{\xi}_0^N) - T_{(\boldsymbol{z}_i, \widehat{\boldsymbol{x}})}^\lambda(\boldsymbol{\xi}_0^1, \cdots, \boldsymbol{\xi}_0^N)|$$
$$+ |T_{(\boldsymbol{z}_i, \widehat{\boldsymbol{x}})}^\lambda(\boldsymbol{\xi}_0^1, \cdots, \boldsymbol{\xi}_0^N) - T_{(\boldsymbol{z}_i, \boldsymbol{z}_j)}^\lambda(\boldsymbol{\xi}_0^1, \cdots, \boldsymbol{\xi}_0^N)|$$
$$\le \|\nabla_{\boldsymbol{x}} T_{(\boldsymbol{x}, \widehat{\boldsymbol{x}})}^\lambda(\boldsymbol{\xi}_0^1, \cdots, \boldsymbol{\xi}_0^N)\|_2 \|\boldsymbol{x} - \boldsymbol{z}_i\|_2 + \|\nabla_{\widehat{\boldsymbol{x}}} T_{(\boldsymbol{x}, \widehat{\boldsymbol{x}})}^\lambda(\boldsymbol{\xi}_0^1, \cdots, \boldsymbol{\xi}_0^N)\|_2 \|\widehat{\boldsymbol{x}} - \boldsymbol{z}_j\|_2$$
$$\le 2L_T \epsilon, \qquad (61)$$

where $L_T = L_T(\boldsymbol{\xi}_0^1, \cdots, \boldsymbol{\xi}_0^N) \overset{\text{def}}{=} \sup_{\boldsymbol{x}, \widehat{\boldsymbol{x}} \in \mathcal{X}} \|\nabla_{\boldsymbol{x}} T_{(\boldsymbol{x}, \widehat{\boldsymbol{x}})}^\lambda(\boldsymbol{\xi}_0^1, \cdots, \boldsymbol{\xi}_0^N)\|_2$ is the Lipschitz constant of the mapping $T$. Note that the Lipschitz constant $L_T$ is a random variable with respect to the random feature samples $\boldsymbol{\xi}_0, \cdots, \boldsymbol{\xi}_N$. Let $(\boldsymbol{x}_*, \widehat{\boldsymbol{x}}_*) \overset{\text{def}}{=} \arg\sup_{\boldsymbol{x}, \widehat{\boldsymbol{x}} \in \mathcal{X}} \|\nabla_{\boldsymbol{x}} T_{(\boldsymbol{x}, \widehat{\boldsymbol{x}})}^\lambda(\boldsymbol{\xi}_0^1, \cdots, \boldsymbol{\xi}_0^N)\|_2$.

We compute an upper bound on the second moment of the random variable $L_T$ as follows

$$\mathbb{E}_{\mu_0}\left[L_T^2\right] = \mathbb{E}_{\mu_0}\left[\|\nabla_{\boldsymbol{x}} T^\lambda_{(\boldsymbol{x}_*,\widehat{\boldsymbol{x}}_*)}(\boldsymbol{\xi}_0^1,\cdots,\boldsymbol{\xi}_0^N)\|_2^2\right]$$

$$= \mathbb{E}_{\mu_0}\left[\left\|\frac{1}{N}\sum_{k=1}^N \nabla_{\boldsymbol{x}} M^{\frac{1}{2\lambda}}_{-\varphi(\boldsymbol{x}_*;\cdot)\varphi(\widehat{\boldsymbol{x}}_*;\cdot)}(\boldsymbol{\xi}_0^k) - \int_{\mathbb{R}^D}\nabla_{\boldsymbol{x}} M^{\frac{1}{2\lambda}}_{-\varphi(\boldsymbol{x}_*;\cdot)\varphi(\widehat{\boldsymbol{x}}_*;\cdot)}(\boldsymbol{\xi})\mu_0(\mathrm{d}\boldsymbol{\xi})\right\|_2^2\right]$$

$$= \mathbb{E}_{\mu_0}\left[\left\|\frac{1}{N}\sum_{k=1}^N \nabla_{\boldsymbol{x}} M^{\frac{1}{2\lambda}}_{-\varphi(\boldsymbol{x}_*;\cdot)\varphi(\widehat{\boldsymbol{x}}_*;\cdot)}(\boldsymbol{\xi}_0^k)\right\|_2^2\right] - \mathbb{E}_{\mu_0}\left[\left\|\int_{\mathbb{R}^D}\nabla_{\boldsymbol{x}} M^{\frac{1}{2\lambda}}_{-\varphi(\boldsymbol{x}_*;\cdot)\varphi(\widehat{\boldsymbol{x}}_*;\cdot)}(\boldsymbol{\xi})\mu_0(\mathrm{d}\boldsymbol{\xi})\right\|_2^2\right]$$

$$\leq \frac{1}{N^2}\mathbb{E}_{\mu_0}\left[\left\|\sum_{k=1}^N \nabla_{\boldsymbol{x}} M^{\frac{1}{2\lambda}}_{-\varphi(\boldsymbol{x}_*;\cdot)\varphi(\widehat{\boldsymbol{x}}_*;\cdot)}(\boldsymbol{\xi}_0^k)\right\|_2^2\right].$$

We further proceed using the triangle inequality as well as the basic inequality $(a_1 + a_2 + \cdots + a_N)^2 \leq N(a_1^2 + a_2^2 + \cdots + a_N^2)$,

$$\mathbb{E}_{\mu_0}\left[L_T^2\right] = \frac{1}{N^2}\mathbb{E}_{\mu_0}\left[\left\|\sum_{k=1}^N \nabla_{\boldsymbol{x}} M^{\frac{1}{2\lambda}}_{-\varphi(\boldsymbol{x}_*;\cdot)\varphi(\widehat{\boldsymbol{x}}_*;\cdot)}(\boldsymbol{\xi}_0^k)\right\|_2^2\right]$$

$$\leq \frac{1}{N^2}\mathbb{E}_{\mu_0}\left[\left(\sum_{k=1}^N \left\|\nabla_{\boldsymbol{x}} M^{\frac{1}{2\lambda}}_{-\varphi(\boldsymbol{x}_*;\cdot)\varphi(\widehat{\boldsymbol{x}}_*;\cdot)}(\boldsymbol{\xi}_0^k)\right\|_2\right)^2\right]$$

$$\leq \frac{1}{N}\sum_{k=1}^N \mathbb{E}_{\mu_0}\left[\left\|\nabla_{\boldsymbol{x}} M^{\frac{1}{2\lambda}}_{-\varphi(\boldsymbol{x}_*;\cdot)\varphi(\widehat{\boldsymbol{x}}_*;\cdot)}(\boldsymbol{\xi}_0^k)\right\|_2^2\right]. \tag{62}$$

To proceed from equation 62, we leverage the following lemma:

**Lemma C.8.** (MOREAU'S ENVELOP OF PARAMETRIC FUNCTIONS) *Consider the parametric function* $f : \mathcal{X} \times \Theta \to \mathbb{R}$ *and the associated Moreau's envelope for a given* $\boldsymbol{\theta} \in \Theta \subset \mathbb{R}^d$:

$$M^\beta_{f(\cdot;\boldsymbol{\theta})}(\boldsymbol{x}) = \inf_{\boldsymbol{y}\in\mathcal{X}}\left\{\frac{1}{2\beta}\|\boldsymbol{x}-\boldsymbol{y}\|_2^2 + f(\boldsymbol{y};\boldsymbol{\theta})\right\}. \tag{63}$$

*Furthermore, define the proximal operator as follows*

$$\mathrm{Prox}^\beta_{f(\cdot;\boldsymbol{\theta})}(\boldsymbol{x}) = \arg\inf_{\boldsymbol{y}\in\mathcal{X}}\left\{\frac{1}{2\beta}\|\boldsymbol{x}-\boldsymbol{y}\|_2^2 + f(\boldsymbol{y};\boldsymbol{\theta})\right\}. \tag{64}$$

*Then, Moreau's envelope has the following upper bound*

$$\left\|\nabla_{\boldsymbol{\theta}} M^\beta_{f(\cdot;\boldsymbol{\theta})}(\boldsymbol{x})\right\|_2 \leq \left\|\nabla_{\boldsymbol{\theta}} f\left(\mathrm{Prox}^\beta_{f(\cdot;\boldsymbol{\theta})}(\boldsymbol{x});\boldsymbol{\theta}\right)\right\|_2. \tag{65}$$

The proof is presented in Appendix D.2.

Equipped with Inequality equation 65 of Lemma C.8, we now compute an upper bound on the right hand side of equation 62 as follows

$$\mathbb{E}_{\mu_0}\left[L_T^2\right] \leq \frac{1}{N}\sum_{k=1}^N \mathbb{E}_{\mu_0}[|\varphi(\widehat{\boldsymbol{x}}_*;\boldsymbol{\xi})|^2 \cdot \|\nabla_{\boldsymbol{x}}\varphi(\boldsymbol{x}_*;\boldsymbol{\xi}_0^k)\|_2^2] \leq L^4, \tag{66}$$

where the last inequality is due to $(\mathbf{A.2})$.

Invoking Markov's inequality now yields

$$\mathbb{P}\left(|T^\lambda_{(\boldsymbol{x},\widehat{\boldsymbol{x}})}(\boldsymbol{\xi}^1_0,\cdots,\boldsymbol{\xi}^N_0) - T^\lambda_{(\boldsymbol{z}_i,\boldsymbol{z}_j)}(\boldsymbol{\xi}^1_0,\cdots,\boldsymbol{\xi}^N_0)| \geq \delta\right) = \mathbb{P}\left(L_T \geq \frac{\delta}{2\epsilon}\right)$$

$$\leq \left(\frac{2\epsilon}{\delta}\right)^2 \mathbb{E}_{\mu_0}[L_T^2]$$

$$\leq \left(\frac{2\epsilon}{\delta}\right)^2 L^4. \tag{67}$$

Now, using the union bound, for every arbitrary pair of data points $(\boldsymbol{x},\widehat{\boldsymbol{x}}) \in \mathcal{X} \times \mathcal{X}$ the following inequality holds

$$\mathbb{P}\left(|T^\lambda_{(\boldsymbol{x},\widehat{\boldsymbol{x}})}(\boldsymbol{\xi}^1_0,\cdots,\boldsymbol{\xi}^N_0)| \geq \delta\right) \leq \mathbb{P}\left(|T^\lambda_{(\boldsymbol{z}_i,\boldsymbol{z}_j)}(\boldsymbol{\xi}^1_0,\cdots,\boldsymbol{\xi}^N_0)| \geq \delta/2\right)$$

$$+ \mathbb{P}\left(|T^\lambda_{(\boldsymbol{x},\widehat{\boldsymbol{x}})}(\boldsymbol{\xi}^1_0,\cdots,\boldsymbol{\xi}^N_0) - T^\lambda_{(\boldsymbol{z}_i,\boldsymbol{z}_j)}(\boldsymbol{\xi}^1_0,\cdots,\boldsymbol{\xi}^N_0)| \geq \delta/2\right) \tag{68}$$

$$\leq \left(\frac{2\epsilon}{\delta}\right)^2 L^4 + \left(\frac{4\mathrm{diam}(\mathcal{X})}{\epsilon}\right)^d \cdot \left(-\frac{N\delta^2}{L^2}\right).$$

Following the proposal of Rahimi & Recht (2008), we select $\epsilon = (\kappa_1/\kappa_2)^{\frac{1}{d+2}}$, where $\kappa_1 \stackrel{\text{def}}{=} (4\mathrm{diam}(\mathcal{X}))^d \cdot e^{-\frac{2N\lambda\delta^2}{2L^2\lambda+L^4}}$ and $\kappa_2 \stackrel{\text{def}}{=} (2/\delta)^2 L^4$. Then,

$$\mathbb{P}\left(\sup_{\boldsymbol{x},\widehat{\boldsymbol{x}}\in\mathcal{X}}|T^\lambda_{(\boldsymbol{x},\widehat{\boldsymbol{x}})}(\boldsymbol{\xi}^1_0,\cdots,\boldsymbol{\xi}^N_0)| \geq \delta\right) \leq 2^8 \left(\frac{L^2\mathrm{diam}(\mathcal{X})}{\delta}\right)^2 \cdot \left(-\frac{N\delta^2}{L^2(d+2)}\right).$$

Thus, with the probability of at least $1 - \varrho$, the following inequality holds

$$\sup_{\lambda\geq 0}\sup_{\boldsymbol{x},\widehat{\boldsymbol{x}}\in\mathcal{X}}|T^\lambda_{(\boldsymbol{x},\widehat{\boldsymbol{x}})}(\boldsymbol{\xi}^1_0,\cdots,\boldsymbol{\xi}^N_0)| \leq \left(\frac{L^2(d+2)}{N}\mathrm{W}\left(\frac{2^8 N\mathrm{diam}^2(\mathcal{X})}{\varrho}\right)\right)^{\frac{1}{2}}, \tag{69}$$

where $\mathrm{W}(\cdot)$ is the Lambert $W$-function.[4] Since $\mathrm{W}(x) \leq \ln(x)$ for $x > e$, we can rewrite the upper bound in terms of elementary functions

$$\sup_{\lambda\geq 0}\sup_{\boldsymbol{x},\widehat{\boldsymbol{x}}\in\mathcal{X}}|T^\lambda_{(\boldsymbol{x},\widehat{\boldsymbol{x}})}(\boldsymbol{\xi}^1_0,\cdots,\boldsymbol{\xi}^N_0)| \leq \sqrt{\frac{L^2(d+2)}{N}}\ln^{\frac{1}{2}}\left(\frac{2^8 N\mathrm{diam}^2(\mathcal{X})}{\varrho}\right), \tag{70}$$

provided that $N$ is sufficiently large and/or $\varrho$ is sufficiently small so that $\frac{2^8 N\mathrm{diam}^2(\mathcal{X})}{\varrho} \geq e$. Plugging Inequality equation 70 in equation 57 now results in the following inequality

$$\left|\sup_{\mu\in\mathcal{P}}\widehat{\mathrm{MMD}}_\mu[P_{\boldsymbol{V}},P_{\boldsymbol{W}}] - \sup_{\widehat{\mu}^N\in\mathcal{P}_N}\widehat{\mathrm{MMD}}_{\widehat{\mu}^N}[P_{\boldsymbol{V}},P_{\boldsymbol{W}}]\right| \leq \sqrt{\frac{L^2(d+2)}{N}}\ln^{\frac{1}{2}}\left(\frac{2^8 N\mathrm{diam}^2(\mathcal{X})}{\varrho}\right), \tag{71}$$

for all $\boldsymbol{W} \in \mathcal{W}$. Employingequation 53 from Lemma C.7 now yields the following upper bound

$$\mathsf{A}_2 = \left|\min_{\boldsymbol{W}\in\mathcal{W}}\sup_{\mu\in\mathcal{P}}\widehat{\mathrm{MMD}}_\mu[P_{\boldsymbol{V}},P_{\boldsymbol{W}}] - \min_{\boldsymbol{W}\in\mathcal{W}}\sup_{\widehat{\mu}^N\in\mathcal{P}_N}\widehat{\mathrm{MMD}}_{\widehat{\mu}^N}\left[P_{\boldsymbol{V}},P_{\boldsymbol{W}}\right]\right|$$

$$\leq \sqrt{\frac{L^2(d+2)}{N}}\ln^{\frac{1}{2}}\left(\frac{2^8 N\mathrm{diam}^2(\mathcal{X})}{\varrho}\right). \tag{72}$$

---

[4]Recall that the lambert $W$-function is the inverse of the function $f(W) = We^W$.

**Upper bound on** $A_3$:

Recall that the solution of the empirical risk function of equation 14 is denoted by

$$(\widehat{\boldsymbol{W}}_*^N, \widehat{\mu}_*^N) \stackrel{\text{def}}{=} \arg\min_{\boldsymbol{W} \in \mathcal{W}} \arg\inf_{\widehat{\mu}^N \in \mathcal{P}_N} \widehat{\text{MMD}}_{\widehat{\mu}^N}^{\alpha}[P_{\boldsymbol{V}}, P_{\boldsymbol{W}}]$$

$$= \arg\min_{\boldsymbol{W} \in \mathcal{W}} \arg\sup_{\widehat{\mu}^N \in \mathcal{P}_N} \frac{8}{n(n-1)} \sum_{1 \leq i < j \leq n} y_i y_j \mathbb{E}_{\widehat{\mu}^N \in \mathcal{P}_N}[\varphi(\boldsymbol{x}_i; \boldsymbol{\xi})\varphi(\boldsymbol{x}_j; \boldsymbol{\xi})]$$

$$- \frac{8}{n(n-1)\alpha} \sum_{1 \leq i < j \leq n} (\mathbb{E}_{\widehat{\mu}^N}[\varphi(\boldsymbol{x}_i; \boldsymbol{\xi})\varphi(\boldsymbol{x}_j; \boldsymbol{\xi})])^2. \tag{73}$$

We also define the solution of the empirical kernel alignment as follows

$$(\widehat{\boldsymbol{W}}_\diamond^N, \widehat{\mu}_\diamond^N) \stackrel{\text{def}}{=} \arg\min_{\boldsymbol{W} \in \mathcal{W}} \arg\sup_{\widehat{\mu}^N \in \mathcal{P}_N} \widehat{\text{MMD}}_{\widehat{\mu}^N}[P_{\boldsymbol{V}}, P_{\boldsymbol{W}}]$$

$$= \frac{8}{n(n-1)} \sum_{1 \leq i < j \leq n} y_i y_j \mathbb{E}_{\widehat{\mu}^N}[\varphi(\boldsymbol{x}_i; \boldsymbol{\xi})\varphi(\boldsymbol{x}_j; \boldsymbol{\xi})]. \tag{74}$$

Due to the optimality of the empirical measure $\widehat{\mu}_*^N$ for the inner optimization in equation 73, the following inequality holds

$$\widehat{\text{MMD}}_{\widehat{\mu}_\diamond^N}^{\alpha}[P_{\boldsymbol{V}}, P_{\widehat{\boldsymbol{W}}_*^N}] \leq \widehat{\text{MMD}}_{\widehat{\mu}_*^N}^{\alpha}[P_{\boldsymbol{V}}, P_{\widehat{\boldsymbol{W}}_*^N}]$$

$$\leq \frac{8}{n(n-1)} \sum_{1 \leq i < j \leq n} y_i y_j \mathbb{E}_{\widehat{\mu}_*^N}[\varphi(\boldsymbol{x}_i; \boldsymbol{\xi})\varphi(\boldsymbol{x}_j; \boldsymbol{\xi})]. \tag{75}$$

Upon expansion of $\widehat{\text{MMD}}_{\widehat{\mu}_\diamond^N}^{\alpha}[P_{\boldsymbol{V}}, P_{\widehat{\boldsymbol{W}}_*^N}]$, and after rearranging the terms in equation 75, we arrive at

$$\widehat{\text{MMD}}_{\widehat{\mu}_\diamond^N}[P_{\boldsymbol{V}}, P_{\widehat{\boldsymbol{W}}_*^N}] - \widehat{\text{MMD}}_{\widehat{\mu}_*^N}[P_{\boldsymbol{V}}, P_{\widehat{\boldsymbol{W}}_*^N}]$$

$$= \frac{8}{n(n-1)} \sum_{1 \leq i < j \leq n} y_i y_j (\mathbb{E}_{\widehat{\mu}_\diamond^N}[\varphi(\boldsymbol{x}_i; \boldsymbol{\xi})\varphi(\boldsymbol{x}_j; \boldsymbol{\xi})] - \mathbb{E}_{\widehat{\mu}_*^N}[\varphi(\boldsymbol{x}_i; \boldsymbol{\xi})\varphi(\boldsymbol{x}_j; \boldsymbol{\xi})])$$

$$\leq \frac{8}{n(n-1)\alpha} \sum_{1 \leq i < j \leq n} \left(\mathbb{E}_{\widehat{\mu}_\diamond^N}[\varphi(\boldsymbol{x}_i; \boldsymbol{\xi})\varphi(\boldsymbol{x}_j; \boldsymbol{\xi})]\right)^2$$

$$\leq \frac{8L^4}{\alpha}, \tag{76}$$

where the last inequality is due to the fact that $\|\varphi\|_\infty < L$ by **(A.1)**. Now, due to optimality of $\widehat{\boldsymbol{W}}_\diamond^N$ for the outer optimization problem in equation 74, we have

$$\widehat{\text{MMD}}_{\widehat{\mu}_\diamond^N}[P_{\boldsymbol{V}}, P_{\widehat{\boldsymbol{W}}_\diamond^N}] \leq \widehat{\text{MMD}}_{\widehat{\mu}_\diamond^N}[P_{\boldsymbol{V}}, P_{\widehat{\boldsymbol{W}}_*^N}]. \tag{77}$$

Putting together Inequalities equation 76 and equation 77 yields

$$\widehat{\text{MMD}}_{\widehat{\mu}_\diamond^N}[P_{\boldsymbol{V}}, P_{\widehat{\boldsymbol{W}}_\diamond^N}] - \widehat{\text{MMD}}_{\widehat{\mu}_*^N}[P_{\boldsymbol{V}}, P_{\widehat{\boldsymbol{W}}_*^N}] \leq \frac{8L^4}{\alpha}. \tag{78}$$

Similarly, due to the optimality of the empirical measure $\widehat{\mu}_\diamond^N$ for the optimization in equation 74 we have that

$$\widehat{\text{MMD}}_{\widehat{\mu}_*^N}\left[P_{\boldsymbol{V}}, P_{\widehat{\boldsymbol{W}}_*^N}\right] \leq \widehat{\text{MMD}}_{\widehat{\mu}_*^N}\left[P_{\boldsymbol{V}}, P_{\widehat{\boldsymbol{W}}_\diamond^N}\right]$$

$$\leq \widehat{\text{MMD}}_{\widehat{\mu}_\diamond^N}\left[P_{\boldsymbol{V}}, P_{\widehat{\boldsymbol{W}}_\diamond^N}\right]. \tag{79}$$

Combining equation 78 and equation 79 then yields

$$\mathsf{A}_3 = \left| \widehat{\mathrm{MMD}}_{\widehat{\mu}_*^N} \left[ P_{\boldsymbol{V}}, P_{\widehat{\boldsymbol{W}}_*^N} \right] - \widehat{\mathrm{MMD}}_{\widehat{\mu}_\diamond^N} \left[ P_{\boldsymbol{V}}, P_{\widehat{\boldsymbol{W}}_\diamond^N} \right] \right| \le \frac{8L^4}{\alpha}. \tag{80}$$

**Upper bound on $\mathsf{A}_4$:**

The upper bound on $\mathsf{A}_4$ can be obtained exactly the same way as $\mathsf{A}_1$. Specifically, from equation 52 it follows directly that

$$\mathsf{A}_4 = \left| \widehat{\mathrm{MMD}}_{\widehat{\mu}_*^N} \left[ P_{\boldsymbol{V}}, P_{\widehat{\boldsymbol{W}}_*^N} \right] - \mathrm{MMD}_{\widehat{\mu}_*^N} \left[ P_{\boldsymbol{V}}, P_{\widehat{\boldsymbol{W}}_*^N} \right] \right|$$

$$\le \sup_{\widehat{\mu}^N \in \mathcal{P}_N} \left| \widehat{\mathrm{MMD}}_{\widehat{\mu}^N} \left[ P_{\boldsymbol{V}}, P_{\widehat{\boldsymbol{W}}_*^N} \right] - \mathrm{MMD}_{\widehat{\mu}^N} \left[ P_{\boldsymbol{V}}, P_{\widehat{\boldsymbol{W}}_*^N} \right] \right| \tag{81}$$

$$\le \max \left\{ \frac{3^{\frac{1}{4}} \times 2^{\frac{11}{2}} \times L^2}{n} \ln^{\frac{1}{2}} \left( \frac{4}{\varrho} \right), \frac{9 \times 2^{12} \times RL^4}{n^2} \ln \left( \frac{4e^{\frac{L^4}{9}}}{\varrho} \right) \right\}. \tag{82}$$

Now, plugging the derived upper bounds in $\mathsf{A}_1$-$\mathsf{A}_4$ in equation 38 and employing the union bound completes the proof. ∎

## C.2 PROOF OF THEOREM 4.2

The proof has three main ingredients and follows the standard procedure in the literature, see, *e.g.*, Wang et al. (2017); Luo & Mattingly (2017). In the first step, we identify the mean-field limit of the particle SGD in equation 15. In the second step, we prove the convergence of the measured-valued process $\{(\mu_t^N)_{0 \le t \le T}\}$ to the mean-field solution by establishing the pre-compactness of Sokhorhod space. Lastly, we prove the uniqueness of the mean-field solution of the particle SGD.

**Step 1-Identification of the scaling limit**: First, we identify the weak limit of converging subsequences via the action of the empirical measure $\widehat{\mu}_m^N(\boldsymbol{\xi}) = \frac{1}{N} \sum_{k=1}^N \delta(\boldsymbol{\xi} - \boldsymbol{\xi}_m^k)$ on a test function $f \in C_b^3(\mathbb{R}^D)$. In particular, we use the standard techniques of computing the scaling limits from Luo & Mattingly (2017).

Recall that the action of an empirical measure on a bounded function is defined as follows

$$\langle f, \widehat{\mu}_m^N \rangle \stackrel{\text{def}}{=} \frac{1}{N} \sum_{k=1}^N f(\boldsymbol{\xi}_m^k). \tag{83}$$

We analyze the evolution of the empirical measure $\widehat{\mu}_m^N$ via its action on a test function $f \in C_b^3(\mathbb{R}^D)$. Using Taylor's expansion, we obtain

$$\langle f, \widehat{\mu}_{m+1}^N \rangle - \langle f, \widehat{\mu}_m^N \rangle = \langle f, \widehat{\mu}_{m+1}^N \rangle - \langle f, \widehat{\nu}_{m+1}^N \rangle$$

$$= \frac{1}{N} \sum_{k=1}^N f(\boldsymbol{\xi}_{m+1}^k) - f(\boldsymbol{\xi}_m^k)$$

$$= \frac{1}{N} \sum_{k=1}^N \nabla f(\boldsymbol{\xi}_m^k)(\boldsymbol{\xi}_{m+1}^k - \boldsymbol{\xi}_m^k)^T + R_m^N.$$

where $R_m^N$ is a remainder term defined as follows

$$R_m^N \stackrel{\text{def}}{=} \frac{1}{N} \sum_{k=1}^N (\boldsymbol{\xi}_{m+1}^k - \boldsymbol{\xi}_m^k)^T \nabla^2 f(\widetilde{\boldsymbol{\xi}}^k)(\boldsymbol{\xi}_{m+1}^k - \boldsymbol{\xi}_m^k), \tag{84}$$

where $\widetilde{\boldsymbol{\xi}}^k \stackrel{\text{def}}{=} (\widetilde{\xi}^k(1), \cdots, \widetilde{\xi}^k(p))$, and $\widetilde{\xi}^k(i) \in [\xi_m^k(i), \xi_{m+1}^k(i)]$, for $i = 1, 2, \cdots, p$.

Plugging the difference term $(\boldsymbol{\xi}_{m+1}^k - \boldsymbol{\xi}_m^k)$ from the SGD equation in equation 15 results in

$$\langle f, \widehat{\mu}_{m+1}^N \rangle - \langle f, \widehat{\mu}_m^N \rangle \tag{85}$$

$$= \frac{\eta}{N^2 \alpha} \sum_{k=1}^N \nabla f(\boldsymbol{\xi}_m^k) \cdot \left( \left( \frac{1}{N} \sum_{\ell=1}^N \varphi(\boldsymbol{x}_m; \boldsymbol{\xi}_m^\ell) \varphi(\widetilde{\boldsymbol{x}}_m; \boldsymbol{\xi}_m^\ell) - \alpha y_m \widetilde{y}_m \right) \nabla_{\boldsymbol{\xi}} \left( \varphi(\boldsymbol{x}_m; \boldsymbol{\xi}_m^k) \varphi(\widetilde{\boldsymbol{x}}_m; \boldsymbol{\xi}_m^k) \right) \right) + R_m^N.$$

Now, we define the drift and Martingale terms as follows

$$D_m^N \stackrel{\text{def}}{=} \frac{\eta}{N\alpha} \iint_{\mathcal{X} \times \mathcal{Y}} \left( \langle \varphi(\boldsymbol{x}, \boldsymbol{\xi}) \varphi(\widetilde{\boldsymbol{x}}, \boldsymbol{\xi}), \widehat{\mu}_m^N \rangle - \alpha y \widetilde{y} \right) \tag{86a}$$

$$\times \langle \nabla f(\boldsymbol{\xi})(\varphi(\widetilde{\boldsymbol{x}}; \boldsymbol{\xi}) \nabla_{\boldsymbol{\xi}} \varphi(\boldsymbol{x}; \boldsymbol{\xi}) + \varphi(\boldsymbol{x}; \boldsymbol{\xi}) \nabla_{\boldsymbol{\xi}} \varphi(\widetilde{\boldsymbol{x}}; \boldsymbol{\xi})), \widehat{\mu}_m^N \rangle \mathrm{d} P_{\boldsymbol{x}, y}^{\otimes 2}((\boldsymbol{x}, y), (\widetilde{\boldsymbol{x}}, \widetilde{y}))$$

$$M_m^N \stackrel{\text{def}}{=} \frac{\eta}{N\alpha} \left( \langle \varphi(\boldsymbol{x}_m, \boldsymbol{\xi}) \varphi(\widetilde{\boldsymbol{x}}_m, \boldsymbol{\xi}), \widehat{\mu}_m^N \rangle - \alpha y_m \widetilde{y}_m \right) \tag{86b}$$

$$\times \langle \nabla f(\boldsymbol{\xi})(\varphi(\widetilde{\boldsymbol{x}}_m; \boldsymbol{\xi}) \nabla_{\boldsymbol{\xi}} \varphi(\boldsymbol{x}_m; \boldsymbol{\xi}) + \varphi(\widetilde{\boldsymbol{x}}_m; \boldsymbol{\xi}) \nabla_{\boldsymbol{\xi}} \varphi(\boldsymbol{x}_m; \boldsymbol{\xi})), \widehat{\mu}_m^N \rangle - \mathcal{D}_m^N.$$

respectively. Using the definitions of $D_m^N$ and $M_m^N$ in equation 86a-equation 86b, we recast Equation equation 85 as follows

$$\langle f, \widehat{\mu}_{m+1}^N \rangle - \langle f, \widehat{\mu}_m^N \rangle = D_m^N + M_m^N + R_m^N. \tag{87}$$

Summation over $\ell = 0, 1, 2 \cdots, m-1$ and using the telescopic sum yields

$$\langle f, \widehat{\mu}_m^N \rangle - \langle f, \widehat{\mu}_0^N \rangle = \sum_{\ell=0}^{m-1} D_\ell^N + \sum_{\ell=0}^{m-1} M_\ell^N + \sum_{\ell=0}^{m-1} R_\ell^N. \tag{88}$$

We also define the following continuous embedding of the drift, martingale, and the remainder terms as follows

$$\mathcal{D}_t^N \stackrel{\text{def}}{=} \sum_{\ell=0}^{\lfloor Nt \rfloor} D_\ell^N \tag{89a}$$

$$\mathcal{M}_t^N \stackrel{\text{def}}{=} \sum_{\ell=0}^{\lfloor Nt \rfloor} M_\ell^N \tag{89b}$$

$$\mathcal{R}_t^N \stackrel{\text{def}}{=} \sum_{\ell=0}^{\lfloor Nt \rfloor} R_\ell^N, \quad t \in (0, T]. \tag{89c}$$

The scaled empirical measure $\mu_t^N \stackrel{\text{def}}{=} \widehat{\mu}_{\lfloor Nt \rfloor}^N$ then can be written as follows

$$\langle f, \mu_t^N \rangle - \langle f, \mu_0^N \rangle = \mathcal{D}_t^N + \mathcal{M}_t^N + \mathcal{R}_t^N. \tag{90}$$

Since the drift process $(D_t^N)_{0 \le t \le T}$ is a piecewise cádlág process, we have

$$D_\ell^N = \int_{\frac{\ell}{N}}^{\frac{\ell+1}{N}} R[\mu_s] \mathrm{d}s, \tag{91}$$

where the functional $R[\mu_s]$ is defined as follows

$$R[\mu_s] \stackrel{\text{def}}{=} \frac{\eta}{\alpha} \iint_{\mathcal{X} \times \mathcal{Y}} (\langle \varphi(\boldsymbol{x}, \boldsymbol{\xi}) \varphi(\widetilde{\boldsymbol{x}}, \boldsymbol{\xi}), \mu_s \rangle - \alpha y \widetilde{y}) \tag{92}$$

$$\times \langle \nabla f(\boldsymbol{\xi})(\varphi(\widetilde{\boldsymbol{x}}; \boldsymbol{\xi}) \nabla_{\boldsymbol{\xi}} \varphi(\boldsymbol{x}; \boldsymbol{\xi}) + \varphi(\boldsymbol{x}; \boldsymbol{\xi}) \nabla_{\boldsymbol{\xi}} \varphi(\widetilde{\boldsymbol{x}}; \boldsymbol{\xi}))^T, \mu_s \rangle P_{\boldsymbol{x}, y}^{\otimes 2}((\mathrm{d}\boldsymbol{x}, \mathrm{d}\widetilde{\boldsymbol{x}}), (\mathrm{d}y, \mathrm{d}\widetilde{y})).$$

Therefore, the expression in equation 90 can be rewritten as follows

$$\langle f, \mu_t^N \rangle - \langle f, \mu_0^N \rangle = \int_0^t R[\mu_s]\mathrm{d}s + \mathcal{M}_t^N + \mathcal{R}_t^N. \tag{93}$$

In the following lemma, we prove that the remainder term $\sup_{0 \leq t \leq T} |\mathcal{R}_t^N|$ vanishes in probabilistic sense as the number of particles tends to infinity $N \to \infty$:

**Lemma C.9.** (LARGE $N$-LIMIT OF THE REMAINDER PROCESS) *Consider the remainder process* $(\mathcal{R}_t^N)_{0 \leq t \leq T}$ *defined via scaling in equation 84-equation 89c. Then, there exists a constant $C_0 > 0$ such that*

$$\sup_{0 \leq t \leq T} |\mathcal{R}_t^N| \leq \frac{C_0 T}{N} \left( \eta L^2 + \frac{2\eta L^4}{\alpha} \right). \tag{94}$$

*and thus* $\limsup_{N \to \infty} \sup_{0 \leq t \leq T} |\mathcal{R}_t^N| = 0.$

*Proof.* The proof is relegated to Appendix D.5. □

We can also prove a similar result for the process defined by the remainder term:

**Lemma C.10.** (LARGE $N$-LIMIT OF THE MARTINGALE PROCESS) *Consider the Martingale process* $(\mathcal{M}_t^N)_{0 \leq t \leq T}$ *defined via scaling in equation 86b-equation 89b. Then, for some constant $C_1 > 0$, the following inequality holds*

$$\mathbb{P}\left( \sup_{0 \leq t \leq T} |\mathcal{M}_t^N| \geq \varepsilon \right) \leq \frac{1}{N\alpha\varepsilon} 4\sqrt{2}L^2 \sqrt{\lfloor NT \rfloor} \eta C_1 (L^2 + \alpha)^2. \tag{95}$$

*In particular, with the probability of at least $1 - \rho$, we have*

$$\sup_{0 \leq t \leq T} |\mathcal{M}_t^N| \leq \frac{1}{N\alpha\rho} 4\sqrt{2}L^2 \sqrt{\lfloor NT \rfloor} \eta C_1 (L^2 + \alpha)^2. \tag{96}$$

*and thus* $\limsup_{N \to \infty} \sup_{0 \leq t \leq T} |\mathcal{M}_t^N| = 0$ *almost surely.*

*Proof.* The proof is deferred to Appendix D.6. □

Now, using the results of Lemmata C.10 and C.9 in conjunction with equation 93 yields the following mean-field equation as $N \to \infty$,

$$\langle \mu_t, f \rangle = \langle \mu_0, f \rangle + \frac{\eta}{\alpha} \int_0^t \left( \iint_{\mathcal{X} \times \mathcal{Y}} (\langle \varphi(\boldsymbol{x}, \boldsymbol{\xi}) \varphi(\widetilde{\boldsymbol{x}}, \boldsymbol{\xi}), \mu_s \rangle - \alpha y \widetilde{y}) \tag{97}$$

$$\times \langle \nabla f(\boldsymbol{\xi})(\varphi(\widetilde{\boldsymbol{x}}; \boldsymbol{\xi}) \nabla_{\boldsymbol{\xi}} \varphi(\boldsymbol{x}; \boldsymbol{\xi}) + \varphi(\boldsymbol{x}; \boldsymbol{\xi}) \nabla_{\boldsymbol{\xi}} \varphi(\widetilde{\boldsymbol{x}}; \boldsymbol{\xi})), \mu_s \rangle P_{\boldsymbol{x}, y}^{\otimes 2}((\mathrm{d}\boldsymbol{x}, \mathrm{d}\widetilde{\boldsymbol{x}}), (\mathrm{d}y, \mathrm{d}\widetilde{y})) \right) \mathrm{d}s.$$

Notice that he mean-field equation in equation 97 is in the weak form. When the Lebesgue density $p_t(\boldsymbol{\xi}) = \mathrm{d}\mu_t / \mathrm{d}\boldsymbol{\xi}$ exists, the McKean-Vlasov PDE in equation 22 can be readily obtained from equation 97.

**Step 2: Pre-compactness of the Skorkhod space**: To establish our results in this part of the proof, we need a definition and a theorem:

**Definition C.11.** (TIGHTNESS) A set $\mathcal{A}$ of probability measures on a metric space $\mathcal{S}$ is tight if there exists a compact subset $\mathcal{S}_0 \subset \mathcal{S}$ such that

$$\nu(\mathcal{S}_0) \geq 1 - \varepsilon, \quad \text{for all } \nu \in \mathcal{A}, \tag{98}$$

for all $\varepsilon > 0$. A sequence $\{X^N\}_{N \in \mathbb{N}}$ of random elements of the metric space $\mathcal{S}$ is tight if there exists a compact subset $\mathcal{S}_0 \subset \mathcal{S}$ such that

$$\nu(X^N \in \mathcal{S}_0) > 1 - \varepsilon, \tag{99}$$

for all $\varepsilon > 0$, and all $N \in \mathbb{N}$.

Now, to show the tightness of the measured valued process $(\mu_t^N)_{0 \leq t \leq T}$, we must verify Jakubowski's criterion (Jakubowski, 1986, Thm. 1):

**Theorem C.12.** (JAKUBOWSKI'S CRITERION (JAKUBOWSKI, 1986, THM. 1)) *A sequence of measured-valued process $\{(\zeta_t^N)_{0 \leq t \leq T}\}_{N \in \mathbb{N}}$ is tight in $\mathcal{D}_{\mathcal{M}(\mathbb{R}^D)}([0, T])$ if and only if the following two conditions are satisfied:*

(**J.1**) *For each $T > 0$ and $\gamma > 0$, there exists a compact set $\mathcal{U}_{T,\gamma}$ such that*

$$\lim_{N \to \infty} \inf \mathbb{P}\left(\zeta_t^N \in \mathcal{U}_{T,\gamma}, \forall t \in (0, T]\right) > 1 - \gamma. \tag{100}$$

*This condition is referred to as the compact-containment condition.*

(**J.2**) *There exists a family $\mathcal{H}$ of real-valued functions $H : \mathcal{M}(\mathbb{R}^D) \mapsto \mathbb{R}$ that separates points in $\mathcal{M}(\mathbb{R}^D)$ and is closed under addition such that for every $H \in \mathcal{H}$, the sequence $\{(H(\xi_t^N))_{0 \leq t \leq T}\}_{N \in \mathbb{N}}$ is tight in $\mathcal{D}_{\mathbb{R}}([0, T])$.*

To establish (**J1**), we closely follow the proof of (Giesecke et al., 2013, Lemma 6.1.). In particular, for each $L > 0$, we define $\mathcal{S}_L = [0, B]^p$. Then, $\mathcal{S}_B \subset \mathbb{R}^p$ is compact, and for each $t \geq 0$, and $N \in \mathbb{N}$, we have

$$\mathbb{E}[\mu_t^N(\mathbb{R}^p / \mathcal{S}_B)] = \frac{1}{N} \sum_{k=1}^{N} \mathbb{P}\left(\|\boldsymbol{\xi}_{\lfloor Nt \rfloor}^k\|_2 \geq B\right) \tag{101}$$

$$\overset{(a)}{\leq} \frac{1}{N} \sum_{k=1}^{N} \frac{\mathbb{E}[\|\boldsymbol{\xi}_{\lfloor Nt \rfloor}^k\|_2]}{B} \tag{102}$$

$$\overset{(b)}{\leq} \frac{c_0 + \eta \alpha L^2 T + 2\eta L^4 T}{B}, \tag{103}$$

where (a) follows from Markov's inequality, and (b) follows from the upper bound on the norm of the particles in equation 177 of Appendix D. We now define the following set

$$\mathcal{U}_B = \left\{\mu \in \mathcal{M}(\mathbb{R}^p) : \mu(\mathbb{R}^p / \mathcal{S}_{(B+j)^2}) < \frac{1}{\sqrt{B+j}} \text{ for all } j \in \mathbb{N}\right\}. \tag{104}$$

We let $\mathcal{U}_{T,\gamma} = \overline{\mathcal{U}}_B$, where $\overline{\mathcal{U}}_B$ is the completion of the set $\mathcal{U}_B$. By definition, $\mathcal{U}_{T,\gamma}$ is a compact subset of $\mathcal{M}(\mathbb{R}^D)$. Now, we have

$$
\begin{aligned}
\mathbb{P}\left(\mu_t^N \notin \mathcal{U}_{T,\gamma}\right) &\leq \sum_{j=1}^\infty \mathbb{P}\left(\mu_t^N(\mathbb{R}^p/\mathcal{S}_{(B+j)^2}) > \frac{1}{\sqrt{B+j}}\right) \\
&\leq \sum_{j=1}^\infty \frac{\mathbb{E}[\mu_t^N(\mathbb{R}^p/\mathcal{S}_{(B+j)^2})]}{1/\sqrt{B+j}} \\
&\leq \sum_{j=1}^\infty \frac{c_0 + \eta L^2 T + 2(\eta/\alpha)L^4 T}{(B+j)^2/\sqrt{B+j}} \\
&= \sum_{j=1}^\infty \frac{c_0 + \eta L^2 T + 2(\eta/\alpha)L^4 T}{(B+j)^{3/2}}.
\end{aligned}
\tag{105}
$$

Now, since

$$
\lim_{B\to\infty} \sum_{j=1}^\infty \frac{c_0 + \eta L^2 T + 2(\eta/\alpha)L^4 T}{(B+j)^{3/2}} = 0,
\tag{106}
$$

this implies that for any $\gamma > 0$, there exists a $B > 0$, such that

$$
\lim_{N\to\infty} \inf \mathbb{P}\left(\mu_t^N \in \overline{\mathcal{U}}_B, \forall t \in (0,T]\right) > 1 - \gamma.
\tag{107}
$$

This completes the proof of (**J.1**). To verify (**J.2**), we consider the following class of functions

$$
\mathcal{H} \stackrel{\text{def}}{=} \{H : \exists f \in C_b^3(\mathbb{R}^D) \text{ such that } H(\mu) = \langle \mu, f\rangle, \forall \mu \in \mathcal{M}(\mathbb{R}^D)\}.
\tag{108}
$$

By definition, every function $H \in \mathcal{H}$ is continuous with respect to the weak topology of $\mathcal{M}(\mathbb{R}^D)$ and further the class of functions $\mathcal{H}$ separate points in $\mathcal{M}(\mathbb{R}^D)$ and is closed under addition. Now, we state the following sufficient conditions to establish (**J.2**). The statement of the theorem is due to (Robert, 2013, Thm. C.9):

**Theorem C.13.** (TIGHTNESS IN $\mathcal{D}_{\mathbb{R}}([0,T])$, (ROBERT, 2013, THM. C.9)) *A sequence* $\{(Z_t^N)_{0\leq t\leq T}\}_{N\in\mathbb{N}}$ *is tight in* $\mathcal{D}_{\mathbb{R}}([0,T])$ *iff for any* $\delta > 0$, *we have*

(**T.1**) *There exists $\epsilon > 0$, such that*

$$
\mathbb{P}(|Z_0^N| > \epsilon) \leq \delta,
\tag{109}
$$

*for all $N \in \mathbb{N}$.*

(**T.2**) *For any $\rho > 0$, there exists $\sigma > 0$ such that*

$$
\mathbb{P}\left(\sup_{t_1,t_2\leq T, |t_1-t_2|\leq\rho} |Z_{t_1}^N - Z_{t_2}^N| > \sigma\right) \leq \delta,
\tag{110}
$$

This completes the tightness proof of the of the laws of the measured-valued process $\{(\mu_t^N)_{0\leq t\leq T}\}_{N\in\mathbb{N}}$. Now, we verify the condition (**J.2**) by showing that the sufficient conditions (**T.1**) and (**T.2**) hold for function values $\{(H(\mu_t^N))_{0\leq t\leq T}\}_{N\in\mathbb{N}}$, where $H \in \mathcal{H}$ and $\mathcal{H}$ is defined in Eq. equation 108. Now, condition (**T.1**) is readily verified since

$$
H(\mu_0^N) = \langle \mu_0^N, f\rangle = \int_{\mathbb{R}^D} f(\boldsymbol{\xi})\mu_0^N(\mathrm{d}\boldsymbol{\xi})
\tag{111}
$$

$$
\leq \|f\|_\infty \int_{\mathbb{R}^D} \mu_0^N(\mathrm{d}\boldsymbol{\xi})
\tag{112}
$$

$$
\leq b,
\tag{113}
$$

where in the last step, we used the fact that $f \in C_b^3(\mathbb{R}^D)$, and hence, $\|f\|_\infty \le b$. Thus, $\mathbb{P}(H(\mu_0^N) \ge b) = 0$ for all $N \in \mathbb{N}$, and the condition $(\mathbf{T.1})$ is satisfied. Now, consider the condition $(\mathbf{T.2})$. From Equation equation 93, and with $0 \le t_1 < t_2 \le T$ we have

$$
\begin{aligned}
|H(\mu_{t_1}^N) - H(\mu_{t_1}^N)| &= |\langle f, \mu_{t_1}^N \rangle - \langle f, \mu_{t_2}^N \rangle| \\
&\le \int_{t_1}^{t_2} |R[\mu_s]|\, \mathrm{d}s + |\mathcal{M}_{t_1}^N - \mathcal{M}_{t_2}^N| + |\mathcal{R}_{t_1}^N - \mathcal{R}_{t_2}^N|.
\end{aligned}
\tag{114}
$$

To bound the first term, recall the definition of $R[\mu_s]$ from equation 92. The following chain of inequalities holds,

$$
\begin{aligned}
|R[\mu_s]| &\le \frac{\eta}{\alpha} \mathbb{E}_{P_{\boldsymbol{x},y}^{\otimes 2}}[|\langle \varphi(\boldsymbol{x},\boldsymbol{\xi})\varphi(\widetilde{\boldsymbol{x}},\boldsymbol{\xi}), \mu_s\rangle - \alpha y\widetilde{y}||\langle \nabla f(\boldsymbol{\xi})(\varphi(\widetilde{\boldsymbol{x}};\boldsymbol{\xi})\nabla_{\boldsymbol{\xi}}\varphi(\boldsymbol{x};\boldsymbol{\xi}) + \varphi(\boldsymbol{x};\boldsymbol{\xi})\nabla_{\boldsymbol{\xi}}\varphi(\widetilde{\boldsymbol{x}};\boldsymbol{\xi}))^T, \mu_s\rangle|] \\
&\le \frac{\eta}{\alpha} \mathbb{E}_{P_{\boldsymbol{x}}^{\otimes 2}}[(|\langle \varphi(\boldsymbol{x},\boldsymbol{\xi})\varphi(\widetilde{\boldsymbol{x}},\boldsymbol{\xi}), \mu_s\rangle| + \alpha)|\langle \nabla f(\boldsymbol{\xi})(\varphi(\widetilde{\boldsymbol{x}};\boldsymbol{\xi})\nabla_{\boldsymbol{\xi}}\varphi(\boldsymbol{x};\boldsymbol{\xi}) + \varphi(\boldsymbol{x};\boldsymbol{\xi})\nabla_{\boldsymbol{\xi}}\varphi(\widetilde{\boldsymbol{x}};\boldsymbol{\xi}))^T, \mu_s\rangle|].
\end{aligned}
\tag{115}
$$

Let $I : \mathbb{R}^D \to \mathbb{R}, I(\boldsymbol{\xi}) = 1$ denotes the identity function. Notice that $\langle I, \mu_s \rangle = \int_{\mathbb{R}^D} \mu_s(\mathrm{d}s) = 1$. From equation 115, we proceed as follows

$$
\begin{aligned}
|R[\mu_s]| &\le \frac{\eta}{\alpha} \mathbb{E}_{P_{\boldsymbol{X}}^{\otimes 2}}[(\|\varphi\|_\infty^2 \cdot |\langle I, \mu_s\rangle| + \alpha) \cdot \|\nabla f(\boldsymbol{\xi})(\varphi(\widetilde{\boldsymbol{x}};\boldsymbol{\xi})\nabla_{\boldsymbol{\xi}}\varphi(\boldsymbol{x};\boldsymbol{\xi}) + \varphi(\boldsymbol{x};\boldsymbol{\xi})\nabla_{\boldsymbol{\xi}}\varphi(\widetilde{\boldsymbol{x}};\boldsymbol{\xi}))^T\|_\infty \cdot |\langle I, \mu_s\rangle|] \\
&\le \frac{\eta}{\alpha} \mathbb{E}_{P_{\boldsymbol{X}}^{\otimes 2}}[(\|\varphi\|_\infty^2 + \alpha) \cdot \|\nabla f(\boldsymbol{\xi})(\varphi(\widetilde{\boldsymbol{x}};\boldsymbol{\xi})\nabla_{\boldsymbol{\xi}}\varphi(\boldsymbol{x};\boldsymbol{\xi}) + \varphi(\boldsymbol{x};\boldsymbol{\xi})\nabla_{\boldsymbol{\xi}}\varphi(\widetilde{\boldsymbol{x}};\boldsymbol{\xi}))^T\|_\infty] \\
&\le \frac{2\eta}{\alpha}(L^2 + \alpha)L^2 C_1,
\end{aligned}
\tag{116}
$$

where the last inequality is due to $(\mathbf{A.1})$. Therefore,

$$
\int_{t_1}^{t_2} |R[\mu_s]|\mathrm{d}s \le \mathfrak{s}_0 |t_2 - t_1|,
\tag{117}
$$

where $\mathfrak{s}_0 \stackrel{\text{def}}{=} \frac{2\eta}{\alpha}(L^2 + \alpha)L^2 C_1$.

Consider the middle term of equation 114. Using the definition of the martingale term in equation 89b, we obtain that

$$
\begin{aligned}
|\mathcal{M}_{t_1}^N - \mathcal{M}_{t_2}^N| &= \left| \sum_{\ell=0}^{\lfloor Nt_1 \rfloor} M_\ell^N - \sum_{\ell=0}^{\lfloor Nt_2 \rfloor} M_\ell^N \right| \\
&\le \left| \sum_{\ell=\lfloor Nt_1 \rfloor}^{\lfloor Nt_2 \rfloor} M_\ell^N \right|.
\end{aligned}
\tag{118}
$$

In Equation of Section D, we have proved the following concentration bound

$$
\mathbb{P}(|M_m^N| \ge \varepsilon) \le 2\left( -\frac{N^2 \alpha^2 \varepsilon^2}{8m L^4 \eta^2 C_1^2 (L^2 + \alpha)^2} \right), \quad \forall m \in [0, NT] \cap \mathbb{N}.
\tag{119}
$$

Now, recall the alternative definition of the sub-Gaussian random variables:

**Definition C.14.** (SUB-GAUSSIAN RANDOM VARIABLES BOUCHERON ET AL. (2013)) A random variable X is $\sigma^2$-sub-Gaussian if

$$
\mathbb{E}[(\lambda(X - \mathbb{E}[X]))] \le \left( \frac{\lambda^2 \sigma^2}{2} \right).
\tag{120}
$$

We enumerate a few standard consequences of sub-Gaussianity Boucheron et al. (2013). If $X_i$ are independent and $\sigma_i^2$-sub-Gaussian, then $\sum_{i=1}^n X_i$ is $\sum_{i=1}^n \sigma_i^2$ -sub-Gaussian. Moreover, $X$ is $\sigma^2$-sub-Gaussian if and only if

$$\mathbb{P}(|X - \mathbb{E}[X]| \geq \varepsilon) \leq \left(-\frac{\varepsilon^2}{2\sigma^2}\right). \tag{121}$$

Now, it is clear from equation 119 andthat $M_m^N$ is sub-Gaussian random variable with a zero mean, and with the parameter $\sigma_m^2 = \dfrac{4mL^4\eta^2 C_1^2(L^2 + \alpha)^2}{N^2\alpha^2}$. Therefore, $\sum_{\ell=\lfloor Nt_1 \rfloor}^{\lfloor Nt_2 \rfloor} M_\ell^N$ is sub-Gaussian with the parameter $\sigma^2(t_1, t_2) \stackrel{\text{def}}{=} \dfrac{2L^4\eta^2 C_1^2(L^2 + \alpha)^2}{N^2\alpha^2}(\lfloor Nt_1 \rfloor - \lfloor Nt_2 \rfloor + 1)(\lfloor Nt_1 \rfloor + \lfloor Nt_2 \rfloor)$. Consequently, from Inequality equation 118 and the concentration inequality in equation 121, we have

$$\mathbb{P}\left(\sup_{t_1,t_2 \leq T, |t_1-t_2| \leq \rho} |\mathcal{M}_{t_1}^N - \mathcal{M}_{t_2}^N| \geq \varepsilon\right) \leq \mathbb{P}\left(\sup_{t_1,t_2 \leq T, |t_1-t_2| \leq \rho} \left|\sum_{\ell=\lfloor Nt_1 \rfloor}^{\lfloor Nt_2 \rfloor} M_\ell^N\right| \geq \varepsilon\right) \tag{122}$$

$$= \mathbb{P}\left(\left|\sum_{\ell=\lfloor Nt_1^* \rfloor}^{\lfloor Nt_2^* \rfloor} M_\ell^N\right| \geq \varepsilon\right) \tag{123}$$

$$\leq 2\left(-\frac{\varepsilon^2}{\sigma^2(t_1^*, t_2^*)}\right) \tag{124}$$

$$\leq 2\left(-\frac{\alpha^2\varepsilon^2}{4L^4\eta^2 C_1^2(L^2 + \alpha)^2(\rho + 1)T}\right), \tag{125}$$

where $(t_1^*, t_2^*) \stackrel{\text{def}}{=} \arg\sup_{t_1,t_2 \leq T, |t_1-t_2| \leq \rho} \left|\sum_{\ell=\lfloor Nt_1 \rfloor}^{\lfloor Nt_2 \rfloor} M_\ell^N\right|$.

We first compute a bound for the last term of equation 114 using the definition of the scaled term $\mathcal{R}_t^N$ from equation 89c. We have

$$|\mathcal{R}_{t_1}^N - \mathcal{R}_{t_2}^N| = \left|\sum_{\ell=0}^{\lfloor Nt_1 \rfloor} R_\ell^N - \sum_{\ell=0}^{\lfloor Nt_2 \rfloor} R_\ell^N\right|$$

$$= \left|\sum_{\ell=\lfloor Nt_1 \rfloor}^{\lfloor Nt_2 \rfloor} R_\ell\right|$$

$$\leq \sum_{\ell=\lfloor Nt_1 \rfloor}^{\lfloor Nt_2 \rfloor} |R_\ell|$$

$$\stackrel{(a)}{\leq} |\lfloor Nt_2 \rfloor - \lfloor Nt_1 \rfloor| \frac{C_0}{N^2}(\eta L^2 + (L^4/\alpha))$$

$$\stackrel{(b)}{\leq} \mathfrak{s}_1|t_2 - t_1|, \tag{126}$$

where (a) follows from the upper bound in equation 178 of Section D, and in (b) we define $\mathfrak{s}_1 \stackrel{\text{def}}{=} \frac{C_0}{N}(\eta L^2 + (L^4/\alpha))$.

Putting together equation 117, equation 122, and equation 126, we conclude from Inequality equation 114 that

$$\mathbb{P}\Big(\sup_{t_1,t_2\leq T,|t_1-t_2|\leq\rho}|H(\mu_{t_1}^N)-H(\mu_{t_1}^N)|\geq\sigma\Big)\leq\mathbb{P}\Big(\sup_{t_1,t_2\leq T,|t_1-t_2|\leq\rho}|\mathcal{M}_{t_1}^N-\mathcal{M}_{t_2}^N|+(\mathfrak{s}_0+\mathfrak{s}_1)\rho\geq\sigma\Big)$$

$$\leq 2\left(-\frac{\alpha^2(\sigma-(\mathfrak{s}_0+\mathfrak{s}_1)\rho)^2}{4L^4\eta^2C_1^2(L^2+\alpha)^2(\rho+1)T}\right).$$

Therefore, condition **(T.2)** is also satisfied. Since the sufficient conditions **(T.1)** and **(T.2)** are satisfied, the condition **(J.2)** is satisfied. This completes the tightness proof of the measured-valued sequence $\{\mu_t^N\}_{N\in\mathbb{N}}$.

Now, we prove its convergence to a mean-field solution $(\mu_t^*)_{0\leq t\leq T}$.

**Theorem C.15.** (PROKHOROV'S THEOREM PROKHOROV (1956)) *A subset of probability measures on a complete separable metric space is tight if and only if it is pre-compact.*

According to Theorem C.15, the tightness of the Skorkhod Space $\mathcal{D}_{\mathcal{M}(\mathbb{R}^D)}([0,T])$ implies its pre-compactness which in turn implies the existence of a converging sub-sequence $\{(\mu_t^N)_{0\leq t\leq T}\}_{N_k}$ of $\{\mu_t^N\}_{N\in\mathbb{N}}$. Notice that $\{(\mu_t^N)_{0\leq t\leq T}\}_{N_k}$ is a stochastic process defined on the Skorkhod space. Therefore, let $\pi^{N_k}$ denotes the law of the converging sub-sequence $\{(\mu_t^N)_{0\leq t\leq T}\}_{N_k}$. By definition, $\pi^{N_k}$ is an element of the measure space $\mathcal{M}(\mathcal{D}_{[0,T]}(\mathcal{M}(\mathbb{R}^D)))$. In the sequel, we closely follow the argument of (Wang et al., 2017, Proposition 4) to show that the limiting measure $\pi^\infty$ is a Dirac's delta function concentrated at *a* mean-field solution $\mu_t^*\in\mathcal{D}_{[0,T]}(\mathcal{M}(\mathbb{R}^D))$. We define the following functional

$$F_t:\mathcal{D}_{[0,T]}(\mathcal{M}(\mathbb{R}^D))\to\mathbb{R},$$

$$\mu_t\mapsto F_t[\mu_t]=\left|\langle\mu_t,f\rangle-\langle\mu_0,f\rangle-\int_0^t R[\mu_s]\mathrm{d}s\right|.\qquad(127)$$

We compute the expectation of the functional $F_t$ with respect to $\pi^{N_k}$. We then have

$$\mathbb{E}_{\pi^{N_k}}[F_t(\mu)]=\mathbb{E}[F_t[\mu_t^N]]$$

$$=\mathbb{E}\left[\left|\langle\mu_t^{N_k},f\rangle-\langle\mu_0^N,f\rangle-\int_0^t R[\mu_s^{N_k}]\mathrm{d}s\right|.\right].\qquad(128)$$

Now, from Equation equation 93, we have that

$$\langle\mu_t^{N_k},f\rangle-\langle\mu_0^{N_k},f\rangle-\int_0^t R[\mu_s^{N_k}]\mathrm{d}s=\mathcal{M}_t^{N_k}+\mathcal{R}_t^{N_k}.\qquad(129)$$

Plugging equation 129 in equation 128 gives

$$\mathbb{E}_{\pi^{N_k}}[F_t(\mu)]=\mathbb{E}[F_t[\mu_t^{N_k}]]$$

$$=\mathbb{E}\left[\left|\mathcal{M}_t^{N_k}+\mathcal{R}_t^{N_k}\right|\right]$$

$$\leq\mathbb{E}\left[\sup_{0\leq t\leq T}|\mathcal{M}_t^{N_k}|\right]+\mathbb{E}\left[\sup_{0\leq t\leq T}|\mathcal{R}_t^{N_k}|\right]$$

$$=\frac{1}{N\alpha\rho}4\sqrt{2}L^2\sqrt{\lfloor NT\rfloor}\eta C_1(L^2+\alpha)^2+\frac{C_0T}{N}(\eta\alpha L^2T+2\eta L^4T),\qquad(130)$$

where the last equality is due to the bounds in equation 94 and equation 95 of Lemmata C.9 and C.10, respectively. Taking the limit of $N\to\infty$ from equation 130 yields

$$\lim_{N_k\to\infty}\mathbb{E}_{\pi^{N_k}}[|F_t[\mu]|]=0.\qquad(131)$$

It can be shown that the functional $F_t[\cdot]$ is continuous and bounded. Therefore, due the weak convergence of the sequence $\{\pi^{N_k}\}_{N_k \in \mathbb{N}}$ to $\pi^\infty$, equation 131 implies that

$$\mathbb{E}_{\pi^\infty}[|F_t(\mu)|] = 0. \tag{132}$$

Since the identity equation 132 holds for all bounded test functions $f \in C_b^3(\mathbb{R}^D)$ and for all $t \in (0, T]$, it follows that $\pi^\infty$ is a Dirac's delta function concentrated at *a* solution $(\mu_t^*)_{0 \le t \le T}$ of the mean-field equation.

**Step 3: Uniqueness of a mean-field solution**: Before we establish the uniqueness result we make two remarks:

First, we make it clear that from the compact-containment condition (**J.1**) of Jakubowski's criterion in Theorem C.12, the support of the measured-valued process $(\mu_t^N)_{0 \le t \le T} = (\widehat{\mu}_{\lfloor Nt \rfloor}^N)_{0 \le t \le T}$ is compact for all $0 \le t \le T$. Moreover, in Step 2 of the proof, we established that the measure valued process $(\mu_t^N)_{0 \le t \le T}$ converges weakly to *a* mean-field solution as the number of particles tends to infinity (*i.e.*, $N \to \infty$). Thus, all the possible solutions of the mean-field equation also have compact supports. Let $\widehat{\Xi} \subset \mathbb{R}^D$ denotes a compact set containing the supports of all such solutions at $0 \le t \le T$. In the sequel, it suffices to establish the uniqueness of the mean-field solution for the test functions with a compact domain, *i.e.*, let $f \in C_b^3(\widehat{\Xi})$.

Second, for all bounded continuous test functions $f \in C_b^3(\widehat{\Xi})$, the operator $f \to \langle \mu_t, f \rangle$ is a linear operator with $\mu_t(\mathbb{R}^D) = 1$. Hence, from Riesz-Markov-Kakutani representation theorem Rudin (1987); Varadarajan (1958) by assuming $\mu_t \in \mathcal{M}(\mathbb{R}^D)$, existence of unique operator implies $f \mapsto \langle f, \mu_t \rangle$ implies the existence of the unique probability measure $\mu_t$. Now, we equip the measure space $\mathcal{M}(\mathbb{R}^D)$ with the following norm

$$\|\mu\| \stackrel{\text{def}}{=} \sup_{\substack{f \in C_b^3(\widehat{\Xi}) \\ \|f\|_\infty \ne 0}} \frac{|\langle f, \mu \rangle|}{\|f\|_\infty}. \tag{133}$$

Given an initial measure $\mu_0$, we next prove that there exists at most one mean-field model solution by showing that there exists at most one real valued process $\langle \mu_t, f \rangle$ corresponding to the mean-field model. Suppose $(\mu_t^{*,1})_{0 \le t \le T}$, $(\mu_t^{*,2})_{0 \le t \le T}$ are two solutions satisfying the mean-field equations equation 97 with the initial distributions $\mu_0^1$, $\mu_0^2 \in \mathcal{M}(\mathbb{R}^D)$, respectively. For any test function $f \in C_b^3(\widehat{\Xi})$ we have that

$$\langle \mu_t^{*,1} - \mu_t^{*,2}, f \rangle = \langle \mu_0^1 - \mu_0^2, f \rangle + \frac{\eta}{\alpha} \int_0^t \left( \iint_{\mathcal{X} \times \mathcal{Y}} \left( \langle \varphi(\boldsymbol{x}, \boldsymbol{\xi}) \varphi(\widetilde{\boldsymbol{x}}, \boldsymbol{\xi}), \mu_s^{*,1} - \mu_s^{*,2} \rangle - \alpha y \widetilde{y} \right) \right.$$

$$\tag{134}$$

$$\left. \times \langle \nabla f(\boldsymbol{\xi})(\nabla_{\boldsymbol{\xi}}(\varphi(\widetilde{\boldsymbol{x}}; \boldsymbol{\xi}) \varphi(\boldsymbol{x}; \boldsymbol{\xi})))^T, \mu_s^{*,1} - \mu_s^{*,2} \rangle P_{\boldsymbol{x},y}^{\otimes 2}((\mathrm{d}\boldsymbol{z}, \mathrm{d}\widetilde{\boldsymbol{z}}) \right) \mathrm{d}s.$$

We bound the first term on the right side of Equation equation 134 as follows

$$\langle \mu_0^1 - \mu_0^2, f \rangle \le \|\mu_0^1 - \mu_0^2\| \cdot \|f\|_\infty \tag{135}$$

$$\le b\|\mu_0^1 - \mu_0^2\|, \tag{136}$$

where used the definition of the norm $\| \cdot \|$ on the measure space $\mathcal{M}(\mathbb{R}^D)$ from equation 133.

Furthermore, let

$$\iint_{\mathcal{X} \times \mathcal{Y}} \alpha y \widetilde{y} \langle \nabla f(\boldsymbol{\xi}) (\nabla_{\boldsymbol{\xi}} (\varphi(\widetilde{\boldsymbol{x}}; \boldsymbol{\xi}) \varphi(\boldsymbol{x}; \boldsymbol{\xi})))^T, \mu_s^{*,1} - \mu_s^{*,2} \rangle P_{\boldsymbol{x},y}^{\otimes 2}(\mathrm{d}(\boldsymbol{x}, y), \mathrm{d}(\widetilde{\boldsymbol{x}}, \widetilde{y}))$$

$$\leq \iint_{\mathcal{X} \times \mathcal{Y}} \alpha |y\widetilde{y}| \cdot |\langle \nabla f(\boldsymbol{\xi}) (\nabla_{\boldsymbol{\xi}} (\varphi(\widetilde{\boldsymbol{x}}; \boldsymbol{\xi}) \varphi(\boldsymbol{x}; \boldsymbol{\xi})))^T, \mu_s^{*,1} - \mu_s^{*,2} \rangle| P_{\boldsymbol{x}}^{\otimes 2}(\mathrm{d}(\boldsymbol{x}, y), \mathrm{d}(\widetilde{\boldsymbol{x}}, \widetilde{y}))$$

$$\leq \alpha \|\mu_s^{*,1} - \mu_s^{*,2}\| \int_{\mathcal{X}} \|\nabla f(\boldsymbol{\xi}) (\nabla_{\boldsymbol{\xi}} (\varphi(\widetilde{\boldsymbol{x}}; \boldsymbol{\xi}) \varphi(\boldsymbol{x}; \boldsymbol{\xi})))^T\| P_{\boldsymbol{x}}^{\otimes 2}(\mathrm{d}\boldsymbol{x}, \mathrm{d}\widetilde{\boldsymbol{x}})$$

$$\leq \alpha \|\mu_s^{*,1} - \mu_s^{*,2}\| \int_{\mathcal{X}} \|\nabla f(\boldsymbol{\xi})\|_{\infty} \cdot \|\nabla_{\boldsymbol{\xi}} \varphi(\widetilde{\boldsymbol{x}}; \boldsymbol{\xi}) \varphi(\boldsymbol{x}; \boldsymbol{\xi})\|_{\infty} P_{\boldsymbol{x}}^{\otimes 2}(\mathrm{d}\boldsymbol{x}, \mathrm{d}\widetilde{\boldsymbol{x}})$$

$$\leq \alpha L^2 C_1 \|\mu_s^{*,1} - \mu_s^{*,2}\|, \tag{137}$$

where in the last inequality, we used the fact that $\|\nabla f(\boldsymbol{\xi})\| \leq C_1$ since the test function is three-times continuously differentiable $f \in C_b^3(\widehat{\Xi})$ on a compact support.

Similarly, we have

$$\int_{\mathcal{X}} \langle \varphi(\boldsymbol{x}, \boldsymbol{\xi}) \varphi(\widetilde{\boldsymbol{x}}, \boldsymbol{\xi}), \mu_s^{*,1} - \mu_s^{*,2} \rangle \langle \nabla f(\boldsymbol{\xi}) (\nabla_{\boldsymbol{\xi}} \varphi(\boldsymbol{x}, \boldsymbol{\xi}) \varphi(\boldsymbol{x}, \boldsymbol{\xi})), \mu_s^{*,1} - \mu_s^{*,2} \rangle P_{\boldsymbol{x}}^{\otimes 2}(\mathrm{d}\boldsymbol{x}, \mathrm{d}\widetilde{\boldsymbol{x}})$$

$$\leq \|\mu_s^{*,1} - \mu_s^{*,2}\|^2 \int_{\mathcal{X}} \|\varphi(\boldsymbol{x}, \boldsymbol{\xi}) \varphi(\widetilde{\boldsymbol{x}}, \boldsymbol{\xi})\|_{\infty} \|\nabla f(\boldsymbol{\xi}) (\nabla_{\boldsymbol{\xi}} \varphi(\boldsymbol{x}, \boldsymbol{\xi}) \varphi(\boldsymbol{x}, \boldsymbol{\xi}))^T\|_{\infty} P_{\boldsymbol{x}}^{\otimes 2}(\mathrm{d}\boldsymbol{x}, \mathrm{d}\widetilde{\boldsymbol{x}})$$

$$\leq L^4 C_1 \|\mu_s^{*,1} - \mu_s^{*,2}\|^2. \tag{138}$$

Putting together the inequalities in equation 136,equation 137, and equation 138 yield

$$\langle \mu_t^{*,1} - \mu_t^{*,2}, f \rangle \leq b\|\mu_0^1 - \mu_0^2\| + L^2 C_1 \eta \int_0^t \|\mu_s^{*,1} - \mu_s^{*,2}\| \mathrm{d}s + \frac{\eta L^4 C_1}{\alpha} \int_0^t \|\mu_s^{*,1} - \mu_s^{*,2}\|^2 \mathrm{d}s. \tag{139}$$

The above inequality holds for all bounded functions $f \in C_b^3(\widehat{\Xi})$. Thus, by taking the supremum with respect to $f$ we obtain

$$\|\mu_t^{*,1} - \mu_t^{*,2}\| = \sup_{f \in C_b^3(\widehat{\Xi})} \langle \mu_t^{*,1} - \mu_t^{*,2}, f \rangle \tag{140}$$

$$\leq b\|\mu_0^1 - \mu_0^2\| + L^2 C_1 \eta \int_0^t \|\mu_s^{*,1} - \mu_s^{*,2}\| \mathrm{d}s + \frac{L^4 C_1 \eta}{\alpha} \int_0^t \|\mu_s^{*,1} - \mu_s^{*,2}\|^2 \mathrm{d}s. \tag{141}$$

Now, we employ the following result which generalizes Gronewall's inequality when higher order terms are involved:

**Lemma C.16.** (EXTENDED GRONEWALL'S INEQUALITY, (WEBB, 2018, THM 2.1.)) *Let $p \in \mathbb{N}$ and suppose that for a.e. $t \in [0, T]$, $u \in L_+^{\infty}[0, T]$ satisfies*

$$u_t \leq c_0(t) + \int_0^t (c_1(s)u_s + c_2(s)u_s^2 + \cdots + c_{p+1}(s)u_s^{p+1}) \mathrm{d}s, \tag{142}$$

*where $c_0 \in L_{\infty}[0, T]$ is non-decreasing, and $c_j \in L_+^1[0, T]$ for $j \in \{1, \cdots, p+1\}$. Then, if*

$$\int_0^T c_{j+1}(s)u_s^j \mathrm{d}s \leq M_j, \quad j \in \{1, 2, \cdots, p\}. \tag{143}$$

*It follows that for a.e. $t \in [0, T]$*

$$u_t \leq c_0(t) \left( \int_0^t c_1(s) \mathrm{d}s \right) (M_1 + \cdots + M_p). \tag{144}$$

We now apply the extended Gronewall's Inequality equation 142 with $p = 1$, $c_0(t) = b\|\mu_0^1 - \mu_0^2\|$, $0 \le t \le T$, $c_1(t) = \eta L^2 C_1$, $0 \le t \le T$, $c_2(t) = \frac{\eta L^4 C_1}{\alpha}$, $0 \le t \le T$, and $u_s = \|\mu_s^{*,1} - \mu_s^{*,2}\|$. In this case, it is easy to see that $M_1 = \frac{2b\eta T L^4 C_1}{\alpha}$. Consequently, from equation 140 and equation 144, we obtain that

$$\|\mu_t^{*,1} - \mu_t^{*,2}\| \le b\|\mu_0^1 - \mu_0^2\| \cdot \left( \eta L^2 C_1 t + \frac{2bT\eta L^4 C_1}{\alpha} \right), \quad 0 \le t \le T. \tag{145}$$

Thus, starting from an initial measure $\mu_0^1 = \mu_0^2 = \mu_0$, there exists at most one solution for the mean-field model equations equation 97.

∎

## C.3 PROOF OF COROLLARY 4.2.1

To establish the proof, we recall from equation 88 that

$$\langle f, \widehat{\mu}_m^N \rangle - \langle f, \widehat{\mu}_0^N \rangle = \sum_{\ell=0}^{m-1} D_\ell^N + \sum_{\ell=0}^{m-1} M_\ell^N + \sum_{\ell=0}^{m-1} R_\ell^N, \tag{146}$$

for all $f \in C_b(\mathbb{R}^D)$. Recall the definition of the total variation distance $\mathrm{TV}(\cdot, \cdot)$ on a metric space $(\mathcal{X}, d)$.

$$\mathrm{TV}(\mu, \nu) = \frac{1}{2} \sup_{\mathcal{A} \subset \mathcal{X}} |\mu(\mathcal{A}) - \nu(\mathcal{A})|. \tag{147}$$

The total variation distance admits the following variational form

$$\mathrm{TV}(\mu, \nu) = \sup_{f : \|f\|_\infty \le \frac{1}{2}} \langle f, \mu \rangle - \langle f, \nu \rangle. \tag{148}$$

Now, using the variation form and by taking the supremum of equation 146 with respect to the functions from the function class $\mathcal{F}_c \overset{\text{def}}{=} \{ f \in C_{1/2}(\mathbb{R}^D) \}$, we obtain the following upper bound on the TV distance

$$\mathrm{TV}(\widehat{\mu}_m^N, \widehat{\mu}_0^N) \le \frac{1}{2} \sum_{\ell=0}^{m-1} |D_\ell^N| + \frac{1}{2} \sum_{\ell=0}^{m-1} |M_\ell^N| + \frac{1}{2} \sum_{\ell=0}^{m-1} |R_\ell^N|. \tag{149}$$

Based on the upper bound equation 178 on the remainder term, we have

$$|R_\ell^N| \le \frac{C_0}{N^2} \left( \eta L^2 + 2\frac{\eta}{\alpha} L^4 \right), \quad \ell \in [0, m-1], \tag{150}$$

for some constant $C_0 > 0$. Moreover, from the concentration inequality equation 189, we also have that with the probability of at least $1 - \delta$, the following inequality holds

$$|M_\ell^N| \le \frac{8\sqrt{\ell}\eta L^2 C_1 (L^2 + \alpha)}{N\alpha} \log\left( \frac{2}{\delta} \right). \tag{151}$$

Lastly, recall the definition of the drift term in equation 86a. By carrying out a similar bounding method leading to equation 116, it can be shown that

$$|\mathcal{D}_\ell^N| \le \frac{2\eta}{N\alpha} (L^2 + \alpha) L^2 C_1. \tag{152}$$

By plugging equation 150, equation 151, and equation 152 into equation 149, we derive that

$$\mathrm{TV}\left( \widehat{\mu}_m^N, \widehat{\mu}_0^N \right) \le \frac{m\eta C_0}{2N^2} \left( L^2 + 2\frac{L^4}{\alpha} \right) + \frac{8m\sqrt{m}\eta L^2 C_1 (L^2 + \alpha)}{N\alpha} \log\left( \frac{2}{\delta} \right) + \frac{m\eta}{N\alpha} (L^2 + \alpha) L^2 C_1, \tag{153}$$

with the probability of $1 - \delta$. We now leverage the following lemma:

**Lemma C.17.** (BOUNDED EQUIVALENCE OF THE WASSERSTEIN AND TOTAL VARIATION DISTANCES, SINGH & PÓCZOS (2018)) *Suppose $(\mathcal{X}, d)$ is a metric space, and suppose $\mu$ and $\nu$ are Borel probability measures on $\mathcal{X}$ with countable support; i.e., there exists a countable set $\mathcal{X}' \subseteq \mathcal{X}$ such that $\mu(\mathcal{X}') = \nu(\mathcal{X}') = 1$. Then, for any $p \geq 1$, we have*

$$\text{Sep}(\mathcal{X}')(2\text{TV}(\mu,\nu))^{\frac{1}{p}} \leq W_p(\mu,\nu) \leq \text{Diam}(\mathcal{X}')(2\text{TV}(\mu,\nu))^{\frac{1}{p}}, \tag{154}$$

*where $\text{Diam}(\mathcal{X}') \stackrel{def}{=} \sup_{x,y \in \mathcal{X}'} d(x,y)$, and $\text{Sep}(\mathcal{X}') \stackrel{def}{=} \inf_{x \neq y \in \mathcal{X}'} d(x,y)$.*

Consider the metric space $(\mathbb{R}^D, \|\cdot\|_2)$. Note that the empirical measures $\widehat{\mu}_m^N, \widehat{\mu}_0^N$ have a countable support $\mathcal{X}' = \{\boldsymbol{\xi}_m^k\}_{k=1}^N \cup \{\boldsymbol{\xi}_0^k\}_{k=1}^N \subset \mathbb{R}^D$. Therefore, using the upper bounds in equation 154 of Lemma C.17 and 153, we conclude that when the step-size is of the order

$$\eta = \mathcal{O}\left(\frac{R^p}{T\sqrt{NT}\log(2/\delta)}\right), \tag{155}$$

then $W_p(\widehat{\mu}_m^N, \widehat{\mu}_0^N) \leq R$ for all $m \in [0, NT] \cap \mathbb{N}$. ∎

# D    PROOFS OF AUXILIARY RESULTS

## D.1    PROOF OF LEMMA C.5

The upper bound follows trivially by letting $\boldsymbol{x} = \boldsymbol{y}$ in the optimization problem equation 44.

Now, consider the lower bound. Define the function $g : [0,1] \to \mathbb{R}, t \mapsto g(t) = f(\boldsymbol{y} + t(\boldsymbol{x} - \boldsymbol{y}))$. Then, when $f$ is differentiable, we have $g'(t) = \langle \boldsymbol{x} - \boldsymbol{y}, \nabla f(\boldsymbol{y} + t(\boldsymbol{x} - \boldsymbol{y})) \rangle$. In addition, $g(0) = f(\boldsymbol{y})$, and $g(1) = f(\boldsymbol{x})$. Based on the basic identity $g(1) = g(0) + \int_0^1 g'(s) \mathrm{d}s$, we derive

$$f(\boldsymbol{x}) = f(\boldsymbol{y}) + \int_0^1 \langle \boldsymbol{x} - \boldsymbol{y}, \nabla f(\boldsymbol{y} + s(\boldsymbol{x} - \boldsymbol{y})) \rangle \mathrm{d}s$$

$$\geq f(\boldsymbol{y}) - \|\boldsymbol{x} - \boldsymbol{y}\|_2 \int_0^1 \|\nabla f(\boldsymbol{y} + s(\boldsymbol{x} - \boldsymbol{y}))\|_2 \mathrm{d}s, \tag{156}$$

where the last step is due to the Cauchy-Schwarz inequality. Using Inequality equation 156 yields the following lower bound on Moreau's envelope

$$M_f^\beta(\boldsymbol{y}) \geq f(\boldsymbol{y}) + \inf_{\boldsymbol{x} \in \mathcal{X}} \left\{ \frac{1}{2\beta}\|\boldsymbol{x} - \boldsymbol{y}\|_2^2 - \|\boldsymbol{x} - \boldsymbol{y}\|_2 \int_0^1 \|\nabla f(\boldsymbol{y} + s(\boldsymbol{x} - \boldsymbol{y}))\|_2 \mathrm{d}s \right\}$$

$$= f(\boldsymbol{y}) + \inf_{\boldsymbol{x} \in \mathcal{X}} \left\{ \left( \frac{1}{\sqrt{2\beta}}\|\boldsymbol{x} - \boldsymbol{y}\|_2 - \sqrt{\frac{\beta}{2}} \int_0^1 \|\nabla f(\boldsymbol{y} + s(\boldsymbol{x} - \boldsymbol{y}))\|_2 \mathrm{d}s \right)^2 \right.$$

$$\left. - \frac{\beta}{2} \left( \int_0^1 \|\nabla f(\boldsymbol{y} + s(\boldsymbol{x} - \boldsymbol{y}))\|_2 \mathrm{d}s \right)^2 \right\}$$

$$\geq f(\boldsymbol{y}) - \frac{\beta}{2} \sup_{\boldsymbol{x} \in \mathcal{X}} \left( \int_0^1 \|\nabla f(\boldsymbol{y} + s(\boldsymbol{x} - \boldsymbol{y}))\|_2 \mathrm{d}s \right)^2$$

$$\stackrel{(a)}{\geq} f(\boldsymbol{y}) - \frac{\beta}{2} \sup_{\boldsymbol{x} \in \mathcal{X}} \int_0^1 \|\nabla f(\boldsymbol{y} + s(\boldsymbol{x} - \boldsymbol{y}))\|_2^2 \mathrm{d}s$$

$$\geq f(\boldsymbol{y}) - \frac{\beta}{2} \int_0^1 \sup_{\boldsymbol{x} \in \mathcal{X}} \|\nabla f(\boldsymbol{y} + s(\boldsymbol{x} - \boldsymbol{y}))\|_2^2 \mathrm{d}s, \tag{157}$$

where (a) is due to Jensen's inequality. ∎

## D.2  PROOF OF LEMMA C.8

Let $\boldsymbol{u} \in \mathbb{R}^d$ denotes an arbitrary unit vector $\|\boldsymbol{u}\|_2 = 1$. From the definition of the gradient of a function, we have that

$$\left\langle \nabla_{\boldsymbol{\theta}} M_{f(\cdot;\boldsymbol{\theta})}^{\beta}(\boldsymbol{x}), \boldsymbol{u} \right\rangle = D_{\boldsymbol{u}}[M_{f(\cdot;\boldsymbol{\theta})}(\boldsymbol{x})], \tag{158}$$

where $D_{\boldsymbol{u}}[M_{f(\cdot;\boldsymbol{\theta})}(\boldsymbol{x})]$ is the directional derivative

$$D_{\boldsymbol{u}}[M_{f(\cdot;\boldsymbol{\theta})}(\boldsymbol{x})] \overset{\text{def}}{=} \lim_{\delta \to 0} \frac{M_{f(\cdot;\boldsymbol{\theta}+\delta\boldsymbol{u})}(\boldsymbol{x}) - M_{f(\cdot;\boldsymbol{\theta})}(\boldsymbol{x})}{\delta}. \tag{159}$$

We now have

$$\begin{aligned}
M_{f(\cdot;\boldsymbol{\theta}+\delta\boldsymbol{u})}(\boldsymbol{x}) &= \inf_{\boldsymbol{x} \in \mathcal{X}} \left\{ \frac{1}{2\beta}\|\boldsymbol{x} - \boldsymbol{y}\|_2^2 + f(\boldsymbol{y};\boldsymbol{\theta}+\delta\boldsymbol{u}) \right\} \\
&= \inf_{\boldsymbol{x} \in \mathcal{X}} \left\{ \frac{1}{2\beta}\|\boldsymbol{x} - \boldsymbol{y}\|_2^2 + f(\boldsymbol{y};\boldsymbol{\theta}) + \delta\langle\nabla_{\boldsymbol{\theta}}f(\boldsymbol{y};\boldsymbol{\theta}),\boldsymbol{u}\rangle \right\} + \mathcal{O}(\delta^2) \\
&\overset{(a)}{\leq} \frac{1}{2\beta}\|\boldsymbol{x} - \mathrm{Prox}_{f(\cdot;\boldsymbol{\theta})}(\boldsymbol{x})\|_2^2 + f(\mathrm{Prox}_{f(\cdot;\boldsymbol{\theta})}(\boldsymbol{x});\boldsymbol{\theta}) \\
&\quad + \delta\langle\nabla_{\boldsymbol{\theta}}f(\mathrm{Prox}_{f(\cdot;\boldsymbol{\theta})}(\boldsymbol{x});\boldsymbol{\theta}),\boldsymbol{u}\rangle + \mathcal{O}(\delta^2),
\end{aligned} \tag{160}$$

where the inequality in (a) follows by letting $\boldsymbol{y} = \mathrm{Prox}_{f(\cdot;\boldsymbol{\theta})}(\boldsymbol{x})$ in the optimization problem. Now, recall that

$$M_{f(\cdot;\boldsymbol{\theta})}(\boldsymbol{x}) = \inf_{\boldsymbol{y} \in \mathcal{X}} \left\{ \frac{1}{2\beta}\|\boldsymbol{x} - \boldsymbol{y}\|_2^2 + f(\boldsymbol{y};\boldsymbol{\theta}) \right\},$$

$$\mathrm{Prox}_{f(\cdot;\boldsymbol{\theta})}(\boldsymbol{x}) = \arg\min_{\boldsymbol{y} \in \mathcal{X}} \left\{ \frac{1}{2\beta}\|\boldsymbol{x} - \boldsymbol{y}\|_2^2 + f(\boldsymbol{y};\boldsymbol{\theta}) \right\}.$$

Therefore,

$$M_{f(\cdot;\boldsymbol{\theta})}(\boldsymbol{x}) = \frac{1}{2\beta}\|\boldsymbol{x} - \mathrm{Prox}_{f(\cdot;\boldsymbol{\theta})}(\boldsymbol{x})\|_2^2 + f(\mathrm{Prox}_{f(\cdot;\boldsymbol{\theta})}(\boldsymbol{x});\boldsymbol{\theta}). \tag{161}$$

Substitution of equation 161 in equation 160 yields

$$M_{f(\cdot;\boldsymbol{\theta}+\delta\boldsymbol{u})}(\boldsymbol{x}) \leq M_{f(\cdot;\boldsymbol{\theta})}(\boldsymbol{x}) + \delta\langle\nabla_{\boldsymbol{\theta}}f(\mathrm{Prox}_{f(\cdot;\boldsymbol{\theta})}(\boldsymbol{x});\boldsymbol{\theta}),\boldsymbol{u}\rangle + \mathcal{O}(\delta^2). \tag{162}$$

Hence, $D_{\boldsymbol{u}}[M_{f(\cdot;\boldsymbol{\theta})}(\boldsymbol{x})] \leq \langle\nabla_{\boldsymbol{\theta}}f(\mathrm{Prox}_{f(\cdot;\boldsymbol{\theta})}(\boldsymbol{x});\boldsymbol{\theta}),\boldsymbol{u}\rangle$. From equation 158 and by using Cauchy-Schwarz inequality, we compute the following bound on the inner product of the gradient with the unit vectors $\boldsymbol{u} \in \mathbb{R}^d, \|\boldsymbol{u}\|_2 = 1$,

$$\left\langle \nabla_{\boldsymbol{\theta}} M_{f(\cdot;\boldsymbol{\theta})}^{\beta}(\boldsymbol{x}), \boldsymbol{u} \right\rangle \leq \|\nabla_{\boldsymbol{\theta}}f(\mathrm{Prox}_{f(\cdot;\boldsymbol{\theta})}(\boldsymbol{x});\boldsymbol{\theta})\|_2 \cdot \|\boldsymbol{u}\|_2 = \|\nabla_{\boldsymbol{\theta}}f(\mathrm{Prox}_{f(\cdot;\boldsymbol{\theta})}(\boldsymbol{x});\boldsymbol{\theta})\|_2. \tag{163}$$

Since the preceding upper bound holds for all the unit vectors $\boldsymbol{u} \in \mathbb{R}^d$, we let $\boldsymbol{u} = \dfrac{\nabla_{\boldsymbol{\theta}} M_{f(\cdot;\boldsymbol{\theta})}^{\beta}(\boldsymbol{x})}{\|\nabla_{\boldsymbol{\theta}} M_{f(\cdot;\boldsymbol{\theta})}^{\beta}(\boldsymbol{x})\|_2}$ to get Inequality equation 65. ∎

## D.3  PROOF OF LEMMA C.6

Let $\boldsymbol{z} \in \mathrm{S}^{d-1}$ denotes an arbitrary vector on the sphere. Define the random variable

$$\begin{aligned}
Q_{\boldsymbol{z}}\Big((y_1,\boldsymbol{x}_1),\cdots,(y_n,\boldsymbol{x}_n)\Big) &\overset{\text{def}}{=} \langle\boldsymbol{z},\nabla E_n(\boldsymbol{\xi})\rangle \\
&= \frac{1}{n(n-1)}\sum_{i \neq j} y_i y_j \Big(\varphi(\boldsymbol{x}_i;\boldsymbol{\xi})\langle\boldsymbol{z},\nabla_{\boldsymbol{\xi}}\varphi(\boldsymbol{x}_j;\boldsymbol{\xi})\rangle + \varphi(\boldsymbol{x}_j;\boldsymbol{\xi})\langle\boldsymbol{z},\nabla_{\boldsymbol{\xi}}\varphi(\boldsymbol{x}_i;\boldsymbol{\xi})\rangle\Big) \\
&\quad - \mathbb{E}_{P_{\boldsymbol{x},y}^{\otimes 2}}\Big[y\widehat{y}\Big(\varphi(\boldsymbol{x}_i;\boldsymbol{\xi})\langle\boldsymbol{z},\nabla_{\boldsymbol{\xi}}\varphi(\boldsymbol{x}_j;\boldsymbol{\xi})\rangle + \varphi(\boldsymbol{x}_j;\boldsymbol{\xi})\langle\boldsymbol{z},\nabla_{\boldsymbol{\xi}}\varphi(\boldsymbol{x}_i;\boldsymbol{\xi})\rangle\Big)\Big].
\end{aligned} \tag{164}$$

Clearly, $\mathbb{E}_{P_{\boldsymbol{x},y}}[Q_{\boldsymbol{z}}] = 0$. Now, let $(\widehat{y}_m, \widehat{\boldsymbol{x}}_m) \in \mathcal{Y} \times \mathcal{X}, 1 \leq m \leq n$. By repeated application of the triangle inequality, we obtain that

$$
\left| Q_{\boldsymbol{z}}((y_1, \boldsymbol{x}_1), \cdots, (y_m, \boldsymbol{x}_m), \cdots, (y_n, \boldsymbol{x}_n)) - Q_{\boldsymbol{z}}((y_1, \boldsymbol{x}_1), \cdots, (\widehat{y}_m, \widehat{\boldsymbol{x}}_m), \cdots, (y_n, \boldsymbol{x}_n)) \right|
$$

$$
\leq \frac{1}{n(n-1)} \left| \sum_{i \neq m} y_i \varphi(\boldsymbol{x}_i; \boldsymbol{\xi}) \langle \boldsymbol{z}, y_m \nabla_{\boldsymbol{\xi}} \varphi(\boldsymbol{x}_m; \boldsymbol{\xi}) - \widehat{y}_m \nabla_{\boldsymbol{\xi}} \varphi(\widehat{\boldsymbol{x}}_m; \boldsymbol{\xi}) \rangle \right|
$$

$$
+ \frac{1}{n(n-1)} \left| \sum_{i \neq m} y_i \langle \boldsymbol{z}, \nabla_{\boldsymbol{\xi}} \varphi(\boldsymbol{x}_i; \boldsymbol{\xi}) \rangle (y_m \varphi(\boldsymbol{x}_m; \boldsymbol{\xi}) - \widehat{y}_m \varphi(\widehat{\boldsymbol{x}}_m; \boldsymbol{\xi})) \right|
$$

$$
\leq \frac{1}{n(n-1)} \sum_{i \neq m} |\varphi(\boldsymbol{x}_i; \boldsymbol{\xi})| \cdot \|\boldsymbol{z}\|_2 \cdot \|y_m \nabla_{\boldsymbol{\xi}} \varphi(\boldsymbol{x}_m; \boldsymbol{\xi}) - \widehat{y}_m \nabla_{\boldsymbol{\xi}} \varphi(\widehat{\boldsymbol{x}}_m; \boldsymbol{\xi})\|_2
$$

$$
+ \frac{1}{n(n-1)} \sum_{i \neq m} \|\boldsymbol{z}\|_2 \cdot \|\nabla_{\boldsymbol{\xi}} \varphi(\boldsymbol{x}_i; \boldsymbol{\xi})\|_2 \cdot |y_m \varphi(\boldsymbol{x}_m; \boldsymbol{\xi}) - \widehat{y}_m \varphi(\widehat{\boldsymbol{x}}_m; \boldsymbol{\xi})|
$$

$$
\leq \frac{4L^2}{n}, \tag{165}
$$

where the last inequality is due to assumption (**A.2**) and the fact that $\|\boldsymbol{z}\|_2 = 1$ for $\boldsymbol{z} \in \mathrm{S}^{d-1}$. In particular, to derive Inequality equation 165, we employed the following upper bounds

$$
|\varphi(\boldsymbol{x}_i; \boldsymbol{\xi})| \leq L,
$$
$$
|y_m \varphi(\boldsymbol{x}_m; \boldsymbol{\xi}) - \widehat{y}_m \varphi(\widehat{\boldsymbol{x}}_m; \boldsymbol{\xi})| \leq |\varphi(\boldsymbol{x}_m; \boldsymbol{\xi})| + |\varphi(\widehat{\boldsymbol{x}}_m; \boldsymbol{\xi})| \leq 2L,
$$
$$
\|\nabla_{\boldsymbol{\xi}} \varphi(\boldsymbol{x}_i; \boldsymbol{\xi})\|_2 \leq L,
$$
$$
\|y_m \nabla_{\boldsymbol{\xi}} \varphi(\boldsymbol{x}_m; \boldsymbol{\xi}) - \widehat{y}_m \nabla_{\boldsymbol{\xi}} \varphi(\widehat{\boldsymbol{x}}_m; \boldsymbol{\xi})\|_2 \leq \|\nabla_{\boldsymbol{\xi}} \varphi(\boldsymbol{x}_m; \boldsymbol{\xi})\|_2 + \|\nabla_{\boldsymbol{\xi}} \varphi(\widehat{\boldsymbol{x}}_m; \boldsymbol{\xi})\|_2 \leq 2L.
$$

Using McDiarmid Martingale's inequality McDiarmid (1989) then gives us

$$
\mathbb{P}\left( \left| Q_{\boldsymbol{z}}((y_1, \boldsymbol{x}_1), \cdots, (y_n, \boldsymbol{x}_n)) \right| \geq u \right) \leq 2\left( -\frac{nu^2}{16L^4} \right), \tag{166}
$$

for $x \geq 0$. Now, for every $p \in \mathbb{N}$, the $2p$-th moment of the random variable $Q_{\boldsymbol{z}}$ is given by

$$
\mathbb{E}\left[ Q_{\boldsymbol{z}}^{2p}((y_1, \boldsymbol{x}_1), \cdots, (y_n, \boldsymbol{x}_n)) \right] = \int_{\mathbb{R}_+} 2pu^{2p-1} \mathbb{P}(Q_{\boldsymbol{z}}((y_1, \boldsymbol{x}_1), \cdots, (y_n, \boldsymbol{x}_n)) \geq u)\mathrm{d}u
$$

$$
\overset{(a)}{\leq} \int_{\mathbb{R}_+} 4pu^{2p-1} \left( -\frac{u^2}{16nL^4} \right) \mathrm{d}u
$$

$$
= 2(16L^4/n)^{2p} p!, \tag{167}
$$

where (a) is due to the concentration bound in equation 166. Now Therefore,

$$
\mathbb{E}\left[ (Q_{\boldsymbol{z}}^2((y_1, \boldsymbol{x}_1), \cdots, (y_n, \boldsymbol{x}_n))/\gamma^2) \right] = \sum_{p=0}^{\infty} \frac{1}{p! \gamma^{2p}} \mathbb{E}\left[ \phi_{\boldsymbol{z}}^{2p}((y_1, \boldsymbol{x}_1), \cdots, (y_n, \boldsymbol{x}_n)) \right]
$$

$$
= 1 + 2 \sum_{p \in \mathbb{N}} \left( \frac{16L^4}{n\gamma} \right)^{2p}
$$

$$
= \frac{2}{1 - (16L^4/n\gamma)^2} - 1.
$$

For $\gamma = 16\sqrt{3}L^4/n$, we obtain $\mathbb{E}\left[ (Q_{\boldsymbol{z}}^2((y_1, \boldsymbol{x}_1), \cdots, (y_n, \boldsymbol{x}_n))/\gamma^2) \right] \leq 2$. Therefore, $\|Q_{\boldsymbol{z}}\|_{\psi_2} = \|\langle \boldsymbol{z}, \nabla E_n(\boldsymbol{\xi}) \rangle\|_{\psi_2} \leq 16\sqrt{3}L^4/n$ for all $\boldsymbol{z} \in \mathrm{S}^{n-1}$ and $\boldsymbol{\xi} \in \mathbb{R}^D$. Consequently, by the definition

of the sub-Gaussian random vector in equation 35 of Definition C.2, we have $\|\nabla E_n(\boldsymbol{\xi})\|_{\psi_2} \leq 16\sqrt{3}L^4/n$ for every $\boldsymbol{\xi} \in \mathbb{R}^D$. We invoke the following lemma proved by the first author in (Khuzani & Li, 2017, Lemma 16):

**Lemma D.1.** (THE ORLICZ NORM OF THE SQUARED VECTOR NORMS, (KHUZANI & LI, 2017, LEMMA 16)) *Consider the zero-mean random vector $\boldsymbol{Z}$ satisfying $\|\boldsymbol{Z}\|_{\psi_\nu} \leq \beta$ for every $\nu \geq 0$. Then, $\|\|\boldsymbol{Z}\|_2^2\|_{\psi_{\frac{\nu}{2}}} \leq 2 \cdot 3^{\frac{2}{\nu}} \cdot \beta^2$.*

Using Lemma D.1, we now have that $\|\|\nabla E_n(\boldsymbol{\xi})\|_2^2\|_{\psi_1} \leq 4608L^4/n^2$ for every $\boldsymbol{\xi} \in \mathbb{R}^D$. Applying the exponential Chebyshev's inequality with $\beta = 4608L^4/n^2$ yields

$$\mathbb{P}\left(\int_{\mathbb{R}^D}\int_0^1 \left|\|\nabla E_n((1-s)\boldsymbol{\xi}+s\boldsymbol{\zeta}_*)\|_2^2 - \mathbb{E}_{\boldsymbol{x},y}[\|\nabla E_n((1-s)\boldsymbol{\xi}+s\boldsymbol{\zeta}_*)\|_2^2]\right|\mu_0(\mathrm{d}\boldsymbol{\xi}) \geq \delta\right)$$

$$\leq e^{-\frac{n^2\delta}{4608L^4}}\mathbb{E}_{\boldsymbol{x},y}\left[e^{\left(\frac{n^2}{4608L^4}\int_{\mathbb{R}^D}\int_0^1\left|\|\nabla E_n((1-s)\boldsymbol{\xi}+s\boldsymbol{\zeta}_*)\|_2^2-\mathbb{E}_{\boldsymbol{x},y}[\|\nabla E_n((1-s)\boldsymbol{\xi}+s\boldsymbol{\zeta}_*)\|_2^2]\right|\mathrm{d}s\mu_0(\mathrm{d}\boldsymbol{\xi})\right)}\right]$$

$$\overset{(a)}{\leq} e^{-\frac{n^2\delta}{4608L^4}}\int_{\mathbb{R}^D}\int_0^1 \mathbb{E}_{\boldsymbol{x},y}\left[e^{\frac{n^2}{4608L^4}(|\|\nabla E_n((1-s)\boldsymbol{\xi}+s\boldsymbol{\zeta}_*)\|_2^2-\mathbb{E}_{\boldsymbol{x},y}[\|\nabla E_n((1-s)\boldsymbol{\xi}+s\boldsymbol{\zeta}_*)\|_2^2]|)}\right]\mathrm{d}s\mu_0(\mathrm{d}\boldsymbol{\xi})$$

$$\overset{(b)}{\leq} 2e^{-\frac{n^2\delta}{4608L^4}},$$

where $(a)$ follows by Jensen's inequality, and $(b)$ follows from the fact that

$$\mathbb{E}_{\boldsymbol{x},y}\left[e^{\frac{n^2}{4608L^4}(|\|\nabla E_n((1-s)\boldsymbol{\xi}+s\boldsymbol{\zeta}_*)\|_2^2-\mathbb{E}_{\boldsymbol{x},y}[\|\nabla E_n((1-s)\boldsymbol{\xi}+s\boldsymbol{\zeta}_*)\|_2^2])|}\right] \leq 2, \tag{168}$$

due to Definition C.1. Therefore,

$$\mathbb{P}\left(\int_{\mathbb{R}^D}\int_0^1 \|\nabla E_n((1-s)\boldsymbol{\xi}+s\boldsymbol{\zeta}_*)\|_2^2\mathrm{d}s\mu(\mathrm{d}\boldsymbol{\xi}) \geq \delta\right)$$

$$\leq 2\left(-\frac{n^2(\delta - \int_0^1\int_{\mathbb{R}^D}\mathbb{E}_{\boldsymbol{x},y}[\|\nabla E_n((1-s)\boldsymbol{\xi}+s\boldsymbol{\zeta}_*)\|_2^2]\mathrm{d}s\mu_0(\mathrm{d}\boldsymbol{\xi}))}{4608L^4}\right). \tag{169}$$

It now remains to compute an upper bound on the expectation $\mathbb{E}_{\boldsymbol{x},y}[\|\nabla E_n((1-s)\boldsymbol{\xi}+s\boldsymbol{\zeta}_*)\|_2^2]$. But this readily follows from equation 167 by letting $p=1$ and $\boldsymbol{z} = \frac{\nabla E_n((1-s)\boldsymbol{\xi}+s\boldsymbol{\zeta}_*)}{\|\nabla E_n((1-s)\boldsymbol{\xi}+s\boldsymbol{\zeta}_*)\|_2}$ as follows

$$\mathbb{E}_{\boldsymbol{x},y}[\|\nabla E_n((1-s)\boldsymbol{\xi}+s\boldsymbol{\zeta}_*)\|_2^2] = \mathbb{E}_{\boldsymbol{x},y}\left[\left\langle\frac{\nabla E_n((1-s)\boldsymbol{\xi}+s\boldsymbol{\zeta}_*)}{\|\nabla E_n((1-s)\boldsymbol{\xi}+s\boldsymbol{\zeta}_*)\|_2}, \nabla E_n((1-s)\boldsymbol{\xi}+s\boldsymbol{\zeta}_*)\right\rangle^2\right]$$

$$= \mathbb{E}_{\boldsymbol{x},y}\left[Q^2_{\frac{\nabla E_n((1-s)\boldsymbol{\xi}+s\boldsymbol{\zeta}_*)}{\|\nabla E_n((1-s)\boldsymbol{\xi}+s\boldsymbol{\zeta}_*)\|_2}}\right]$$

$$\leq 2^9\frac{L^8}{n^2}. \tag{170}$$

Plugging the expectation upper bound of equation 170 into equation 169 completes the proof of the first part of Lemma C.6.

The second part of Lemma C.6 follows by a similar approach and we thus omit its proof.

∎

### D.4 PROOF OF LEMMA C.7

Let $\boldsymbol{W}_* \stackrel{\text{def}}{=} \arg\min_{\boldsymbol{W} \in \mathcal{W}} \Psi(\boldsymbol{W})$ and $\boldsymbol{W}_\diamond \stackrel{\text{def}}{=} \arg\min_{\boldsymbol{W} \in \mathcal{W}} \Phi(\boldsymbol{W})$. Then, since $|\Psi(\boldsymbol{W}) - \Phi(\boldsymbol{W})| \leq \delta$ for all $\boldsymbol{W} \in \mathcal{W}$, we have that

$$|\Psi(\boldsymbol{W}_*) - \Phi(\boldsymbol{W}_*)| = \left| \min_{\boldsymbol{W} \in \mathcal{W}} \Psi(\boldsymbol{W}) - \Phi(\boldsymbol{W}_*) \right| \leq \delta. \tag{171}$$

Therefore,

$$\min_{\boldsymbol{W} \in \mathcal{W}} \Phi(\boldsymbol{W}) \leq \Phi(\boldsymbol{W}_*) \leq \min_{\boldsymbol{W} \in \mathcal{W}} \Psi(\boldsymbol{W}) + \delta. \tag{172}$$

Similarly, it can be shown that

$$\min_{\boldsymbol{W} \in \mathcal{W}} \Psi(\boldsymbol{W}) \leq \Psi(\boldsymbol{W}_\diamond) \leq \min_{\boldsymbol{W} \in \mathcal{W}} \Phi(\boldsymbol{W}) + \delta. \tag{173}$$

Combining equation 172 and equation 173 yields the desired inequality. ∎

### D.5 PROOF OF LEMMA C.10

We recall the expression of the remainder term $\{R_m^N\}_{0 \leq m \leq NT}$ from equation 84. For each $0 \leq m \leq NT$, $N \in \mathbb{N}$, we can bound the absolute value of the remainder term as follows

$$\left| R_m^N \right| = \frac{1}{N} \left| \sum_{k=1}^N (\boldsymbol{\xi}_{m+1}^k - \boldsymbol{\xi}_m^k) \nabla^2 f(\widetilde{\boldsymbol{\xi}}^k)(\boldsymbol{\xi}_{m+1}^k - \boldsymbol{\xi}_m^k)^T \right|$$

$$\leq \frac{1}{N} \sum_{k=1}^N \left| (\boldsymbol{\xi}_{m+1}^k - \boldsymbol{\xi}_m^k) \nabla^2 f(\widetilde{\boldsymbol{\xi}}^k)(\boldsymbol{\xi}_{m+1}^k - \boldsymbol{\xi}_m^k)^T \right|$$

$$\leq \frac{1}{N} \sum_{k=1}^N \|\boldsymbol{\xi}_{m+1}^k - \boldsymbol{\xi}_m^k\|_2^2 \cdot \left\| \nabla^2 f(\widetilde{\boldsymbol{\xi}}^k) \right\|_F. \tag{174}$$

Next, we characterize a bound on the difference term $\|\boldsymbol{\xi}_{m+1}^k - \boldsymbol{\xi}_m^k\|_2$. To attain this goal, we use the iterations of the particle SGD in Equation equation 15. We have that

$$\|\boldsymbol{\xi}_{m+1}^k - \boldsymbol{\xi}_m^k\|_2$$

$$\leq \frac{\eta}{N} \left\| \left( y_m \widetilde{y}_m - \frac{1}{N\alpha} \sum_{k=1}^N \varphi(\boldsymbol{x}_m; \boldsymbol{\xi}_m^k) \varphi(\widetilde{\boldsymbol{x}}_m; \boldsymbol{\xi}_m^k) \right) \nabla_{\boldsymbol{\xi}} \left( \varphi(\boldsymbol{x}_m; \boldsymbol{\xi}_m^k) \varphi(\widetilde{\boldsymbol{x}}_m; \boldsymbol{\xi}_m^k) \right) \right\|_2$$

$$\leq \frac{\eta}{N} |y_m \widetilde{y}_m| \left( |\varphi(\boldsymbol{x}_m; \boldsymbol{\xi}_m^k)| \cdot \|\nabla_{\boldsymbol{\xi}} \varphi(\widetilde{\boldsymbol{x}}_m; \boldsymbol{\xi}_m^k)\|_2 + |\varphi(\widetilde{\boldsymbol{x}}_m; \boldsymbol{\xi}_m^k)| \cdot \|\nabla_{\boldsymbol{\xi}} \varphi(\boldsymbol{x}_m; \boldsymbol{\xi}_m^k)\|_2 \right)$$

$$+ \frac{\eta}{N} \left( \frac{1}{N\alpha} \sum_{k=1}^N \left| \varphi(\boldsymbol{x}_m; \boldsymbol{\xi}_m^k) \varphi(\widetilde{\boldsymbol{x}}_m; \boldsymbol{\xi}_m^k) \right| \right) \left( |\varphi(\boldsymbol{x}_m; \boldsymbol{\xi}_m^k)| \|\nabla_{\boldsymbol{\xi}} \varphi(\widetilde{\boldsymbol{x}}_m; \boldsymbol{\xi}_m^k)\|_2 + |\varphi(\widetilde{\boldsymbol{x}}_m; \boldsymbol{\xi}_m^k)| \|\nabla_{\boldsymbol{\xi}} \varphi(\boldsymbol{x}_m; \boldsymbol{\xi}_m^k)\|_2 \right)$$

$$\stackrel{\text{(a)}}{\leq} \frac{\eta L^2}{N} + \frac{2\eta L^4}{N\alpha}, \tag{175}$$

where in (a), we used the fact that $\|\varphi\|_\infty < L$ and $\|\nabla_{\boldsymbol{\xi}} \varphi(\boldsymbol{x}, \boldsymbol{\xi})\|_2 < L$ due to (A.1), and $y_m, \widetilde{y}_m \in \{-1, 1\}$. Plugging the last inequality in equation 175 yields

$$|R_m^N| \leq \frac{1}{N^3} \left( \eta L^2 + \frac{2\eta L^4}{\alpha} \right) \sum_{k=1}^N \|\nabla^2 f(\widetilde{\boldsymbol{\xi}}^k)\|_F. \tag{176}$$

We next compute an upper bound on the Frobenious norm of the Hessian matrix $\nabla^2 f(\widetilde{\boldsymbol{\xi}}^k)$. To this end, we first show that there exists a compact set $\mathcal{C} \subset \mathbb{R}^D$ such that $\boldsymbol{\xi}_m^k \in \mathcal{C}$ for all $k = 1, 2, \cdots, N$ and all $m \in [0, NT] \cap \mathbb{N}$. For each $k = 1, 2, \cdots, N$, from Inequality equation 175 we obtain that

$$
\begin{aligned}
\|\boldsymbol{\xi}_m^k\|_2 &\leq \|\boldsymbol{\xi}_{m-1}^k\|_2 + \frac{\eta L^2}{N} + \frac{2\eta L^4}{N\alpha} \\
&= \|\boldsymbol{\xi}_0^k\|_2 + \frac{m\eta L^2}{N} + \frac{2m\eta L^4}{N\alpha} \\
&\leq \|\boldsymbol{\xi}_0^k\|_2 + \eta L^2 T + 2(\eta/\alpha)L^4 T.
\end{aligned}
\tag{177}
$$

Now, $\|\boldsymbol{\xi}_0^k\|_2 < c_0$ for some constant $c_0 > 0$ since the initial samples $\boldsymbol{\xi}_0^1, \cdots, \boldsymbol{\xi}_0^N$ are drawn from the measure $\mu_0$ whose support $\mathrm{support}(\mu_0) = \Xi$ is assumed to be compact by **(A.3)**. From upper bound in equation 177, it thus follows that $\|\boldsymbol{\xi}_m^k\|_2 < C$ for some constant $C > 0$, for all $m \in [0, NT] \cap \mathbb{N}$. Now, recall that $\widetilde{\boldsymbol{\xi}}^k = (\widetilde{\xi}^k(1), \cdots, \widetilde{\xi}^k(p))$, where $\widetilde{\xi}^k(i) \in [\xi_m^k(i), \xi_{m+1}^k(i)], i = 1, 2, \cdots, m+1$, for $i = 1, 2, \cdots, p$. Therefore, $\widetilde{\boldsymbol{\xi}}^k \in \mathcal{C}$. Since all the test function $f \in C_b^3(\mathbb{R}^3)$ are three-times continuously differentiable, it follows that there exists a constant $C_0 \stackrel{\text{def}}{=} C_0(T) > 0$ such that $\sup_{\widetilde{\boldsymbol{\xi}} \in \mathcal{C}} \|\nabla^2 f(\widetilde{\boldsymbol{\xi}})\|_F < C_0$. From Inequality equation 176, it follows that

$$
|R_m^N| \leq \frac{C_0}{N^2} \left( \eta L^2 + \frac{2\eta L^4}{\alpha} \right), \quad m \in [0, NT] \cap \mathbb{N}.
\tag{178}
$$

Now, recall the definition of the scaled term $\mathcal{R}_t^N$ from equation 89c. Using the Inequality equation 178 as well as the definition of $\mathcal{R}_t^N$, we obtain

$$
\sup_{0 \leq t \leq T} |\mathcal{R}_t^N| \leq \frac{C_0 T}{N} \left( \eta L^2 + \frac{2\eta L^4}{\alpha} \right).
\tag{179}
$$

$\blacksquare$

### D.6 PROOF OF LEMMA C.10

Let $\mathcal{F}_{m-1} = \sigma((\boldsymbol{x}_k, y_k)_{0 \leq k \leq m-1}, (\widetilde{\boldsymbol{x}}_k, \widetilde{y}_k)_{0 \leq k \leq m-1})$ denotes the $\sigma$-algebra generated by the samples up to time $m-1$. We define $\mathcal{F}_{-1} \stackrel{\text{def}}{=} \emptyset$. Further, define the following random variable

$$
\Delta_m^N \stackrel{\text{def}}{=} \left( \langle \varphi(\boldsymbol{x}_m, \boldsymbol{\xi})\varphi(\widetilde{\boldsymbol{x}}_m, \boldsymbol{\xi}), \widehat{\mu}_m^N \rangle - \alpha y_m \widetilde{y}_m \right) \times \langle \nabla f(\boldsymbol{\xi})\nabla_{\boldsymbol{\xi}}(\varphi(\widetilde{\boldsymbol{x}}_m; \boldsymbol{\xi})\varphi(\boldsymbol{x}_m; \boldsymbol{\xi})), \widehat{\mu}_m^N \rangle.
\tag{180}
$$

Notice that $\frac{1}{N}\mathbb{E}[\Delta_m^N | \mathcal{F}_{m-1}] = D_m^N$. We now rewrite the martingale term in equation 86b in term of $\Delta_m^N$,

$$
M_m^N \stackrel{\text{def}}{=} \frac{\eta}{N\alpha} \sum_{\ell=0}^m (\Delta_\ell^N - \mathbb{E}[\Delta_\ell^N | \mathcal{F}_{\ell-1}]),
\tag{181}
$$

with $M_0^N = 0$.

By construction of $M_m^N$ in equation 181, it is a Martingale $\mathbb{E}[M_m^N | \mathcal{F}_{m-1}] = M_{m-1}^N$. We now prove that $M_m^N$ has also bounded difference. To do so, we define the shorthand notations

$$
a_m^N \stackrel{\text{def}}{=} \langle \varphi(\boldsymbol{x}_m, \boldsymbol{\xi})\varphi(\widetilde{\boldsymbol{x}}_m, \boldsymbol{\xi}), \widehat{\mu}_m^N \rangle - \alpha y_m \widetilde{y}_m,
\tag{182}
$$

$$
b_m^N \stackrel{\text{def}}{=} \langle \nabla f(\boldsymbol{\xi})(\nabla_{\boldsymbol{\xi}}(\varphi(\widetilde{\boldsymbol{x}}_m; \boldsymbol{\xi})\varphi(\boldsymbol{x}_m; \boldsymbol{\xi})))^T, \widehat{\mu}_m^N \rangle.
\tag{183}
$$

Then, we compute

$$
\begin{aligned}
|M_m^N - M_{m-1}^N| &= \frac{\eta}{N\alpha} \left| \Delta_m^N - \mathbb{E}[\Delta_m^N | \mathcal{F}_{m-1}] \right| \\
&\leq \frac{\eta}{N\alpha} |\Delta_m^N| + \frac{\eta}{N\alpha} \mathbb{E}[|\Delta_m^N||\mathcal{F}_{m-1}] \\
&\leq \frac{\eta}{N\alpha} |a_m^N| \cdot |b_m^N| + \frac{\eta}{N\alpha} \mathbb{E}\left[|a_m^N| \cdot |b_m^N||\mathcal{F}_{m-1}\right].
\end{aligned}
\tag{184}
$$

For the difference terms, we derive that

$$
\begin{aligned}
|a_m^N| &= \left| \langle \varphi(\boldsymbol{x}_m, \boldsymbol{\xi})\varphi(\widetilde{\boldsymbol{x}}_m, \boldsymbol{\xi}), \widehat{\mu}_m^N \rangle - \alpha y_m \widetilde{y}_m \right| \\
&\leq \frac{1}{N} \sum_{k=1}^N \left| \varphi(\boldsymbol{x}_m, \boldsymbol{\xi}_m^k)\varphi(\widetilde{\boldsymbol{x}}_m, \boldsymbol{\xi}_m^k) \right| + \alpha |y_m \widetilde{y}_m| \\
&\leq L^2 + \alpha,
\end{aligned}
\tag{185}
$$

where the last step follows from the fact that $\|\varphi\|_\infty \leq L$ due to **(A.1)**. Similarly, we obtain that

$$
\begin{aligned}
|b_m^N| &= |\langle \nabla f(\boldsymbol{\xi})(\nabla_{\boldsymbol{\xi}}(\varphi(\widetilde{\boldsymbol{x}}_m; \boldsymbol{\xi})\varphi(\boldsymbol{x}_m; \boldsymbol{\xi})))^T, \widehat{\mu}_m^N \rangle| \\
&\leq \frac{1}{N} \sum_{k=1}^N |\varphi(\widetilde{\boldsymbol{x}}_m; \boldsymbol{\xi}_m^k)| \cdot \left| \nabla f(\boldsymbol{\xi}_m^k)(\nabla_{\boldsymbol{\xi}}\varphi(\widetilde{\boldsymbol{x}}_m; \boldsymbol{\xi}_m^k))^T \right| \\
&\quad + \frac{1}{N} \sum_{k=1}^N |\varphi(\boldsymbol{x}_m; \boldsymbol{\xi}_m^k)| \cdot \left| \nabla f(\boldsymbol{\xi}_m^k)(\nabla_{\boldsymbol{\xi}}\varphi(\widetilde{\boldsymbol{x}}_m; \boldsymbol{\xi}_m^k))^T \right| \\
&\stackrel{(a)}{\leq} \frac{L}{N} \sum_{k=1}^N \left| \nabla f(\boldsymbol{\xi}_m^k)(\nabla_{\boldsymbol{\xi}}\varphi(\widetilde{\boldsymbol{x}}_m; \boldsymbol{\xi}_m^k))^T \right| + \frac{L}{N} \sum_{k=1}^N \left| \nabla f(\boldsymbol{\xi}_m^k)(\nabla_{\boldsymbol{\xi}}\varphi(\widetilde{\boldsymbol{x}}_m; \boldsymbol{\xi}_m^k))^T \right| \\
&\stackrel{(b)}{\leq} \frac{L}{N} \sum_{k=1}^N \|\nabla f(\boldsymbol{\xi}_m^k)\|_2 \cdot \|\nabla_{\boldsymbol{\xi}}\varphi(\widetilde{\boldsymbol{x}}_m; \boldsymbol{\xi}_m^k)\|_2 + \frac{L}{N} \sum_{k=1}^N \|\nabla f(\boldsymbol{\xi}_m^k)\|_2 \cdot \|\nabla_{\boldsymbol{\xi}}\varphi(\widetilde{\boldsymbol{x}}_m; \boldsymbol{\xi}_m^k)\|_2 \\
&\stackrel{(c)}{\leq} \frac{2L^2}{N} \sum_{k=1}^N \|\nabla f(\boldsymbol{\xi}_m^k)\|_2,
\end{aligned}
\tag{186}
$$

where (a) and (c) follows from (**A**.1), and (b) follows from the Cauchy-Schwarz inequality. From Inequality equation 177 and the ensuing disucssion in Appendix D.5, we recall that $\|\boldsymbol{\xi}_m^k\|_2 < C$ for some constant and for all $m \in [0, NT] \cap \mathbb{N}$, and $k = 1, 2, \cdots, N$. For the two times continuously test function $f \in C_b^3(\mathbb{R}^D)$, it then follows that $|\nabla f(\boldsymbol{\xi}_m^k)\|_2 \leq C_1$ for some constant $C_1 > 0$. The following bound can now be computed from equation 186,

$$
|b_m^N| \leq \frac{2C_1 L^2}{N}.
\tag{187}
$$

Plugging the upper bounds on $|a_m^N|$ and $|b_m^N|$ from equation 185-equation 187 into equation 184 we obtain that

$$
|M_m^N - M_{m-1}^N| \leq \frac{4\eta C_1}{N\alpha} L^2 (L^2 + \alpha).
\tag{188}
$$

Thus, $(M_m^N)_{m \in [0, NT] \cap \mathbb{N}}$ is a Martingale process with bounded difference. From the Azuma-Hoeffding inequality it follows that

$$
\mathbb{P}(|M_m^N| \geq \varepsilon) = \mathbb{P}(|M_m^N - M_0^N| \geq \varepsilon) \leq 2 \left( -\frac{N^2 \alpha^2 \varepsilon^2}{8m L^4 \eta^2 C_1^2 (L^2 + \alpha)^2} \right), \quad \forall m \in [0, NT] \cap \mathbb{N}.
\tag{189}
$$

Therefore, since $\mathcal{M}_t^N = M_{\lfloor Nt \rfloor}^N$, we have

$$\mathbb{P}(|\mathcal{M}_T^N| \geq \varepsilon) \leq 2 \left( -\frac{N^2 \alpha^2 \varepsilon^2}{8L^4 \lfloor NT \rfloor \eta^2 C_1^2 (L^2 + \alpha)^2} \right). \tag{190}$$

Then,

$$\begin{aligned}
\mathbb{E}\left[ |\mathcal{M}_T^N| \right] &= \int_0^\infty \mathbb{P}(|\mathcal{M}_T^N| \geq \varepsilon) \mathrm{d}\varepsilon \\
&\leq 2 \int_0^\infty \left( -\frac{N^2 \alpha^2 \varepsilon^2}{8L^4 \lfloor NT \rfloor \eta^2 C_1^2 (L^2 + \alpha)^2} \right) \\
&= \frac{1}{N\alpha} 4\sqrt{2} L^2 \sqrt{\lfloor NT \rfloor} \eta C_1 (L^2 + \alpha)^2.
\end{aligned} \tag{191}$$

where the inequality follows from equation 190.

By Doob's Martingale inequality Doob (1953), the following inequality holds

$$\mathbb{P}\left( \sup_{0 \leq t \leq T} |\mathcal{M}_t^N| \geq \varepsilon \right) \leq \frac{\mathbb{E}[|\mathcal{M}_T^N|]}{\varepsilon} \tag{192}$$

$$\leq \frac{1}{N\alpha\varepsilon} 4\sqrt{2} L^2 \sqrt{\lfloor NT \rfloor} \eta C_1 (L^2 + \alpha)^2. \tag{193}$$

In particular, with the probability of at least $1 - \rho$, we have

$$\sup_{0 \leq t \leq T} |\mathcal{M}_t^N| \leq \frac{1}{N\alpha\rho} 4\sqrt{2} L^2 \sqrt{\lfloor NT \rfloor} \eta C_1 (L^2 + \alpha)^2. \tag{194}$$

$\blacksquare$

## E  CHAOTICITY AND PROPAGATION OF CHAOS IN PARTICLE SGD

In this appendix, we establish the so called 'propagation of chaos' property of particle SGD. We now establish the so called 'propagation of chaos' property of particle SGD. At a high level, the propagation of chaos means that when the number of samples $\{\boldsymbol{\xi}^k\}_{k=1}^N$ tends to infinity ($N \to +\infty$), their dynamics are decoupled.

**Definition E.1.** (EXCHANGABLITY) Let $\nu$ be a probability measure on a Polish space $\mathcal{S}$ and. For $N \in \mathbb{N}$, we say that $\nu^{\otimes N}$ is an exchangeable probability measure on the product space $\mathcal{S}^n$ if it is invariant under the permutation $\boldsymbol{\pi} \stackrel{\text{def}}{=} (\pi(1), \cdots, \pi(N))$ of indices. In particular,

$$\nu^{\otimes N}(\boldsymbol{\pi} \cdot B) = \nu^{\otimes N}(B), \tag{195}$$

for all Borel subsets $B \in \mathcal{B}(\mathcal{S}^n)$.

An interpretation of the exchangablity condition equation 195 can be provided via De Finetti's representation theorem which states that the joint distribution of an infinitely exchangeable sequence of random variables is as if a random parameter were drawn from some distribution and then the random variables in question were independent and identically distributed, conditioned on that parameter.

Next, we review the mathematical definition of chaoticity, as well as the propagation of chaos in the product measure spaces:

**Definition E.2.** (CHAOTICITY) Suppose $\nu^{\otimes N}$ is exchangeable. Then, the sequence $\{\nu^{\otimes N}\}_{N \in \mathbb{N}}$ is $\nu$-chaotic if, for any natural number $\ell \in \mathbb{N}$ and any test function $f_1, f_2, \cdots, f_k \in C_b^2(\mathcal{S})$, we have

$$\lim_{N \to \infty} \left\langle \prod_{k=1}^\ell f_k(s^k), \nu^{\otimes N}(\mathrm{d}s^1, \cdots, \mathrm{d}s^N) \right\rangle = \prod_{k=1}^\ell \langle f_k, \nu \rangle \tag{196}$$

According to equation 196 of Definition E.2, a sequence of probability measures on the product spaces $\mathcal{S}$ is $\nu$-chaotic if, for fixed $k$ the joint probability measures for the first $k$ coordinates tend to the product measure $\nu(\mathrm{d}s_1)\nu(\mathrm{d}s_2)\cdots\nu(\mathrm{d}s_k) = \nu^{\otimes k}$ on $\mathcal{S}^k$. If the measures $\nu^{\otimes N}$ are thought of as giving the joint distribution of $N$ particles residing in the space $\mathcal{S}$, then $\{\nu^{\otimes N}\}$ is $\nu$-chaotic if $k$ particles out of $N$ become more and more independent as $N$ tends to infinity, and each particles distribution tends to $\nu$. A sequence of symmetric probability measures on $\mathcal{S}^N$ is chaotic if it is $\nu$-chaotic for some probability measure $\nu$ on $\mathcal{S}$.

If a Markov process on $\mathcal{S}^N$ begins in a random state with the distribution $\nu^{\otimes N}$, the distribution of the state after $t$ seconds of Markovian random motion can be expressed in terms of the transition function $\mathcal{K}^N$ for the Markov process. The distribution at time $t > 0$ is the probability measure $U_t^N \nu^{\otimes N}$ is defined by the kernel

$$U_t^N \nu^{\otimes N}(B) \overset{\text{def}}{=} \int_{\mathcal{S}^N} \mathcal{K}^N(s, B, t)\nu^{\otimes N}(\mathrm{d}s). \tag{197}$$

**Definition E.3.** (PROPAGATION OF CHAOS) A sequence functions

$$\left\{\mathcal{K}^N(s, B, t)\right\}_{N \in \mathbb{N}} \tag{198}$$

whose $N$-th term is a Markov transition function on $\mathcal{S}^N$ that satisfies the permutation condition

$$\mathcal{K}^N(s, B, t) = \mathcal{K}^N(\boldsymbol{\pi} \cdot s, \boldsymbol{\pi} \cdot B, t), \tag{199}$$

propagates chaos if whenever $\{\nu^{\otimes N}\}_{N \in \mathbb{N}}$ is chaotic, so is $\{U_t^N\}$ for any $t \geq 0$, where $U_t^N$ is defined in equation 197.

We note that for finite systems size $N$, the states of the particles are not independent of each other. However, as we prove in the following result, in the limiting system $N \to +\infty$, the particles are mutually independent. This phenomena is known as the propagation of chaos (*a.k.a.* asymptotic independence):

**Theorem E.4.** (CHAOTICITY IN PARTICLE SGD) *Consider Assumptions* $(\mathbf{A}.1) - (\mathbf{A}.3)$*. Furthermore, suppose that* $\{\boldsymbol{\xi}_0^k\}_{1 \leq k \leq N} \sim_{i.i.d.} \mu_0$ *is exchangable in the sense that the joint law is invariant under the permutation of indices. Then, at each time instant* $t \in (0, T]$*, the scaled empirical measure* $\mu_t^N \in \mathcal{M}(\mathbb{R}^D)$ *defined via scaling*

$$\mu_t^N(\mathrm{d}\boldsymbol{\xi}^1, \cdots, \mathrm{d}\boldsymbol{\xi}^N) \overset{\text{def}}{=} \widehat{\mu}_{\lfloor Nt \rfloor}^N(\mathrm{d}\boldsymbol{\xi}^1, \cdots, \mathrm{d}\boldsymbol{\xi}^N) = \mathbb{P}\{\boldsymbol{\xi}_{\lfloor Nt \rfloor}^1 \in \mathrm{d}\boldsymbol{\xi}^1, \cdots, \boldsymbol{\xi}_{\lfloor Nt \rfloor}^N \in \mathrm{d}\boldsymbol{\xi}^N\}, \tag{200}$$

*is* $\mu_t^*$*-chaotic, where* $\mu_t^*$ *is mean-field solution of equation 97.*

**Proof.** To establish the proof, it suffices to show that for every integer $\ell \in \mathbb{N}$, and for all the test functions $f_1, \cdots, f_k \in C_b^3(\mathbb{R}^D)$, we have

$$\lim_{N \to \infty} \sup \left| \mathbb{E}\left[\prod_{k=1}^{\ell} f_k(\boldsymbol{\xi}_{\lfloor Nt \rfloor}^k)\right] - \prod_{k=1}^{\ell} \langle \mu_t^*, f_k \rangle \right| = 0. \tag{201}$$

Using the triangle inequality, we now have that

$$\left| \mathbb{E}\left[\prod_{k=1}^{\ell} f_k(\boldsymbol{\xi}_{\lfloor Nt \rfloor}^k)\right] - \prod_{k=1}^{\ell} \langle \mu_t^*, f_k \rangle \right|$$

$$\leq \left| \mathbb{E}\left[\prod_{k=1}^{\ell} \langle \widehat{\mu}_t^N, f_k \rangle\right] - \prod_{k=1}^{\ell} \langle \mu_t^*, f_k \rangle \right| + \left| \mathbb{E}\left[\prod_{k=1}^{\ell} \langle \widehat{\mu}_t^N, f_k \rangle\right] - \mathbb{E}\left[\prod_{k=1}^{\ell} f_k(\boldsymbol{\xi}_{\lfloor Nt \rfloor}^k)\right] \right|. \tag{202}$$

For the first term on the right side of equation 202 we have

$$
\limsup_{N\to\infty} \left| \mathbb{E}\left[ \prod_{k=1}^{\ell} \langle \widehat{\mu}_t^N, f_k \rangle \right] - \prod_{k=1}^{\ell} \langle \mu_t^*, f_k \rangle \right| \overset{(a)}{\leq} \limsup_{N\to\infty} \mathbb{E}\left[ \left| \prod_{k=1}^{\ell} \langle \widehat{\mu}_t^N, f_k \rangle - \prod_{k=1}^{\ell} \langle \mu_t^*, f_k \rangle \right| \right]
$$

$$
\overset{(b)}{\leq} \mathbb{E}\left[ \limsup_{N\to\infty} \left| \prod_{k=1}^{\ell} \langle \widehat{\mu}_t^N, f_k \rangle - \prod_{k=1}^{\ell} \langle \mu_t^*, f_k \rangle \right| \right]
$$

$$
\overset{(c)}{\leq} b^{\ell-1} \mathbb{E}\left[ \sum_{k=1}^{\ell} \limsup_{N\to\infty} \left| \langle \widehat{\mu}_t^N, f_k \rangle - \langle \mu_t^*, f_k \rangle \right| \right]
$$

$$
\overset{(d)}{=} 0, \tag{203}
$$

where (a) is by Jensen's inequality, (b) is by Fatou's lemma, (c) follows from the basic inequality $\left| \prod_{i=1}^N a_i - \prod_{i=1}^N b_i \right| \leq \sum_{i=1}^N |a_i - b_i|$ for $|a_i|, |b_i| \leq 1$, $i = 1, 2, \cdots, N$, as well as the fact that $\langle \mu_t^*, f_k \rangle \leq b$ and $\langle \widehat{\mu}_t^N, f_k \rangle \leq b$ for all $k = 1, 2, \cdots, N$ due to the boundedness of the test functions $f_1, \cdots, f_\ell \in C_b^3(\mathbb{R}^D)$, and (d) follows from the weak convergence $\widehat{\mu}_t^N \overset{\text{weakly}}{\to} \mu_t^*$ to the mean-field solution equation 97.

Now, consider the second term on the right hand side of equation 202. Due to the exchangeability of the initial states $(\boldsymbol{\xi}_0^k)_{1 \leq k \leq N}$, the law of the random variables $(\boldsymbol{\xi}_0^k)_{1 \leq k \leq N}$ is also exchangeable. Therefore, we obtain that

$$
\mathbb{E}\left[ \prod_{k=1}^{\ell} f_k(\boldsymbol{\xi}_{\lfloor Nt \rfloor}^k) \right] = \frac{\ell!}{N!} \mathbb{E}\left[ \sum_{\pi \in \Pi(\ell, N)} \prod_{k=1}^{\ell} f_k(\boldsymbol{\xi}_{\lfloor Nt \rfloor}^{\pi(k)}) \right], \tag{204}
$$

where $\Pi(\ell, N)$ is the set of all permutations of $\ell$ numbers selected from $\{1, 2, \cdots, N\}$. Notice that the right hand side of equation 204 is the symmetrized version of the left hand side equation 205.

Further, by the definition of the empirical measure $\widehat{\mu}_t^N$ we obtain that

$$
\mathbb{E}\left[ \prod_{k=1}^{\ell} \langle \widehat{\mu}_t^N, f_k \rangle \right] = \frac{1}{N^\ell} \mathbb{E}\left[ \prod_{k=1}^{\ell} \left( \sum_{m=1}^N f_k(\boldsymbol{\xi}_{\lfloor Nt \rfloor}^m) \right) \right] \tag{205}
$$

$$
= \frac{1}{N^\ell} \mathbb{E}\left[ \sum_{\pi \in \widetilde{\Pi}(\ell, N)} \left( \prod_{k=1}^{\ell} f_k(\boldsymbol{\xi}_{\lfloor Nt \rfloor}^{\pi(k)}) \right) \right]. \tag{206}
$$

Therefore, subtracting equation 204 and equation 205 yields

$$
\left| \mathbb{E}\left[ \prod_{k=1}^{\ell} \langle \widehat{\mu}_t^N, f_k \rangle \right] - \mathbb{E}\left[ \prod_{k=1}^{\ell} f_k(\boldsymbol{\xi}_{\lfloor Nt \rfloor}^k) \right] \right| \leq b^\ell \left( 1 - \frac{N!}{\ell! N^\ell} \right). \tag{207}
$$

Hence,

$$
\limsup_{N\to\infty} \left| \mathbb{E}\left[ \prod_{k=1}^{\ell} \langle \widehat{\mu}_t^N, f_k \rangle \right] - \mathbb{E}\left[ \prod_{k=1}^{\ell} f_k(\boldsymbol{\xi}_{\lfloor Nt \rfloor}^k) \right] \right| = 0. \tag{208}
$$

Combining equation 203-equation 208 yields the desired result. ∎

