# OpenReview forum: "A Mean-Field Theory for Kernel Alignment with Random Features in Generative Adverserial Networks"
_ICLR.cc/2020/Conference — Reject_

### Official Review · AnonReviewer2 · 2019-10-22
**Official Blind Review #2**

**Rating:** 1

**Review:**

This paper proposes to learn a kernel for training MMD-GAN by optimizing over the probability distribution that defines the kernel by means of random features. This is unlike the usual setting of MMD-GAN where the kernel is parametrized by composing a fixed top-kernel with a discriminator network that is optimized during training. The main motivation for this approach is to avoid having to 'manually' fix some parameters of the top-level kernel like the bandwidth of the RBF kernel. They provide an algorithm to achieve such optimization in probability space along with some consistency results and perform experiments on MNIST and Cifar10 to demonstrate empirically the advantage of such an approach over those that fix the top-level kernel.

Theory:
	Theorem 4.1 provides a convergence result of an oracle finite-sample estimator: that is the one obtained by exactly solving the optimization problem in 19b. In that case, they show the consistency of the proposed estimator. The result is somehow expected but the proof relies on nice duality results for measures and is very technical.
The clarity of the proof could be improved:
- Currently, the structure of the main proof mixes direct lemma (lemma B.7) with less obvious ones (lemma B.6). Also, some concepts are introduced in the main proof but not necessary for its understanding: The notion of Orlicz norm is in introduced in Definition B.3 on the fly to state lemma B.6, but only equations 50 and 51 are used in this lemma which does not make use of the notion of Orlicz norm at all.

	Theorem 4.1 doesn't say anything about the consistency of the algorithm itself. To partly address this, the authors show in theorem 4.2 that as the number of particles grows the empirical process converges to a McKean Vlasov PDE (equation 22). This means that the proposed algorithm is approximating some gradient flow in metric space (25).
- However, this gradient flow is a non-convex optimization problem and there is no guarantee that a global solution is reached.  Recent work provides cases when global convergence occurs [Chizat2018] but it is not the case in general. Some further clarification about the connection between Thm 4.1 and Thm 4.2 would be therefore useful.

Thm 4.2 is also curious in the sense that the process defined by equation 16, which is noisy since it relies on one sample from the data, would converge towards (22) which is a Mc-valsov equation with a drift only (no diffusion or other noise).  What happened to the noise coming from sampling from P_v  and P_w in equation 16? Wouldn't there be some sort of diffusion term as in [Hu2018]?

More generally it would be nice to have a discussion of the assumptions and results in the paper as they seem to rely on methods that people in the machine learning community are not totally familiar with.

Experiments:  The experiments are not convincing for several reasons:
 - The comparison with the other methods is somehow unfair since the bandwidth is manually tuned for the competing methods. It is easy to adaptively learn the bandwidth as well:  in this case, it will be just an additional parameter of a discriminator network. This was done in [Arbel2018] where a single gaussian kernel is used and a regularization of the critic allows to learn the bandwidth without manual tuning. Does the proposed method offer an additional advantage compared to those?
- In practice and for a scaling parameter alpha=1, isn't the algorithm strictly equivalent to considering an MMD-GAN with a dot product kernel and a discriminator given by the feature \phi(x,\zeta)?
- Mnist and Cifar10 are somehow very simple, what would happen on more complicated datasets (CelebA or imagenet)?

- I also think there is an ablation that is missing: If the auto-encoder also needs to be optimized then does it also help to optimize over the particles as well or is optimizing the auto-encoder discriminator enough to achieve a similar performance? In other words, does optimizing the auto-encoder compensate for the need to learn the distribution mu? Of course, this would depend on how the auto-encoder is parametrized but I don't see why it wouldn't in many cases.


Overall, I'm not convinced that the proposed approach would lead to any substantial improvement for MMD-GAN in practice, and the experiments are not really convincing as they are now. However, I find the theoretical results interesting and might be used to better analyse the dynamics of GAN's. But as the paper is currently framed, it is hard to put these theoretical results in perspective.


**Experience Assessment:**

I have published one or two papers in this area.

**Review Assessment: Checking Correctness Of Derivations And Theory:**

I assessed the sensibility of the derivations and theory.

**Review Assessment: Checking Correctness Of Experiments:**

I carefully checked the experiments.

**Review Assessment: Thoroughness In Paper Reading:**

I read the paper at least twice and used my best judgement in assessing the paper.

---

> ### Author Response · Authors · 2019-11-15
> **More experiments are added.**
>
> We appreciate the reviewer for careful read of our manuscript, and encouraging comments regarding the theoretical results. We believe that the main merit of our proposed method is its simplicity as the implementation of the SGD algorithm is relatively easy compared to other kernel learning methods. We address your comments regarding the theoretical and experimental results below:
>
> Theory:
> -The notion of the Orlicz norm is indeed used to state Lemma C1. We have moved these definitions to the notation section of the Supplementary instead of presenting them on the fly.
> -We appreciate the reviewer for bringing the interesting work of [Chizat2018] to our attention. We indeed agree with the reviewer that the distributional optimization is a non-convex optimization and at this moment, we do not have a guarantee that the Wasserstein gradient flow converges to the global solution of the distributional optimization in Eq. (11). However, as the authors of [Chizat2018] have shown, such conditions can be found.
>
> With regard to Thm. 4.1. and Thm. 4.2., we would like to provide more clarification about their connections. We notice that Thm 4.1. is a general statement about the optimal solution of the population MMD optimization in Eq. (8), and the optimization of its empirical estimator in Equation (13) in the revised draft. This result is not concerned with any particular optimization algorithm.  In contrast, Thm 4.2., deals with a specific optimization algorithm, namely SGD. Although SGD is solving a non-convex optimization problem, and the resulting solution from this algorithm is not necessarily a global solution, Thm 4.2. ensures that when the number of particles $N\rightarrow\infty$ tend to infinity, the resulting empirical measure from SGD solution is a local minima for the distributional optimization problem in Eq. (11).
>
> - Thank you for this interesting comment. The authors is correct that there is a stochastic term involved due to the random sampling in particle SGD. This stochastic term is precisely the Martingale process defined in Equation 87b. However, as shown in Lemma B.10, the absolute value of this process goes to zero almost surely in the asymptotic of large number of particles $N\rightarrow \infty$. Indeed, this is a consequence of the law of large number. A similar result is shown for the mean field equation of the online (streaming) PCA in [Chuang2017]. In the work of [Hu2018], the dynamic of each particle is governed by a linear stochastic linear system that involves a Brownian motion term. Therefore, the limiting measure of the empirical measure must satisfy a PDE that captures the effect of this diffusion term in the dynamic of each particle.
>
> Experiments:
> -Thank you. The bandwidths are not manually tuned by us in our paper. We used the bandwidths that are selected in the following paper:
> [1] MMD GAN: Towards Deeper Understanding of Moment Matching Network, NIPS 2017.
> We presume that these bandwidths in [1] are to provide the best scores.  The bandwidths can also be optimized using the techniques of [Arbel2018]. But, we suspect the results after bandwidth tuning would be the same as in [1]. We also mention that [Arbel2018]  consider the class of parametric kernels. In practice, the choice of this kernel class itself can create a model selection problem.
>
> -Thank you.  We point out that alpha is merely a scaling parameter that determines the separation of features after applying a kernel. If alpha is equal to infinity, this means that the learned kernel separates the features better than the kernel learned using a small alpha. In this regard, alpha=1 does not results in any special case. The discriminator for alpha=1 is still described by the test statistic given in Eqs. (28) and (29).
>
> - To address the concern of the reviewer, we have added more simulations in the revised manuscript using CelebA, and LSUN Bedroom datasets. The results are provided in Section B of the Supplementary in the revised paper. Unfortunately, we need more time to generate all the scores due to the need to optimize the kernel bandwidths parameters for MMD GAN.
>
> -To address this interesting point raised by the reviewer, we have added Figures 5 in the Supplementary. In Figure 5, we show the MMD value during the two phase procedure for kernel learning. In the first phase, we optimize the auto-encoder as in MMD GAN paper, and in the second phase, we train the kernel using the embedded features from auto-encoder. From Figure 5, it is clear that optimizing the kernel after auto-encoder is necessary and it significantly improves the MMD value. In Figure 4, we show the power of the test in hypothesis testing for high threshold values for the test statistic given in Equation (30). We observe the improvement in the statistical power due to the kernel training in Figure 4. Clearly, our two-phase method yield higher statistical power for larger values of the threshold in Figure 4(a) compared to the auto-encoder in Figure 4(b).

---

### Official Review · AnonReviewer3 · 2019-10-23
**Official Blind Review #3**

**Rating:** 6

**Review:**

The paper addresses the problem of kernel learning in MMD GAN using particle stochastic gradient descent to solve an approximation of the intractable distributional optimization problem for random features. The paper provides theoretical guarantees for the consistency of approximations, although proofs are deferred to Appendix.
It seems to be a good result theoretically thanks to the consistency guarantees for the particle SGD approximation of the optimization problem. However, its practical efficacy is not completely clear.
1. There is no discussion on how the method fares in terms of time/space complexity and if it is scalable to higher-dimensional datasets or larger batch sizes. How many steps T for good results are needed? How much time does it take to learn the model compared to the Implicit Kernel Learning or original MMD GAN?
2. For a more detailed analysis of performance, it would be helpful to see the benefits of the kernel learned with the proposed method on synthetic data and its performance on supervised learning tasks compared with other kernel learning methods on supervised tasks.

Some minor remarks:
1. Scaling parameter alpha has become parameter beta on page ix.
2. Some citations and equation references should be fixed.

**Experience Assessment:**

I have read many papers in this area.

**Review Assessment: Checking Correctness Of Derivations And Theory:**

I assessed the sensibility of the derivations and theory.

**Review Assessment: Checking Correctness Of Experiments:**

I carefully checked the experiments.

**Review Assessment: Thoroughness In Paper Reading:**

I read the paper at least twice and used my best judgement in assessing the paper.

---

> ### Author Response · Authors · 2019-11-15
> **Simulations on synthetic data are provided in Supplementary.**
>
> Thank you for your constructive feedback. We appreciate you for noting the strength of the theoretical results and appreciate the opportunity to improve the clarity so that the paper can be made accessible. We address the specific questions you raised below and will incorporate that feedback into the updated version of the paper:
>
>
>
> - We have addressed the concern of the reviewer by adding Subsection 3.4 about the computational complexity of our kernel learning method. Notice that in the implicit kernel learning as well as the MMD GAN papers, the kernel is learned via applying a SGD to the empirical MMD and a batch size of larger than 2 is used. In our method, the batch size is fixed at 2 and the complexity of our method also depends on the number of particles that is used. So, a direct comparison between the complexity of our method and that of MMD GAN and implicit kernel learning is not
>
>
> - Thank you for your suggestion. In the supplementary, we have indeed provided more experiments on synthetic data-set for a two-sample test between two multivariate Gaussian distributions with different covariances. Indeed, in Figure 4, we have shown that the kernel learning method can improve the statistical power of the hypothesis test in Eq. (30) for larger values of the threshold. It is clear from Figure 4 and the new Figure 5 added in the revised draft that our kernel learning method improves the MMD value, beyond what is attainable by an auto-encoder. We agree with the reviewer that our kernel learning approach can also be evaluated for supervised tasks such as in the kernel support vector machines. However, the focus of this paper is on generative models for unsupervised learning problems.
>
> -Thank you. The typo in the scaling parameter is fixed.
> -Thank you. The citations are fixed.

---

### Official Review · AnonReviewer1 · 2019-10-26
**Official Blind Review #1**

**Rating:** 6

**Review:**

This paper aims to improve the kernel selection issue of the MMD-based generative models. The author formulates the kernels via inverse Fourier transform and the goal is to learn the optimal N finite random Fourier features (RFF). The RFF samples are optimized by the proposed kernel alignment loss where the positive and negative labels are defined as samples coming from real and negative data distributions, respectively. Some theoretical analysis regarding the consistency of the learned kernel is provided. Experiment results on the IS score and FID on CIFAR-10 show improvement of the proposed methods over MMD-GAN baselines, while the results are not comparable to the original MMD-GAN due to unknown results.

While motivated from the mean-field theory, the algorithm 1 is essentially doing stochastic gradient on the RFF samples with fixed learning rate. Learning spectral distribution of kernel via optimising RFF samples is also not entirely new, as [0] presented in the Appendix C4 of [1]. they show the difference between two different realization of kernel learning.

I would love to increase my score if the author could address the following comments:
(1) Can you explain why the IS and FID results of MMD-GAN presented in Table 1 is inconsistent (i.e. considerably worse) with other papers [1,2,3]?
(2) In experiment setting, the learning rate eta of learning RFF samples is fixed to 10. Does this guarantee the learned spectral distribution lying in the constraint set P as specify in Eq (8)?

[1] Implicit kernel learning, AISTATS 2019
[2] DEMYSTIFYING MMD GANS, ICLR 2018
[3] MMD GAN: Towards Deeper Understanding of Moment Matching Network, NIPS 2017


**Experience Assessment:**

I have published one or two papers in this area.

**Review Assessment: Checking Correctness Of Derivations And Theory:**

I assessed the sensibility of the derivations and theory.

**Review Assessment: Checking Correctness Of Experiments:**

I carefully checked the experiments.

**Review Assessment: Thoroughness In Paper Reading:**

I read the paper at least twice and used my best judgement in assessing the paper.

---

> ### Author Response · Authors · 2019-11-15
> **The discrepancy in FID and IS scores is (potentially) caused by the implementation of the Inception-v3 network.**
>
> We thank the reviewer for positive feedback and constructive comments. Please find the answer to your comments below:
>
> 1- With regard to the Inception Score (IS), we used the standard implementation of IS score using the codes released by the authors of the following paper on Github:
>
> [1] Improved techniques for training gans. In NIPS, 2016.
>
> The authors of [2] mention in their work that they also use the same implementation to compute their inception scores.
>
> [2] MMD GAN: Towards Deeper Understanding of Moment Matching Network, NIPS 2017.
>
> However, we noticed some discrepancies between the scores reported in [1] and [2] for the implementation on the real data. Our IS is in agreement with that of [1]. Indeed, our IS for the real data (CIFAR-10 images) is 11.237±.116, which is in close agreement with the number 11.24 ± .12 reported in [1]. In contrast, the number reported in [2] for the real data is 11.95±0.20. We suspect this discrepancy between the numbers in [1] or [2] is due to the different implementation of the Inception-v3 Network in Pytorch and Tensorflow. Unfortunately, the authors of [2] have not released their Python codes for the inception score calculation. So we can only speculate about the cause of this discrepancy.
>
> With regard to the implementation of FID score, we used the TTUR codes released by reference  [3] on Github:
>
> [3] Gans trained by a two time-scale update rule converge to a local Nash equilibrium, NIPS 2017,
>
> We suspect the discrepancy in the reported FID scores in our paper and other works is due to the following reasons:
>
> -We note that the FID score is computed by measuring the mean and covariance statistics of generated and real data using the 2048-dimensional activations of the Inception-v3 pool3 layer. Once again, different implementations of the Inception-v3 in Pytorch and Tensorflow can causes a problem.
>
> -The FID score is sensitive to the number of generated and real data samples that is used for its calculation. For instance, consider the simple experiment where we use the samples from CIFAR-10 training data-set as the real images and the CIFAR-10 test data-set as the generated images. In theory, the FID score is expected be zero since real and generated images both are from the same data-set. Nevertheless, in practice, for a finite number of samples from each data-set, the FID score is different from zero. Indeed, using 100 samples from each of the training and test data-sets, we calculated the FID score to be 176.916. For 1000 samples, the FID score reduces to 49.340. From this simple experiment, it is clear that the FID is very sensitive to the number of samples. In our experiments, we used 50000 samples from fake and real images.
>
> 2-The result in Corollary 4.2.1. is indeed a statement about the order of $\eta$, and therefore the exact value of $\eta$ cannot be derived from this Corollary. Furthermore, Corollary 4.2.1 is a probabilistic bound, and depending on the choice of the confidence level, different bounds for $\eta$ can be derived. Nevertheless,  Corollary 4.2.1 implies that when $R$ is sufficiently large, or $\eta$ is sufficiently small, the feasibility constraint on the empirical measure can be satisfied. This can be used as a guideline for choosing the step-size in numerical experiments. In our simulations,  a large value for $R$ is used which guarantees the feasibility of the empirical distribution for $\eta=10$. Thanks to the result of Thm. 4.1, the consistency still can be guaranteed for a large $R$ if the number of training samples $n$ is also large (which is the case for CIFAR-10 and MNIST).  In practice and for the general case, a backtracking line search is needed to adequately compute the step-size that satisfies the distribution ball constraint on the empirical measures.

---

### Decision · Program_Chairs · 2019-12-19

**Decision:**

Reject

**Comment:**

This paper was assessed by three reviewers who scored it as 6/1/6. The main criticism included somewhat weak experiments due to the manual tuning of bandwidth, the use of old (and perhaps mostly solved/not challenging) datasets such as Mnist and Cifar10, lack of ablation studies. The other issue voiced in the review is that the proposed method is very close to a MMD-GAN with a kernel plus random features. Taking into account all positives and negatives, we regret to conclude that this submission falls short of the quality required by ICLR2020, thus it cannot be accepted at this time.